# RETHINKING REWARD MODELS FOR MULTI-DOMAIN TEST-TIME SCALING

## ABSTRACT

The reliability of large language models (LLMs) during test-time scaling is often assessed with *external verifiers* or *reward models* that distinguish correct reasoning from flawed logic. Prior work generally assumes that process reward models (PRMs), which score every intermediate reasoning step, outperform outcome reward models (ORMs) that assess only the final answer. This view is based mainly on evidence from narrow, math-adjacent domains. We present the first unified evaluation of four reward model variants, discriminative ORM and PRM (**dO**RM, **dP**RM) and generative ORM and PRM (**gO**RM, **gP**RM), across 14 diverse domains. Contrary to conventional wisdom, we find that (i) **dO**RM performs on par with **dP**RM, (ii) **gP**RM is not competitive, and (iii) overall, **gO**RM is the most robust, yielding significant and consistent gains across every tested domain. We attribute this to PRM-style stepwise scoring, which inherits label noise from LLM auto-labeling and has difficulty evaluating long reasoning trajectories, including those involving self-correcting reasoning. Our theoretical analysis shows that step-wise aggregation compounds errors as reasoning length grows, and our empirical observations confirm this effect. These findings challenge the prevailing assumption that fine-grained supervision is always better and support generative outcome verification for multi-domain deployment. We publicly release our code, datasets, and checkpoints at this anonymous repository to facilitate future research in multi-domain settings.

## 1 INTRODUCTION

Test-time scaling (TTS) enables large language models (LLMs) to generate diverse, reliable solutions, *i.e.*, chain-of-thought (CoT; Wei et al., 2022; Kojima et al., 2022; Yao et al., 2023b; Madaan et al., 2023) and has shown impressive results on challenging reasoning tasks (Yao et al., 2023a; Snell et al., 2025; Wu et al., 2024). A widely adopted TTS approach uses *external verifiers* that select the best among the candidates (Snell et al., 2025): these verifiers, outcome reward models (ORMs), are typically implemented as discriminative classifiers that assign a scalar *reward* to a CoT (Cobbe et al., 2021; Uesato et al., 2022; Yu et al., 2024). ORMs are trained only on outcome-level signals, which are often coarse. Recent work has introduced process reward models (PRMs; Lightman et al., 2024; Wang et al., 2024a; Setlur et al., 2025; Zheng et al., 2024) that score each step of a CoT and aggregate the scores into a trajectory-level reward. Supervised with high-quality, carefully constructed process labels, *e.g.*, manual annotation (Lightman et al., 2024) or Monte Carlo rollouts (Wang et al., 2024a), PRMs have been shown to outperform ORMs when combined with TTS.

Beyond discriminative verifiers, several studies have shown that the generative ability of LLMs can improve CoT verification, such as *LLM-as-a-judge* (Wang et al., 2023; Liu et al., 2023; Zheng et al., 2023). Based on this idea, other works fine-tune LLMs to generate a verification rationale for a CoT and compute the final reward from token probabilities (Zhang et al., 2025a; Khalifa et al., 2025; Zhao et al., 2025). To obtain verification CoTs for training, most previous work adopts *consensus-filtering*: (i) generate verification CoTs, and (ii) retain the verification CoT if its parsed verdict aligns with outcome or process labels. After training, these generative verifiers have shown strong performance in math-adjacent reasoning tasks, outperforming discriminative verifiers.

However, most of the research efforts on TTS with external verifiers have been devoted primarily to math-adjacent domains. This narrow scope limits the potential for LLM deployment in high-stakes real-world applications, such as legal (Guha et al., 2023; Cui et al., 2023; Fei et al., 2024) and medical (Singhal et al., 2023; Kung et al., 2023; Singhal et al., 2025) domains, where trustworthiness is paramount and rigorous verification of LLM outputs is especially important. Recently, Zeng

et al. (2025) proposed multi-domain PRMs trained on the graduate level benchmark (MMLU-Pro; Wang et al., 2024c), covering 14 diverse domains, and showed that multi-domain training for PRMs significantly improves TTS performance across diverse domains. However, the study is *limited* to discriminative PRMs and the broader potential of different verifier types (*e.g.*, ORMs vs. PRMs, discriminative vs. generative) in the multi-domain setting still remains *underexplored*.

To this end, we present the first unified evaluation of *four verifier variants*, discriminative ORM and PRM (**dO**RM, **dP**RM), and generative ORM and PRM (**gO**RM, **gP**RM), across 14 diverse domains. We rigorously review these variants in §2 and, under controlled conditions, evaluate them on math and multi-domain benchmarks (PRM800K, ProcessBench, and MMLU-Pro; Lightman et al., 2024; Zheng et al., 2024; Wang et al., 2024c) in §3. In the math domain, trends across the four variants are consistent with previous works (Lightman et al., 2024; Zhang et al., 2025a; Khalifa et al.,

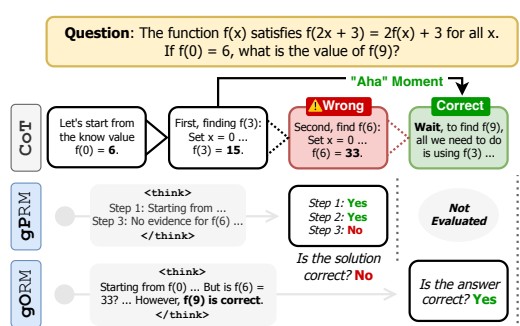

Figure 1: **Evaluating CoTs using gORM and gPRM.**

2025). **dP**RM outperforms **dO**RM, and generative variants outperform discriminative ones. In the multi-domain, however, we observe contrasting results. **dO**RM performs on par with **dP**RM, **gP**RM is not competitive, and overall, **gO**RM delivers **consistent and significant gains** over the others.

In §4, we identify two factors underlying the failure of **gP**RM. First, on more difficult multi-domain problems, LLMs tend to produce longer CoTs which PRMs struggle to evaluate. As illustrated in Fig. 1, stepwise aggregation in PRMs often fails to reward long CoTs that recover from earlier errors ("aha" moments; Guo et al., 2025), because verification stops at the first mistake. In §4.1, we analyze how this PRM-style aggregation compounds errors as the chain length increases, and confirm this effect with our empirical results. Second, label noise is prevalent in multi-domain datasets. Given that step annotation in specialized domains is costly, prior work such as Zeng et al. (2025) depends on LLM-based auto-labeling, which can introduce noise. In §4.2, we show that, under a simulated label-noise analysis in the math domain, **dO**RM is particularly sensitive to noisy step labels, whereas **gO**RM remains robust. We further attribute the degradation of **gP**RM in the multi-domain setting, despite its robustness to noise in the math domain, to a severe shift in its CoT-length distribution induced by *consensus filtering*. Based on this analysis, we present practical guidelines for selecting among the four variants and discuss limitations and future work in §5.

Our contributions and findings are summarized as follows:

- We present the first unified and controlled evaluation of four verifier variants, **dO**RM, **dP**RM, **gO**RM, and **gP**RM, across 14 diverse domains.

- In contrast to conventional wisdom in math, we observe that (i) **dO**RM performs similarly to **dP**RM, (ii) **gP**RM is not competitive; and (iii) overall, **gO**RM **delivers consistent gains** over the others.

- To explain the empirical observations, we provide two perspectives: (i) a theoretical analysis, with empirical support, showing that *PRM risk increases with CoT length*; and (ii) evidence of *process label risk* in the multi-domain setting with length-distribution shift induced by *consensus filtering*.

## 2 BACKGROUND AND RELATED WORK

In this section, we review background and related work. We first formalize notation and test-time scaling in §2.1, and then discuss reward-model variants in §2.2, summarized in Fig. 2.

### 2.1 PROBLEM FORMULATION

**Notation.** For a given question $q$ with the corresponding ground-truth (GT) answer $a$, we leverage the reasoning ability of large language models (LLMs) to reliably predict $a$ by generating a chain-of-thought (CoT), *i.e.*, $r_{1:T} := (r_1, \ldots, r_T) \sim p_{\text{LLM}}(\cdot \mid q)$. Following Zeng et al. (2025), we segment the reasoning steps $r_{1:T}$ using the delimiter "\n\n", where $T$ is the number of reasoning steps. Let $x := (q, r_{1:T}) \in \mathcal{X}$, where $\mathcal{X}$ denotes the space of questions and reasoning chains, and let $x_{1:t} := (q, r_{1:t})$ be the prefix up to the $t$-th step. We consider two types of labels: (1) the

Figure 2: **Conceptual illustration of reward models**: $r_2$ is the first incorrect step; the final answer is correct.

*outcome label* $y = \mathbb{1}(\hat{a}(r_T) = a) \in \{0,1\}$, where $\hat{a}(r_T)$ is the predicted answer parsed from the last reasoning step $r_T$ and $\mathbb{1}$ is the indicator function; and (2) the *process labels* $z_{1:T} = (z_1, \ldots, z_T) \in \{0,1\}^T$, where each $z_t$ indicates whether the corresponding reasoning step $r_t$ is correct. Note that $y$ represents the correctness label for the last reasoning step, so $y = z_T$.

**Test-time scaling (TTS) with reward models.** Reward models have many applications, including LLM training via reinforcement learning (Ziegler et al., 2019; Ouyang et al., 2022; Achiam et al., 2023; Dubey et al., 2024; Team et al., 2024; Yang et al., 2025), preference labeling (Dong et al., 2024; Meng et al., 2024; Adler et al., 2024), rejection sampling (Gulcehre et al., 2023; Dong et al., 2023), and data filtering (Dubey et al., 2024; Albalak et al., 2024; Yang et al., 2025). In this work, we focus on parallel or sampling-based (Wu et al., 2024) TTS with reward models, such as Best-of-$N$ (Bo$N$; Charniak & Johnson, 2005; Khalifa et al., 2023; Snell et al., 2025), which allocates more compute at test time (*i.e.*, generates $N$ CoTs) and selects the candidate $\hat{a}(r_T^{(i_\star)})$ with the the *reward*:

$$i_\star = \underset{i \in \{1,\ldots,N\}}{\arg\max} f(x^{(i)}), \quad \text{where } x^{(i)} := (q, r_{1:T}^{(i)}), \text{ and } r_{1:T}^{(i)} \overset{\text{i.i.d.}}{\sim} p_{\text{LLM}}(\cdot \mid q). \tag{1}$$

Here, $f : \mathcal{X} \to [0,1]$ is the *true (unknown) reward function* that assigns higher scores to CoTs that yield more reasonable and correct answers. However, $f$ is unknown, so we train an *external verifier* $\hat{f} : \mathcal{X} \to [0,1]$ to approximate $f$ and use $\hat{f}$ as a surrogate in Eq. 1, which are detailed in §2.2.

## 2.2 REWARD MODELS

**Discriminative outcome reward model (dORM).** Early studies on reward models (Cobbe et al., 2021; Uesato et al., 2022; Yu et al., 2024) train a binary classifier $\hat{f}_{\text{dORM}} : \mathcal{X} \mapsto [0,1]$ on *outcome labels* $y \in \{0,1\}$ *only*, without requiring the intermediate process labels $(z_1, \ldots, z_{T-1})$. Specifically, they sample CoTs and answers for given questions, construct a training dataset $\mathcal{D}_{\text{dORM}} := \{(x,y)\}$, and train $\hat{f}_{\text{dORM}}$ with the binary cross-entropy (BCE) loss to approximate true $p(y = 1 \mid x)$:

$$\mathcal{L}_{\text{dORM}} := \frac{1}{|\mathcal{D}_{\text{dORM}}|} \sum_{(x,y) \in \mathcal{D}_{\text{dORM}}} \ell_{\text{BCE}} \left( \hat{f}_{\text{dORM}}(x), y \right), \tag{2}$$

with $\ell_{\text{BCE}}(x,y) = -[y \log x + (1-y) \log(1-x)]$. dORM considers only outcome correctness and ignores step-wise accuracy, making its reward signal potentially less faithful than $f(x)$ in Eq. 1.

**Discriminative process reward model (dPRM).** dPRM seeks to improve the reward signal by training on fine-grained feedback for intermediate reasoning steps, *i.e.*, *process labels* $z_{1:T}$. For dPRM, the quality of these labels is the primary factor. Accordingly, prior work has proposed collecting process labels for sampled CoTs via manual annotation (Lightman et al., 2024), Monte Carlo (MC) rollouts (Wang et al., 2024a), automatically generated labels from LLMs (Zeng et al., 2025), or combinations thereof (Zhang et al., 2025b). After collecting the process labels, we construct the training set $\mathcal{D}_{\text{dPRM}} := \{(x, z_{1:T})\}$ and train $\hat{f}_{\text{dPRM}}$ using the BCE loss at each step:

$$\mathcal{L}_{\text{dPRM}} := \frac{1}{|\mathcal{D}_{\text{dPRM}}|} \sum_{(x,z_{1:T}) \in \mathcal{D}_{\text{dPRM}}} \frac{1}{T'} \sum_{t=1}^{T'} \ell_{\text{BCE}} \left( \hat{f}_{\text{dPRM}}(x_{1:t}), z_t \right), \tag{3}$$

where $T'$ is the first incorrect reasoning step, *i.e.*, $T' := \min(\{t \in \{1, \ldots, T\} : z_t = 0\} \cup \{T\})$. Training up to the $T'$-th step reflects a common assumption in the literature (Lightman et al., 2024; Wang et al., 2024a; Zheng et al., 2024; Zeng et al., 2025): once a reasoning step is incorrect, *subsequent steps are also incorrect*, *i.e.*, if $z_t = 0$ then $z_{t'} = 0$ for all $t' \in \{t+1, \ldots, T\}$. At test time, we approximate $f$ in Eq. 1 by aggregating the step rewards with the *minimum* (Zeng et al., 2025).

**LLM-as-a-judge** (Wang et al., 2023; Liu et al., 2023; Zheng et al., 2023) shows that the task-generalization ability of LLMs can extend to verification (*i.e.*, zero-shot CoT verification). However, LLMs often overthink (Bavaresco et al., 2025) and, without additional training, remain practically limited (Zheng et al., 2024), implying the need of LLMs explicitly trained for verification.

**Generative outcome reward model (gORM).** Zhang et al. (2025a) proposed gORM, trained to generate a *verification CoT* together with a binary verdict, *e.g.*, "Verification: Is the answer correct? Yes" or "No". Because GT verification CoTs are unavailable, they synthesize training data via a consensus-filtering mechanism (Wang et al., 2024b; Zhu et al., 2025). We first sample a verification CoT and verdict from an LLM-as-a-judge, *i.e.*, $v_{1:L} \sim p_{\text{LLM-j}}(\cdot \mid x)$ using the prompt format in Fig. 20. Here, $v_{1:L} \in \mathcal{V}^L$ denotes the verification-CoT token sequence (including the verdict tokens), $\mathcal{V}$ is the vocabulary, and let $\hat{y} \in \{0,1\}$ be the parsed verdict (1 for "Yes", 0 for "No"). We then include $(x, v_{1:L})$ in the training set $\mathcal{D}_{\text{gORM}}$ only if $\hat{y}$ agrees with the known outcome label $y$. We train $p_{\text{gORM}}$ with the next-token prediction over verification CoTs $v_{1:L}$:

$$\mathcal{L}_{\text{gORM}} := \frac{1}{|\mathcal{D}_{\text{gORM}}|} \sum_{(x,v_{1:L}) \in \mathcal{D}_{\text{gORM}}} \frac{1}{L} \sum_{i=1}^{L} -\log p_{\text{gORM}}(v_i \mid x, v_{<i}). \quad (4)$$

$-\log p_{\text{gORM}}$ is implemented as the cross-entropy loss over $\mathcal{V}$. At test time, we approximate $f$ with:

$$\hat{f}_{\text{gORM}}(x) := \mathbb{E}_{v_{1:L} \sim p_{\text{gORM}}(\cdot|x)}\big[p_{\text{gORM}}(y=1 \mid x, v_{1:L})\big] \approx \frac{1}{M} \sum_{i=1}^{M} p_{\text{gORM}}\left(y=1 \mid x, v_{1:L}^{(i)}\right), \quad (5)$$

where $v_{1:L}^{(i)} \overset{\text{i.i.d.}}{\sim} p_{\text{gORM}}(\cdot \mid x)$. Here, the expectation is approximated with $M$ MC samples and the model's normalized probability of predicting the verdict "Yes" at the last verdict position:

$$p_{\text{gORM}}(y=1 \mid v_{1:L}, x) := \frac{p_{\text{gORM}}(\text{"Yes"} \mid x, v_{1:(L-1)})}{p_{\text{gORM}}(\text{"Yes"} \mid x, v_{1:(L-1)}) + p_{\text{gORM}}(\text{"No"} \mid x, v_{1:(L-1)})}. \quad (6)$$

**Generative Process Reward Model (gPRM).** Beyond gORM, Khalifa et al. (2025) proposed gPRM, which is trained to generate verification CoTs $v_{1:L}$ with *stepwise process verdicts*, *e.g.*, "Step t: The step is \boxed{correct}" or "\boxed{incorrect}". Let the predicted verdict sequence be $\hat{z}_{1:T'} \in \{0,1\}^{T'}$, defined up to the first predicted incorrect step $T'$[1]. Following Khalifa et al. (2025), we append a final verdict prompt, yielding the token sequence $v_{1:L^+}$ by concatenating either "Is the solution correct? Yes" or "No"—"Yes" if all predicted process labels are 1 ($\hat{z}_{1:T'} = \mathbf{1}_{T'}$), and "No" otherwise. We then construct $\mathcal{D}_{\text{gPRM}} := \{(x, v_{1:L^+})\}$ only when the predicted prefix agrees with the GT ($\hat{z}_{1:T'} = z_{1:T'}$). We train $p_{\text{gPRM}}$ with $v_{1:L^+}$:

$$\mathcal{L}_{\text{gPRM}} := \frac{1}{|\mathcal{D}_{\text{gPRM}}|} \sum_{(x,v_{1:L^+}) \in \mathcal{D}_{\text{gPRM}}} \frac{1}{L^+} \sum_{i=1}^{L^+} -\log p_{\text{gPRM}}(v_i \mid x_{1:T'}, v_{<i}). \quad (7)$$

We condition on $x_{1:T'}$ rather than the full input $x$ for training (Khalifa et al., 2025), since the model $p_{\text{gPRM}}$ is prompted to stop verification once it reaches the first incorrect step, analogous to the data-generation process (Fig. 22). At test time, consistent with Eqs. 5 and 6, we approximate $f$ in Eq. 1 by sampling from $p_{\text{gPRM}}$ and computing the normalized probability of a positive final verdict:

$$\hat{f}_{\text{gPRM}}(x) := \mathbb{E}_{v_{1:L^+} \sim p_{\text{gPRM}}(\cdot|x)}\big[p_{\text{gPRM}}(y=1 \mid x, v_{1:L^+})\big] \approx \frac{1}{M} \sum_{i=1}^{M} p_{\text{gPRM}}\left(y=1 \mid x, v_{1:L^+}^{(i)}\right), \quad (8)$$

$$p_{\text{gPRM}}(y=1 \mid x, v_{1:L^+}) := \frac{p_{\text{gPRM}}(\text{"Yes"} \mid x, v_{1:(L^+-1)})}{p_{\text{gPRM}}(\text{"Yes"} \mid x, v_{1:(L^+-1)}) + p_{\text{gPRM}}(\text{"No"} \mid x, v_{1:(L^+-1)})}, \quad (9)$$

where $v_{1:L^+}^{(i)} \overset{\text{i.i.d.}}{\sim} p_{\text{gPRM}}(\cdot \mid x)$ and we now condition on the full input $x$ at test time (Khalifa et al., 2025). Byeond Khalifa et al. (2025), Zhao et al. (2025) also proposed a gPRM with code verification and more advanced training; however, it does not directly extend to multi-domain data (*e.g.*, legal or medical domains), so we follow the approach of Khalifa et al. (2025) in this work.

## 3 EXPERIMENTS

In this section, we evaluate dORM, dPRM, gORM, and gPRM in the math domain and the multi-domain setting. We detail experimental setups (§3.1), and present experimental results (§3.2).

---

[1]As shown in Fig. 21, when generating verification CoTs for gPRM (*i.e.*, $v_{1:L} \sim p_{\text{LLM-j}}(\cdot \mid q)$), Khalifa et al. (2025) instruct the LLM-as-a-judge $p_{\text{LLM-j}}$ to stop once it detects the first incorrect step.

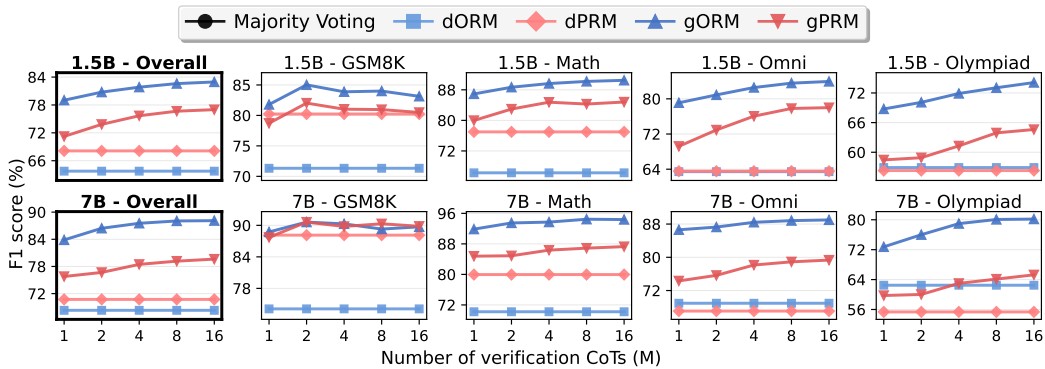

Figure 3: **Outcome verification results** on ProcessBench in the math domain.

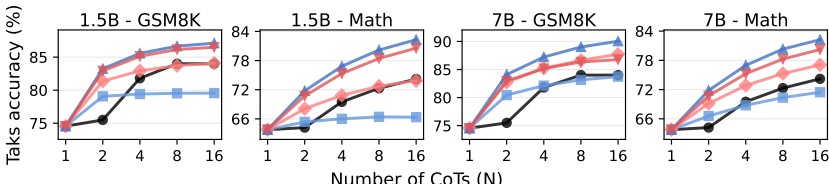

Figure 4: **Best-of-$N$ results** using Qwen2.5-7B-Instruct on GSM8K and Math in the math domain.

## 3.1 EXPERIMENTAL SETUPS

**Math Datasets.** For the math domain, we use **PRM800K** (Lightman et al., 2024) for training, where the process labels $z_{1:T}$ are human-annotated. As a testbed, we use **ProcessBench** (Zheng et al., 2024) with four splits: GSM8K/Math/Omni-Math/OlympiadBench. We generate $N=16$ CoTs per question in GSM8K and Math with Qwen2.5-7B-Instruct (Team, 2024a) for TTS; since we only seek to verify that a controlled evaluation reproduces prior findings, we restrict TTS to this setting.

**Multi-domain datasets.** Following Zeng et al. (2025), we adopt **MMLU-Pro** (Wang et al., 2024c), a 10-choice benchmark spanning *14 domains*. For training/evaluation of reward models, each question is paired with 16/128 CoTs generated by Llama-3.1-8B-Instruct (Dubey et al., 2024), where process labels $z_{1:T}$ are *automatically annotated* by Llama-3.1-70B-Instruct. To assess generalization across different $p_{\text{LLM}}$, we generate $N=16$ CoTs per question using SmolLM3-3B (Bakouch et al., 2025), Qwen2.5-7B-Instruct, gemma-2-9b-it (Team et al., 2024), and Llama-3.1-70B-Instruct. We defer further details, including prompts and dataset statistics to §C.

**Implementation details.** For reward-model backbones, we use R1-Distill models (Guo et al., 2025): Qwen-1.5B and Qwen-7B/Llama-8B and Qwen-14B in the math/multi-domain settings. For prompt templates of **gO**RM/**gP**RM, we follow Zhang et al. (2025a)/Khalifa et al. (2025) (Figs. 20 and 22). We optimize reward models using AdamW (Loshchilov & Hutter, 2019) with LoRA (Hu et al., 2022). Following Zeng et al. (2025), we pre-train both **dO**RM/**dP**RM on PRM800K with the **dP**RM objective (*cf.* Eq. 3) before multi-domain training. For **gO**RM and **gP**RM, we sample $M=16/10$ verification CoTs (*cf.* Eqs. 5 and 8) in the math/multi-domain settings, using vLLM (Kwon et al., 2023). See §D and Tab. 2 for more details, and this repository for anonymized artifacts.

**Verification CoTs.** Following Zhang et al. (2025a) and Khalifa et al. (2025), we construct verification–CoT datasets for **gO**RM and **gP**RM by prompting QwQ-32B (Qwen Team, 2025) with the formats in Figs. 20 and 21. We discard any verification CoT whose parsed labels are inconsistent with the targets (*e.g.*, $y$ or $z_{1:T}$), corresponding to the *consensus filtering* in §2.2. The training sets of **gO**RM/**gP**RM contain 34,286/35,666 and 171,780/94,156 verification CoTs for the math and multi-domain settings. See §D and Figs. 23 and 24 for more details and examples.

## 3.2 EXPERIMENTAL RESULTS

**Math-domain results.** First, we evaluate the four verifier variants in the math domain. We compare outcome-verification performance with a 0.5 decision threshold, *i.e.*, $\hat{y} := \mathbb{1}(\hat{f}(x) > 0.5)$. Fig. 3 reports F1 score (%) on ProcessBench splits. **dP**RM outperforms **dO**RM overall, consistent with prior findings (Lightman et al., 2024), and shows a slight drop in Omni-Math/OlympiadBench with 7B backbones. For **gO**RM/**gP**RM, the overall performance improves with $M$. At small $M$, **gP**RM may lag behind discriminative models (*e.g.*, OlympiadBench). **gO**RM generally outperforms **gP**RM (except 7B-GSM8K), and the gap widens on Omni-Math/OlympiadBench.

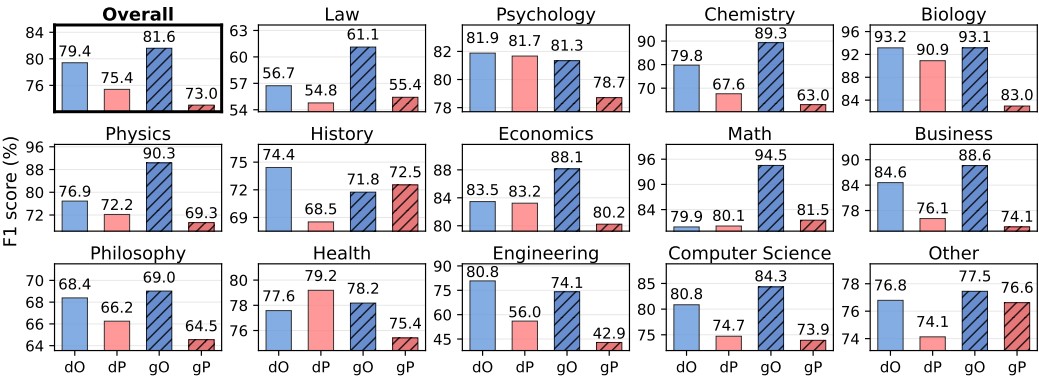

Figure 5: **Outcome verification results** on MMLU-Pro in the multi-domain setting.

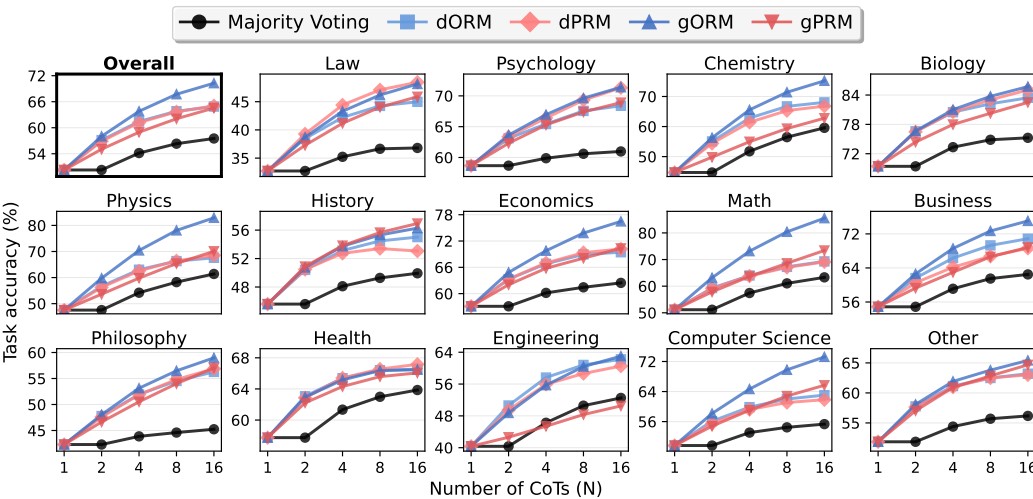

Figure 6: **Best-of-$N$ results** using **Llama-3.1-8B-Instruct** on MMLU-Pro in the multi-domain setting.

Although TTS has been well studied in the math domain, evaluations are not fully controlled: (i) models are rarely compared with a shared backbone, and (ii) **g**ORM and **g**PRM have not been directly compared. We therefore evaluate the reward models with Bo$N$ under controlled conditions. As shown in Fig. 4, and consistent with the findings of Lightman et al. (2024), **d**PRM outperforms **d**ORM. Notably, **d**ORM even underperforms majority voting (MV) with 1.5B backbones, demonstrating the limitations of coarse outcome-level supervision. In line with Zhang et al. (2025a) and Khalifa et al. (2025), generative models outperform discriminative ones, with **g**ORM slightly surpassing **g**PRM.

**Multi-domain results.** Next, we compare the four variants in the multi-domain setting. Fig. 5 reports F1 scores (%) for outcome-verification, with a 0.5 decision threshold, using R1-Distill-Qwen-14B as the reward model backbone. dO/dP/gO/gP denote **d**ORM/**d**PRM/**g**ORM/**g**PRM. In contrast to the math domain results in Fig. 3, ORM variants achieve higher F1 scores than PRM variants.

Fig. 6 shows the Bo$N$ performance using Llama-3.1-8B-Instruct as $p_{LLM}$ and R1-Distill-Qwen-14B as the reward model backbone. In this setting, **d**ORM *performs comparably* to **d**PRM, while **g**PRM is *not competitive*, which is contrary to previous work (Lightman et al., 2024; Khalifa et al., 2025) and our math-domain results in Fig. 4. Overall, **g**ORM outperforms **d**ORM/**d**PRM/**g**PRM, without notable degradation in any domain relative to the others. As shown in Fig. 7, this trend holds across different $p_{LLM}$. Using 8B reward backbones, we observe the same pattern (Figs. 28 and 29). Detailed per-domain and weighted majority voting results are provided in §F.

**Effect of multi-domain training.** To assess the effect of multi-domain training, we train and evaluate all four variants *only* on each MMLU-Pro domain and compare each variant to its multi-domain counterpart. Fig. 8 reports the degradation of Bo$N$ performance with $N=16$ under domain-specialized training. We observe *severe drops* for **d**ORM/**d**PRM, with a slightly larger decline for **d**ORM, likely because outcome-only supervision is sparser than step-level supervision and both are relatively data-hungry. In contrast, **g**ORM and **g**PRM appear more *sample-efficient*: even without multi-domain training, their performance decreases only modestly (or even increases), demonstrat-

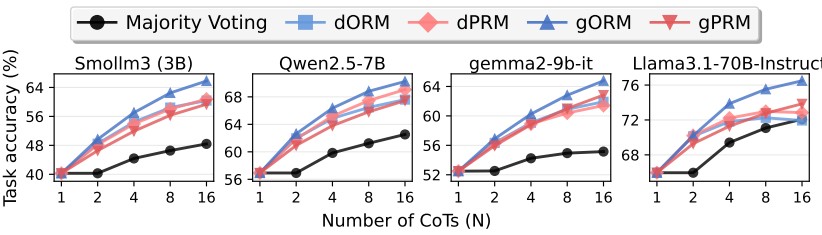

Figure 7: **Overall Best-of-$N$ results using four different $p_{\text{LLM}}$ on MMLU-Pro in the multi-domain setting.**

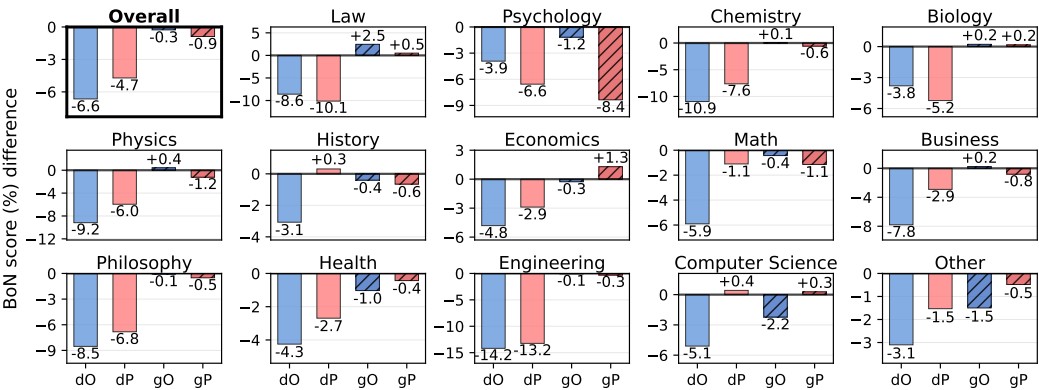

Figure 8: **Best-of-$N$ performance gap between all-domain and single-domain training on MMLU-pro.**

ing the efficiency of generative reward models. We defer complete results of single-domain training for four reward models to Figs. 26 and 27 of §F.

## 4 ANALYSIS ON WHY PRMS FAIL IN MULTI- DOMAIN

In this section, we analyze the failure modes of PRMs observed in the multi-domain setting of §3.

### 4.1 RISK OF PRMS WITH COT LENGTH

**"Aha" CoTs.** As noted in §2.2, PRMs typically assume that once a reasoning step is incorrect, *all subsequent steps are incorrect*. However, recent reasoning models can recover from earlier mistakes and still arrive at the correct answer (an **"aha" moment**;

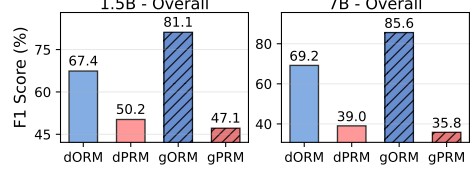

Figure 9: **Results on "aha" CoTs.**

Guo et al., 2025). In such cases, PRMs can miss the recovery due to a monotonicity bias induced by their training data. To demonstrate this, we evaluate on "aha" CoTs from ProcessBench that contain at least one incorrect step ($\exists\, t \in \{1, \dots, T\} : z_t = 0$) but a correct outcome ($y = 1$). Overall, "aha" CoTs account for 15.3% of the cases. In Fig. 9, we report F1 scores (%) for the "aha" CoTs using $M{=}16$ for **g**ORM/**g**PRM. We observe that PRM variants perform particularly **poorly on "aha" CoTs**. Moreover, scaling the backbone from 1.5B to 7B improves ORM performance, whereas PRM performance degrades with larger backbones, possibly because larger PRMs are more likely to follow the PRM assumption inherent in their training data and objective (*cf.* Eqs. 3 and 7).

**Do ORMs overfit on "aha" CoTs?** A natural concern about ORM results on "aha" CoTs in Fig. 9 is *overfitting*: ORMs might only memorize questions and their answers, thereby correctly verifying "aha" CoTs without checking the correctness of intermediate reasoning steps. This memorization issue in the math domain has recently been studied by Wu et al.

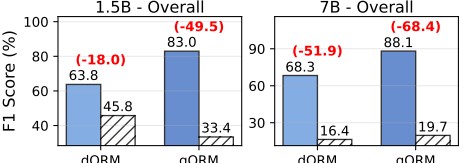

Figure 10: **Results on randomly shuffled CoTs.**

(2025). To investigate this, we conduct the following test: (i) **replace the intermediate reasoning steps** $r_{1:T-1}$ with $r'_{1:T-1}$ **taken from other CoTs**, and (ii) evaluate ORMs on these perturbed CoTs. If ORMs only memorize the answer in the final reasoning step $r_T$, their performance should remain largely unaffected. However, Fig. 10 shows a **significant drop** for ORMs (dashed), indicating the reliance on intermediate steps. Interestingly, the degradation is greater with the 7B backbone than with the 1.5B backbone for both **d**ORM and **g**ORM. This suggests that larger models rely *more heavily* on intermediate reasoning steps during verification.

**Risk increases with CoT length.** "Aha" moments can also lengthen CoTs, an effect especially pronounced in the multi-domain setting (Fig. 11), where LLMs struggle more than in math. As shown in Fig. 38, majority voting results degrade significantly with increasing CoT length in the multi-domain setting. Consistent with the outcome-verification failures of PRMs

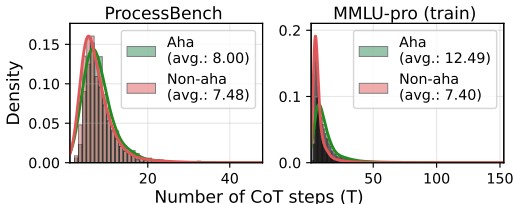

Figure 11: **Length dist. of "aha" CoTs**.

on "aha" CoTs, we argue that *the error of PRM variants grows with CoT length* ($T$). Intuitively, as a CoT grows longer, the chance that a PRM misclassifies at least one intermediate step rises, making it more likely to *prematurely* conclude the CoT is incorrect. Longer CoTs also create more opportunities for "aha" recoveries that PRMs systematically miss. We formalize this as follows:

**Theorem 4.1** (Log-error bound of **d**O**RM** and **g**O**RM**). *Let $\epsilon \in \{\epsilon_d, \epsilon_g\}$ with $\epsilon_d := \log \hat{f}_{dORM}(x) - \log f(x)$ and $\epsilon_g := \log \hat{f}_{gORM}(x) - \log f(x)$. Define $\bar{m} := \mathbb{E}[\epsilon \mid x], \bar{\xi} := \epsilon - \bar{m}, \beta_{orm}^2 := \mathbb{E}[\bar{m}^2]$. Then $\epsilon = \bar{m} + \bar{\xi}$ with $\mathbb{E}[\bar{\xi} \mid x] = 0$. If $\mathrm{Var}(\bar{\xi} \mid x) \leq \tau_{orm}^2$ for some constant $\tau_{orm}^2$ independent of $T$, then $\mathbb{E}[\epsilon^2] = \mathbb{E}\left[\mathrm{Var}(\bar{\xi} \mid x)\right] + \mathbb{E}[\bar{m}^2] \leq \tau_{orm}^2 + \beta_{orm}^2$.*

**Theorem 4.2** (Log-error lower bound of **d**P**RM**). *Let $\Delta_{dPRM} := \log \hat{f}_{dPRM}(x) - \log f(x)$. Under the assumptions in §A.1, $\mathbb{E}\left[\Delta_{dPRM}^2\right] \geq (\sigma^2 - 2\gamma)T$.*

**Theorem 4.3** (Log-error lower bound of **g**P**RM**). *Let $\Delta_{gPRM} := \log \tilde{f}_{gPRM}(x) - \log f(x)$, where $\tilde{f}_{gPRM}(x)$ is approximation of $\hat{f}_{gPRM}$ in Eq. 8 with a single MC sample. Under the assumptions in §A.1, $\mathbb{E}\left[\Delta_{gPRM}^2\right] \geq (\sigma^2 + \tau^2 - 2\gamma)T$.*

All details and proofs are deferred to §A. Theorem. 4.1, 4.2 and 4.3 establish log-error bounds for the four reward-model variants. The **d**O**RM**/**g**O**RM** bounds are independent of $T$, whereas the **d**P**RM**/**g**P**RM** lower bounds grow linearly with $T$. In Theorem. A.4, we also show that for **g**P**RM** with Monte Carlo estimation (*cf.* Eq. 5), the log-error lower bound increases linearly with $T$.

**Empirical support.** To empirically support Theorems. 4.1 to 4.3, we plot the F1 score (%) for outcome-verification in the multi-domain setting as a function of the number of reasoning CoT steps ($T$) in Fig. 12-(**Left**). We bin CoTs into eight categories. As $T$ increases, **d**P**RM**/**g**P**RM** degrade considerably relative to **d**O**RM**/**g**O**RM**. Fig. 12-(**Right**) shows the performance improvements over majority voting of different categories w.r.t the average number of

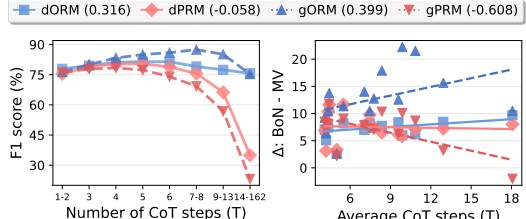

Figure 12: (**Left**): Outcome verification vs. CoT length; (**Right**): TTS improvement vs. average CoT length.

CoT steps. We observe negative correlations for **d**P**RM** (-0.058) and **g**P**RM** (-0.608), while **d**O**RM** (0.316) and **g**O**RM** (0.399) shows rather positive correlations. These results not only provide empirical support for Theorems. 4.1 to 4.3 but also demonstrate that increasing CoT length can degrade TTS performance for **d**P**RM** and **g**P**RM** in the multi-domain setting.

## 4.2 LABEL NOISE OF PRMS

**Label noise risk.** Beyond CoT-length effects, *label noise* poses an additional risk, especially in multi-domain settings. Since human annotation of long CoTs is more costly in specialized domains such as law and medicine than in math, prior work often relies on LLMs to auto-label process steps (Zeng et al., 2025), which introduces noise that can degrade PRM performance. We study this by injecting synthetic noise into the process labels of

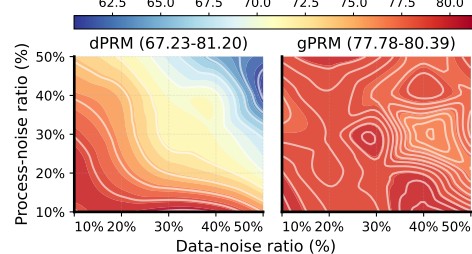

Figure 13: **Effect of labe noise** on GSM8K.

PRM800K. We vary the level of noise along two axes: (i) *process-noise ratio* (the per-step probability of flipping a process label) and (ii) *data-noise ratio* (the fraction of examples to which noise is applied). We report the outcome-verification F1 score (%) in Fig. 13 using 1.5B backbones, using greedy decoding for generative variants (full results with 7B backbones are provided in Fig. 40). **d**P**RM** is **highly sensitive** to label noise, demonstrating its potential vulnerability in multi-domain data. In contrast, **g**P**RM** is more robust, which is consistent with reports that LLM memorization can make random label noise act as a mild regularizer in math (Wu et al., 2025).

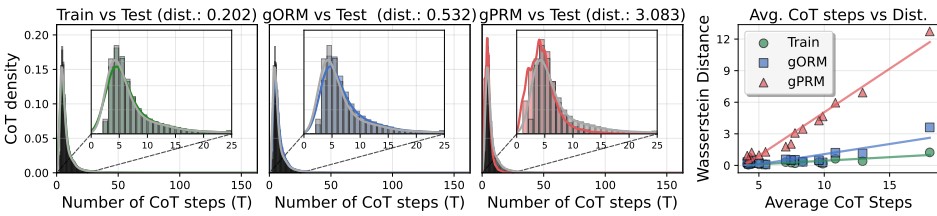

Figure 14: **Length distribution shift** on MMLU-Pro (overall/per-domain) measured by Wasserstein distance.

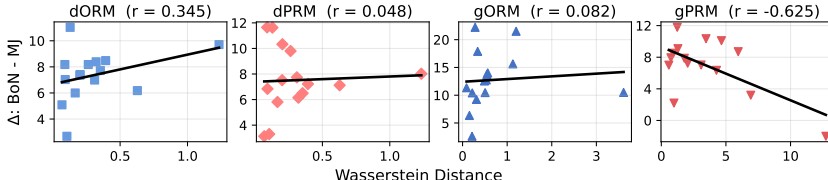

Figure 15: **Per-domain Bo$N$ improvement over majority voting** vs. **Wasserstein distance** on MMLU-Pro.

**Length shift hurts gPRM.** We furhter analyze why **gP**RM degrades in the multi-domain setting, despite its robustness to label noise in math. As CoTs become longer, the imperfect $p_{\texttt{LLM-j}}$ struggles to align stepwise verification rationales with process labels. As a result, *consensus filtering* **prunes long CoTs**, shifting the training CoT-length distribution away from the test set (Fig. 14).

We quantify the above length distribution shift with the Wasserstein distance (Kantorovich, 1960), reporting distances from the test set to the unfiltered pool (**Train**), the **gO**RM training set, and the **gP**RM training set. In the math domain (Tab. 4), **gP**RM has the smallest distance (*e.g.*, overall: 2.760/2.430/1.600 for Train/**gO**RM/**gP**RM), whereas in the multi-domain setting (Fig. 14 and Tab. 3) it has **the largest** distance (*e.g.*, overall: 0.202/0.532/3.083 for Train/**gO**RM/**gP**RM).

The distribution shift of **gP**RM also corresponds to its degradation across domains, observed in the multi-domain setting (Fig. 6). Fig. 15 shows a strong negative correlation between the Wasserstein distance and per-domain improvement over majority voting ($N$=16) for **gP**RM ($-0.625$), whereas correlations are weak for the other methods ($0.345/0.048/0.082$ for **dO**RM/**dP**RM/**gO**RM). Together, these results suggest that *consensus filtering* induces a length-distribution shift that disproportionately affects **gP**RM in the multi-domain setting, despite its robustness to label noise.

## 5 PRACTICAL GUIDELINES, LIMITATIONS, AND FUTURE WORK

This section suggests practical guidance, clarifies limitations, and outlines future directions:

| | | |
|---|---|---|
| (i) | *Short* CoTs, *clean* labels, *tight* latency | **dP**RM |
| (ii) | *Long* CoTs / *frequent error recoveries* | **gO**RM if compute permits; else **dO**RM |
| (iii) | *Mixed/shifting domains* | **gO**RM |
| (iv) | *High label noise* | ORM |
| | | PRMs amplify early errors |
| (v) | *Strict compute/latency* | **dO**RM/**dP**RM |
| | | **gO**RM and **gP**RM add sampling overhead |
| (vi) | *Limited training data* | **gO**RM/**gP**RM |
| | | Higher sample efficiency |

**Limitations and future work.** While we present a thorough analysis of four reward model variants, our study has several limitations: (i) we evaluate only tasks with **verifiable outcomes** (*e.g.*, math and non-math multiple–choice), which may not generalize to open-ended generation. (ii) All models are trained via **supervised fine–tuning**. One could instead use a generative verifier to roll out rationales and treat agreement between their verdict and the GT label as a reward signal for reinforcement learning (RL). Because using RL to train verifiers/reward models is uncommon and introduces additional confounders, we exclude RL-based training from our analysis. (iii) Owing to computational constraints, we adopt **LoRA adapters** rather than full-parameter fine-tuning. This choice may affect performance and scaling behavior, however, we expect the qualitative trends to hold. (iv) Following most of the PRM literature (Lightman et al., 2024; Zeng et al., 2025), we do not consider **tool use**, however, Gou et al. (2024) showed that tool use can help reduce auto-label noise. In future work, we plan to extend our analysis to open-ended generation, broader task domains, and a wider range of model families and training regimes, and to explicitly study tool-augmented verification and inference pipelines to assess their impact on label quality.

ETHICS STATEMENT

This work evaluates verification strategies for test-time scaling of LLMs across multiple domains. It **does not** involve human subjects, user studies, or the collection of personally identifiable information. All datasets used are **publicly available** benchmarks and were accessed under their respective licenses. To the best of our knowledge, they do not contain sensitive personal data.

A natural direction for future work is to increase the trustworthiness of LLM outputs in real systems by verifying them, thereby reducing reasoning errors and hallucinations. Although our experiments include legal and medical themed datasets (*e.g.*, law and health), the models and methods are research artifacts and are **not** intended for real-world legal, medical, or other high-stakes decision-making. They should not substitute professional judgment, and any deployment in such settings would require additional domain-specific validation, safety auditing, and regulatory compliance.

REPRODUCIBILITY STATEMENT

For reproducibility, we believe that we provide sufficient materials, including prompts, hyperparameters, model backbones, training details, and the synthetic data generation process, throughout the main paper (Sections 2 and 3.1). Additional details are deferred to Sections B to E due to space constraints. Upon acceptance, we commit to **publicly releasing** all relevant artifacts for reproducibility: **(i) code**, **(ii) datasets** (including any we generate), and **(iii) model checkpoints**.

In the meantime, anonymized artifacts are available at this repository as follows:

**Training datasets.**

- `train`: multi-domain training dataset for **d**ORM/**d**PRM, adapted from VersaPRM (Zeng et al., 2025).
- `train_orm`: multi-domain training dataset for **g**ORM.
- `train_prm`: multi-domain training dataset for **g**PRM.

**Test datasets.**

- `test`: multi-domain test dataset generated by Llama-3.1-8B-Instruct, adapted from VersaPRM (Zeng et al., 2025).
- `test_smollm`: multi-domain test dataset generated by SmolLM3-3B.
- `test_qwen`: multi-domain test dataset generated by Qwen2.5-7B-Instruct.
- `test_gemma`: multi-domain test dataset generated by gemma-2-9b-it.
- `test_llama70B`: multi-domain test dataset generated by Llama-3.1-70B-Instruct.

**Model checkpoints.**

- `dORM-14B`: **d**ORM with DeepSeek-R1-Distill-Qwen-14B backbone, trained on `train`.
- `dPRM-14B`: **d**PRM with DeepSeek-R1-Distill-Qwen-14B backbone, trained on `train`.
- `gORM-14B`: **g**ORM with DeepSeek-R1-Distill-Qwen-14B backbone, trained on `train_orm`.
- `gPRM-14B`: **g**PRM with DeepSeek-R1-Distill-Qwen-14B backbone, trained on `train_prm`.

**Code.**

- `versaprm/`: training/evaluation code for *discriminative* variants (**d**ORM/**d**PRM), adapted from VersaPRM (Zeng et al., 2025).
- `multigenprm/`: training/evaluation code for *generative* variants (**g**ORM/**g**PRM).

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

APPENDIX OVERVIEW

This appendix provides supplementary materials to support the main paper as follows:

- **Theoretical Analysis** (§A): details notations, assumptions, and proofs for Theorems. 4.1 to 4.3.
- **Prompts** (§B): presents the detailed prompt formats.
- **Datasets** (§C): describes the datasets used in our experiments.
- **Implementation Details** (§D): provides implementation details, such as (i) backbones for reward models, (ii) hyperparameters, and (iii) verification CoTs for **gO**RM and **gP**RM.
- **Training Examples** (§E): contains training examples including verification CoTs of **gO**RM and **gP**RM in the law domain of MMLU-pro.
- **Additional Results** (§F): includes the complete results of §3.2 (omitted in the main paper due to the space limit), such as results on MMLU-pro using weighted majority voting.
- **Additional Analysis** (§G): includes the complete results of §4.
- **Use of LLMs** (§H): outlines our use of LLMs in accordance with the ICLR 2026 submission policy.

## A THEORETICAL ANALYSIS

### A.1 ANALYSIS ON LOG-ERROR BOUND

**Notation.** We assume that a correct final step, $y = z_T = 1$, implies all previous steps are correct. Define the stepwise conditional probabilities $u_t := \Pr(z_t = 1 \mid x, z_1 = 1, \ldots, z_{t-1} = 1)$ for $t \in [T]$. By the chain rule and the assumption, the true reward function,

$$f(x) = p(y = 1 \mid x) = p(z_T = 1 \mid x) = p(z_{1:T} = 1 \mid x) = \prod_{t=1}^{T} u_t(x)$$

and we write $\zeta(x) := \log f(x) = \sum_{t=1}^{T} \log u_t(x)$. For **dP**RM, we define the stepwise conditional distribution $\hat{u}_t(x) := \hat{f}_{\mathbf{dP}\text{RM}}(x_{1:t})$ and use product for the aggregation, $i.e. \hat{f}_{\mathbf{dP}\text{RM}}(x) := \prod_{t=1}^{T} \hat{u}_t(x)$. Similarly, we define the conditional distribution $F_t(x, v_{\le t}) \in [0,1]$ to be the **gP**RM's normalized probability that step $t$ is correct given the verification prefix, $i.e. \hat{f}_{\mathbf{gP}\text{RM}}(x) := \mathbb{E}_{v_{1:L^+}}[\prod_{t=1}^{T} F_t(x, v_{\le t})]$. To bound log probability, we assume there is $\varsigma \in (0, 1/2]$ such that all probabilities/predictors appearing inside logarithms are clipped into $[\varsigma, 1 - \varsigma]$. Hence all logs are finite and $|\log(\cdot)| \le \log(1/\varsigma)$.

**Error terms.**

1. **dP**RM. Define $\delta_t := \log \hat{u}_t - \log u_t$ (evaluated at the appropriate prefixes), and

$$m_t := \mathbb{E}[\delta_t \mid x], \quad \xi_t := \delta_t - m_t,$$

so $\mathbb{E}[\xi_t \mid x] = 0$.

2. **dO**RM or **gO**RM. Let $\epsilon \in \{\epsilon_d, \epsilon_g\}$,

$$\epsilon_d := \log \hat{f}_{\mathbf{dO}\text{RM}}(x) - \log f(x), \quad \epsilon_g := \log \hat{f}_{\mathbf{gO}\text{RM}}(x) - \log f(x),$$

and decompose

$$\bar{m} := \mathbb{E}[\epsilon \mid x], \quad \bar{\xi} := \epsilon - \bar{m}, \quad \beta_{\text{orm}}^2 := \mathbb{E}[\bar{m}^2],$$

so that $\mathbb{E}[\bar{\xi} \mid x] = 0$.

3. **gP**RM. For a single rollout $v_{1:L^+} \sim p_{\mathbf{gP}\text{RM}}(\cdot \mid x)$, define

$$\tilde{u}_t := F_t(x, v_{\le t}), \quad \tilde{f}_{\mathbf{gP}\text{RM}}(x) := \prod_{t=1}^{T} \tilde{u}_t.$$

The sampled **gP**RM log-error is

$$\Delta_{\mathbf{gP}\text{RM}} := \log \tilde{f}_{\mathbf{gP}\text{RM}}(x) - \zeta(x) = \sum_{t=1}^{T} \delta_t^{(g)}, \quad \delta_t^{(g)} := \log \tilde{u}_t - \log u_t.$$

Let
$$m_t^{(g)} := \mathbb{E}[\delta_t^{(g)} \mid x], \quad \xi_t^{(g)} := \delta_t^{(g)} - m_t^{(g)},$$

so that $\mathbb{E}[\xi_t^{(g)} \mid x] = 0$.

**Assumptions.** There exist constants $\sigma^2 > 0$ and $\gamma \geq 0$ (independent of $T$) such that for all $x$,

1. (Variance floors) $\mathrm{Var}(\xi_t \mid x) \geq \sigma^2$, $\mathrm{Var}(\xi_t^{(g)} \mid x) \geq \sigma^2 + \tau^2$

2. (Weak anti-correlation) $\displaystyle\sum_{1 \leq s < t \leq T} \mathrm{Cov}(\xi_s, \xi_t \mid x) \geq -\gamma T$, $\displaystyle\sum_{1 \leq s < t \leq T} \mathrm{Cov}(\xi_s^{(g)}, \xi_t^{(g)} \mid x) \geq -\gamma T$

3. (Positive slope) $\sigma^2 > 2\gamma$.

For **g**PRM with *sampled* verification CoTs, sampling contributes per-step noise: $\mathrm{Var}(\xi_t^{(g)} \mid x) \geq \sigma^2 + \tau^2$ for some $\tau^2 > 0$. For ORMs, assume $\mathrm{Var}(\bar{\xi} \mid x) \leq \tau_{\mathrm{orm}}^2 < \infty$ (no $T$-dependence).

**Theorem A.1** (Log-error lower bound of **dP**RM). *Let $\Delta_{\boldsymbol{dP}RM} := \log \hat{f}_{\boldsymbol{dP}RM}(x) - \zeta(x)$. Under the assumptions above,*
$$\mathbb{E}\left[\Delta_{\boldsymbol{dP}RM}^2\right] \geq (\sigma^2 - 2\gamma) T.$$

**Theorem A.2** (Log-error bound of **dO**RM or **gO**RM). *Let $\epsilon \in \{\epsilon_d, \epsilon_g\}$ and write $\epsilon = \bar{m} + \bar{\xi}$ with $\mathbb{E}[\bar{\xi} \mid x] = 0$. If $\mathrm{Var}(\bar{\xi} \mid x) \leq \tau_{\mathrm{orm}}^2$ (independent of $T$), then*
$$\mathbb{E}[\epsilon^2] = \mathbb{E}[\mathrm{Var}(\bar{\xi} \mid x)] + \mathbb{E}[\bar{m}^2] \leq \tau_{\mathrm{orm}}^2 + \beta_{\mathrm{orm}}^2,$$

*a bound that does not depend on the CoT length $T$.*

**Theorem A.3** (Log-error lower bound of **gP**RM). *Under the assumptions above,*
$$\mathbb{E}\left[(\Delta_{\boldsymbol{gP}RM})^2\right] \geq (\sigma^2 + \tau^2 - 2\gamma) T.$$

**Jensen-gap representation (mean predictor).** Let $L(x,v) := \sum_{t=1}^{T} \log F_t(x, v_{\leq t})$ and $K_x(\theta) := \log \mathbb{E}[e^{\theta L} \mid x]$. Define the mean predictor $\mu(x) := \mathbb{E}[e^L \mid x]$ and $\Delta_{\mathrm{mean}}(x) := \log \mu(x) - \zeta(x)$. Then with $B^{(g)}(x) := \mathbb{E}[L \mid x] - \zeta(x)$, we have the exact decomposition

$$\Delta_{\mathrm{mean}}(x) = B^{(g)}(x) + \delta_J(x), \quad \delta_J(x) = K_x(1) - K_x'(0) = \int_0^1 (1-\theta) \mathrm{Var}_\theta(L \mid x)\, d\theta \geq 0,$$

where $\mathrm{Var}_\theta$ denotes variance under the exponentially tilted law $d\mathbb{P}_\theta \propto e^{\theta L} d\mathbb{P}$, *i.e.*, $d\mathbb{P}_\theta(v) = \mathbb{1}\{M(\theta) > 0\} e^{\theta L(x,v)} M(\theta)^{-1} d\mathbb{P}(v)$ with $M(\theta) := \mathbb{E}[e^{\theta L} \mid x]$.

**Theorem A.4** (Log-error lower bound of mean-**gP**RM). *Assume the conditions of Theorem A.3. In addition, suppose there exists $\kappa \in (0,1]$ such that for all $\theta \in [0,1]$,*
$$\mathrm{Var}_\theta(L \mid x) \geq \kappa \mathrm{Var}(L \mid x).$$

*Then, for every $x$,*

$$\Delta_{\mathrm{mean}}(x) \geq B^{(g)}(x) + \frac{\kappa}{2} \mathrm{Var}(L \mid x) \geq B^{(g)}(x) + \frac{\kappa}{2}\left((\sigma^2 + \tau^2 - 2\gamma)T\right).$$

*Consequently,*

$$\mathbb{E}[\Delta_{\mathrm{mean}}] \geq \frac{\kappa}{2}\left((\sigma^2 + \tau^2 - 2\gamma)T\right) - \sqrt{\mathbb{E}[B^{(g)}(x)^2]}, \quad \mathbb{E}[\Delta_{\mathrm{mean}}^2] \geq (\max\{0, \mathbb{E}[\Delta_{mean}]\})^2.$$

**Takeaways.** Under mild anti-correlation and variance-floor assumptions, **dP**RM and sampled **gP**RM incur log-error that grows at least linearly in the CoT length $T$, and the additional sampling noise $\tau^2$ makes **gP**RM strictly worse. In contrast, ORM estimators admit error bounds that are independent of $T$ provided the conditional noise is bounded, which makes them preferable for long CoTs. For mean-**gP**RM, the Jensen gap introduces a strictly nonnegative bias that scales with the variance of $L$ and hence with $T$, so even a calibrated predictor ($B^{(g)} = 0$) exhibits error that increases with chain length. All proofs are deferred to §A.2.

## A.2 PROOFS

**Proof of Theorem. A.1.**

*Proof.* Let

$$B := \sum_{t=1}^{T} m_t, \qquad N := \sum_{t=1}^{T} \xi_t,$$

so $\Delta_{\mathbf{dP}\mathrm{RM}} = B + N$ with $\mathbb{E}[N \mid x] = 0$. By the tower property,

$$\mathbb{E}\left[\Delta_{\mathbf{dP}\mathrm{RM}}^2\right] = \mathbb{E}\left[\mathbb{E}[(B+N)^2 \mid x]\right] = \mathbb{E}\left[B^2 + 2B\mathbb{E}[N \mid x] + \mathbb{E}[N^2 \mid x]\right]$$
$$= \mathbb{E}\left[\mathrm{Var}(N \mid x)\right] + \mathbb{E}[B^2] \geq \mathbb{E}\left[\mathrm{Var}(N \mid x)\right].$$

Expanding,

$$\mathrm{Var}(N \mid x) = \sum_{t=1}^{T} \mathrm{Var}(\xi_t \mid x) + 2 \sum_{1 \leq s < t \leq T} \mathrm{Cov}(\xi_s, \xi_t \mid x).$$

Apply the variance floors and weak anti-correlation to get $\mathrm{Var}(N \mid x) \geq +T\sigma^2 - 2\gamma T$. Taking expectations preserves the bound. $\quad\square$

**Proof of Theorem. A.2**

*Proof.* By the conditional bias–variance decomposition (law of total variance),

$$\mathbb{E}[\epsilon^2] = \mathbb{E}\left[\mathrm{Var}(\epsilon \mid x)\right] + \mathbb{E}\left[(\mathbb{E}[\epsilon \mid x])^2\right] = \mathbb{E}\left[\mathrm{Var}(\bar{\xi} \mid x)\right] + \mathbb{E}\left[\bar{m}^2\right].$$

The assumption $\mathrm{Var}(\bar{\xi} \mid x) \leq \tau_{\mathrm{orm}}^2$ for all $x$ gives $\mathbb{E}\left[\mathrm{Var}(\bar{\xi} \mid x)\right] \leq \tau_{\mathrm{orm}}^2$, and by definition $\beta_{\mathrm{orm}}^2 = \mathbb{E}[\bar{m}^2]$. $\quad\square$

**Proof of Theorem. A.3**

*Proof.* Decompose

$$\Delta_{\mathbf{gP}\mathrm{RM}} = \sum_{t=1}^{T} \delta_t^{(g)} = \underbrace{\sum_{t=1}^{T} m_t^{(g)}}_{=: B^{(g)}} + \underbrace{\sum_{t=1}^{T} \xi_t^{(g)}}_{=: N^{(g)}}.$$

Conditional mean-zero $\mathbb{E}[N^{(g)} \mid x] = 0$ implies

$$\mathbb{E}\left[(\Delta_{\mathbf{gP}\mathrm{RM}})^2\right] = \mathbb{E}\left[\mathrm{Var}\left(N^{(g)} \mid x\right)\right] + \mathbb{E}\left[(B^{(g)})^2\right] \geq \mathbb{E}\left[\mathrm{Var}\left(N^{(g)} \mid x\right)\right].$$

Now expand $\mathrm{Var}(N^{(g)} \mid x)$:

$$\mathrm{Var}(N^{(g)} \mid x) = \sum_{t=1}^{T} \mathrm{Var}(\xi_t^{(g)} \mid x) + 2 \sum_{1 \leq s < t \leq T} \mathrm{Cov}(\xi_s^{(g)}, \xi_t^{(g)} \mid x) \geq T(\sigma^2 + \tau^2) - 2\gamma T.$$

Taking expectations in $x$ gives the stated bound. $\quad\square$

**Proof of Theorem. A.4**

*Proof.* **1) Exponential tilting and log-mgf.** Define $M(\theta) := \mathbb{E}[e^{\theta L} \mid X]$ and $K_X(\theta) := \log M(\theta)$. Since $e^{\theta L} \in (0,1]$ for $\theta \in [0,1]$ and $\mathbb{E}[|L|^2] < \infty$, dominated convergence yields $M'(\theta) = \mathbb{E}[Le^{\theta L} \mid X]$ and $M''(\theta) = \mathbb{E}[L^2 e^{\theta L} \mid X]$. Let $d\mathbb{P}_\theta(C) := e^{\theta L(X,C)} M(\theta)^{-1} d\mathbb{P}(C)$ and $\mathbb{E}_\theta[\cdot] := \mathbb{E}[\cdot\, e^{\theta L}]/M(\theta)$. Then

$$K_X'(\theta) = \frac{M'(\theta)}{M(\theta)} = \mathbb{E}_\theta[L \mid X], \qquad K_X''(\theta) = \frac{M''(\theta)M(\theta) - (M'(\theta))^2}{M(\theta)^2} = \mathrm{Var}_\theta(L \mid X).$$

**2) Jensen-gap identity.** Taylor with integral remainder at $\theta = 0$ gives

$$K_X(1) = K_X(0) + K'_X(0) + \int_0^1 (1 - \theta)K''_X(\theta)\, d\theta.$$

Since $K_X(0) = 0$ and $K'_X(0) = \mathbb{E}[L \mid X]$, we obtain

$$\log \mu(X) = \mathbb{E}[L \mid X] + \int_0^1 (1 - \theta)\operatorname{Var}_\theta(L \mid X)\, d\theta.$$

By definition of the mean predictor,

$$\Delta_{\mathrm{mean}}(X) = \log \mu(X) - \zeta_A(X), \text{ where } \mu(X) = \mathbb{E}[e^L \mid X].$$

Plugging $\log \mu(X) = \Delta_{\mathrm{mean}}(X) + \zeta_A(X)$ with $B^{(g)}(X) := \mathbb{E}[L \mid X] - \zeta_A(X)$, this yields

$$\Delta_{\mathrm{mean}}(X) = B^{(g)}(X) + \delta_J(X), \quad \delta_J(X) := \int_0^1 (1 - \theta)\operatorname{Var}_\theta(L \mid X)\, d\theta \ \geq 0.$$

**3) Lower bound on $\delta_J$ and variance linkage.** By tilt-stability,

$$\delta_J(X) \geq \frac{\kappa}{2}\operatorname{Var}(L \mid X).$$

Moreover, since $L = \zeta_A + \Delta_{\mathrm{g\text{-}prm}} = \zeta_A + B^{(g)} + N^{(g)}$ with $\mathbb{E}[N^{(g)} \mid X] = 0$, and since $\zeta_A$ and $B^{(g)}(X)$ are constants when conditioning on $X$, we have

$$\operatorname{Var}(L \mid X) = \operatorname{Var}(N^{(g)} \mid X).$$

Expanding and using the variance floors and weak anti-correlation conditions (as in Theorem A.3),

$$\operatorname{Var}(N^{(g)} \mid X) \ \geq \ \sigma_A^2 + T(\sigma^2 + \tau^2) - 2\gamma T - 2\gamma_A.$$

Combining this gives the pointwise bound

$$\Delta_{\mathrm{mean}}(X) \ \geq \ B^{(g)}(X) + \frac{\kappa}{2}\Big(\sigma_A^2 + T(\sigma^2 + \tau^2) - 2\gamma T - 2\gamma_A\Big).$$

**4) Expectations and MSE.** Taking expectations over $X$ and applying Cauchy–Schwarz to $\mathbb{E}[B^{(g)}(X)]$ yields

$$\mathbb{E}[\Delta_{\mathrm{mean}}] \ \geq \ \frac{\kappa}{2}\Big((\sigma^2 + \tau^2 - 2\gamma)T + (\sigma_A^2 - 2\gamma_A)\Big) - \sqrt{\mathbb{E}[B^{(g)}(X)^2]}.$$

Finally, Jensen's inequality gives $(\max\{0, \mathbb{E}[\Delta_{\mathrm{mean}}]\})^2 \leq \mathbb{E}[\Delta_{\mathrm{mean}}^2]$, so the MSE bound follows. In the calibrated case $B^{(g)} \equiv 0$, the stated simplified bounds hold. $\qquad\square$

# B  PROMPTS

In this section, we present prompt formats used in this work:

- Fig. 16: **User prompt format for generating CoTs** on GSM8K (Cobbe et al., 2021) and MATH (Hendrycks et al., 2021).
- Fig. 17: **User prompt format for generating CoTs** on MMLU-Pro (Wang et al., 2024c) proposed by Zeng et al. (2025).
- Fig. 18: **System prompt format for auto-labeling process labels** on MMLU-Pro (Wang et al., 2024c) proposed by Zeng et al. (2025).
- Fig. 19: **User prompt format for auto-labeling process labels** on MMLU-Pro (Wang et al., 2024c) proposed by Zeng et al. (2025).
- Fig. 20: **Prompt format of gORM** (Zhang et al., 2025a). We use this format for both generating synthetic verification-CoTs and training/evaluation of gORM.

- Fig. 21: **Prompt format for generating verification-CoTs** for gPRM following Khalifa et al. (2025).

- Fig. 22: **Prompt format of gPRM** for training and evaluation.

---

**[user]** Solve the following math problem efficiently and clearly:

- For simple problems (2 steps or fewer):
Provide a concise solution with minimal explanation.

- For complex problems (3 steps or more):
Use this step-by-step format:
## Step 1: [Concise description]
[Brief explanation and calculations]
## Step 2: [Concise description]
[Brief explanation and calculations]
`[OMITTED...]`

Regardless of the approach, always conclude with:
`Therefore, the final answer is: $\\boxed{answer}$.`

I hope it is correct. Where `[answer]` is just the final number or expression that solves the problem.

[Problem]
*{problem}*
**[/user] [assistant]**

---

Figure 16: **User prompt format for generating CoTs** on GSM8K (Cobbe et al., 2021) and MATH (Hendrycks et al., 2021).

---

**[user]** Given the following question and candidate answers, choose the best answer.
[Question]
*{question #1}*
**[/user] [assistant]**
*{assistant response #1}*
**[/assistant]**

**[user]** Given the following question and candidate answers, choose the best answer.
[Question]
*{question #2}*
**[/user] [assistant]**
*{assistant response #2}*
**[/assistant]**
`[OMITTED...]`

**[user]** Given the following question and candidate answers, choose the best answer.
[Question]
*{question}*
**[/user] [assistant]**

---

Figure 17: **User prompt format for generating CoTs** on MMLU-Pro (Wang et al., 2024c) proposed by Zeng et al. (2025)

**[system]** You are an experienced evaluator specializing in assessing the quality of reasoning steps in problem-solving. Your task is to find the first BAD step in a student's solution to a multiple choice question.

You will judge steps as GOOD, OK, or BAD based on the following criteria:

**1. GOOD Step** A step is classified as GOOD if it meets all of these criteria:

- **Correct**: Everything stated is accurate and aligns with known principles or the given problem.
- **Verifiable**: The step can be verified using common knowledge, simple calculations, or a quick reference (e.g., recalling a basic theorem). If verifying requires extensive effort (e.g., detailed calculations or obscure references), mark it BAD instead.
- **Appropriate**: The step fits logically within the context of the preceding steps. If a prior mistake exists, a GOOD step can correct it.
- **Insightful**: The step demonstrates reasonable problem-solving direction. Even if ultimately progressing in the wrong direction, it is acceptable as long as it represents a logical approach.

**2. OK Step** A step is classified as OK if it is:

- **Correct and Verifiable**: Contains no errors and can be verified.
- **Unnecessary or Redundant**: Adds little value, such as restating prior information or providing basic encouragement (e.g., "Good job!").
- **Partially Progressing**: Makes some progress toward the solution but lacks decisive or significant advancement.

**3. BAD Step** A step is classified as BAD if it:

- **Is Incorrect**: Contains factual errors, misapplies concepts, derives an incorrect result, or contradicts the ground truth answer.
- **Is Hard to Verify**: Requires significant effort to confirm due to poor explanation.
- **Is Off-Topic**: Includes irrelevant or nonsensical information.
- **Derails**: Leads to dead ends, circular reasoning, or unreasonable approaches.

**Task Description**
You will be provided with:

1. A Multiple Choice Question
2. A Ground Truth Answer
3. A Student's Step-by-Step Solution, where each step is enclosed with tags and indexed from 0.

Once you identify a BAD step, return the index of the earliest BAD step. Otherwise, return the index of -1 (which denotes all steps are GOOD or OK). Please put your final answer (i.e., the index) in \boxed{}.
**[/system]**

Figure 18: **System prompt format for auto-labeling process labels** on MMLU-Pro (Wang et al., 2024c) proposed by Zeng et al. (2025)

**[user]** The following is a multiple choice question and its ground truth answer. You are also given a student's solution (split into steps, enclosed with tags and indexed from 0):

[Multiple Choice Question]
{*question*}

[Ground Truth Answer]
{*answer*}

[Student Solution]
{*solution*}

**[/user] [assistant]** The first BAD step index is:

Figure 19: **User prompt format for auto-labeling process labels** on MMLU-Pro (Wang et al., 2024c) proposed by Zeng et al. (2025)

> **[user]** You are a {category} teacher. Grade the solution, verifying correctness step by step.
> At the end of Solution verification, when you give your final grade, write it in the form "Verification: Is the answer correct (Yes/No)? X", where X is either Yes or No.
>
> [{Category} Problem]
> {problem}
>
> [Solution]
> {solution}
> **[/user] [assistant] [think]** Let's verify step by step:

Figure 20: **Prompt format of gORM** (Zhang et al., 2025a). We use this format for both generating synthetic verification-CoTs and training/evaluation of gORM.

> **[user]** You are given a {*category*} problem and a proposed multiple-step solution (with a step on each line):
>
> [{*Category*} Problem]
> {*question*}
>
> [Solution]
> {*solution*}
>
> Review and critique the proposed solution steps and determine whether each step is correct. If the solution is incomplete, only critique the steps that are provided. Your output must be in the following format:
>
> Step 1: The step is \boxed{correct/incorrect}
> Step 2: The step is \boxed{correct/incorrect}
> ⋮
> Step $n$: The step is \boxed{correct/incorrect}
>
> Once you find an incorrect step, you should stop since you do not need to analyze the remaining steps. If the solution is incomplete, only verify the provided steps. **[/user] [assistant] [think]** Let's verify step by step:

Figure 21: **Prompt format for generating verification-CoTs** for gPRM following Khalifa et al. (2025).

> **[user]** You are given a {category} problem and a proposed step-by-step solution:
>
> [{category} Problem]
> {problem}
>
> [Solution]
> {solution}
>
> Review and critique each step in the proposed solution to determine whether each step is correct. If the solution is incomplete, only verify the provided steps. **[/user] [assistant] [think]** Let's verify step by step:

Figure 22: **Prompt format of gPRM** for training and evaluation.

## C  DATASET

In this section, we provide more details on the datasets used in this paper.

**Math Datasets.** For the math domain, we use the widely adopted **PRM800K** (Lightman et al., 2024) for training, where the process labels $z_{1:T}$ are human-annotated. For training ORMs, we set the outcome label $y = \mathbb{1}(z_{1:T} = \mathbf{1}_T)$ (rather than $y = \mathbb{1}(\hat{a}(r_T) = a)$), since PRM800K provides high-quality ground-truth process labels. As a testbed, we use **ProcessBench** (Zheng et al., 2024), which comprises four splits: 400 CoTs from GSM8K (Cobbe et al., 2021), 1K from

Math (Hendrycks et al., 2021), 1K from Omni-Math (Gao et al., 2025), and 1K from Olympiad-Bench (He et al., 2024). We evaluate outcome verification by predicting $y \in \{0,1\}$ using the `final_answer_correct` field. We also generate $N = 16$ CoTs per question with Qwen2.5-7B-Instruct (Team, 2024a) to assess test-time scaling (TTS).

Table 1: Dataset statistics for each domain of MMLU-pro (Wang et al., 2024c). We report the number of questions, the number of CoTs, and the average number of CoTs per question for both training and test splits.

| Domain | Training Set | | | Test Set | | |
|---|---|---|---|---|---|---|
| | # Questions | # CoTs | Avg. CoTs / Q | # Questions | # CoTs | Avg. CoTs / Q |
| Law | 500 | 7,806 | 15.61 | 145 | 18,537 | 127.84 |
| Psychology | 498 | 7,901 | 15.87 | 150 | 19,164 | 127.76 |
| Chemistry | 500 | 6,537 | 13.07 | 150 | 15,981 | 106.54 |
| Biology | 417 | 6,420 | 15.40 | 130 | 16,441 | 126.47 |
| Physics | 500 | 6,680 | 13.36 | 150 | 16,460 | 109.73 |
| History | 81 | 1,275 | 15.74 | 150 | 19,159 | 127.73 |
| Economics | 500 | 7,749 | 15.50 | 150 | 18,911 | 126.07 |
| Math | 500 | 6,940 | 13.88 | 150 | 17,014 | 113.43 |
| Business | 489 | 6,969 | 14.25 | 149 | 17,344 | 116.40 |
| Philosophy | 199 | 3,125 | 15.70 | 149 | 18,844 | 126.47 |
| Health | 456 | 7,202 | 15.79 | 140 | 17,862 | 127.59 |
| Engineering | 500 | 6,032 | 12.06 | 150 | 15,708 | 104.72 |
| Computer Science | 110 | 1,638 | 14.89 | 150 | 18,429 | 122.86 |
| Other | 500 | 7,824 | 15.65 | 150 | 18,982 | 126.55 |
| **Total** | 5,750 | 84,098 | 14.63 | 2,063 | 248,836 | 120.62 |

**Multi-domain datasets.** For the multi-domain setting, we adopt **MMLU-Pro** (Wang et al., 2024c), a 10-choice benchmark spanning *14 domains*: law, psychology, chemistry, biology, physics, history, economics, math, business, philosophy, health, engineering, computer science, and other. As shown in Tab. 1, the corpus includes 5,750 training and 2,063 evaluation questions. For each question, Zeng et al. (2025) generate 16/128 CoTs for training/evaluation with Llama-3.1-8B-Instruct (Dubey et al., 2024), and auto-label reasoning steps (*i.e.*, process labels) using Llama-3.1-70B-Instruct with prompts in Figs. 18 and 19; please see Zeng et al. (2025) for more details. To assess generalization across CoTs from different $p_{\text{LLM}}$, we also generate 16 CoTs per evaluation question using SmolLM3-3B (Bakouch et al., 2025), Qwen2.5-7B-Instruct, gemma-2-9b-it (Team et al., 2024), and Llama-3.1-70B-Instruct, spanning diverse model sizes and families.

# D   IMPLEMENTATION DETAILS

In this section, we provide implementation details omitted from the main paper due to space limits.

**Backbones for reward models.** Following Zhang et al. (2025a) and Khalifa et al. (2025), we use R1-Distill-Qwen-1.5B and R1-Distill-Qwen-7B (Guo et al., 2025) for the math domain, and R1-Distill-Llama-8B and R1-Distill-Qwen-14B for the multi-domain setting, as reward-model backbones. Note that VersaPRM (Zeng et al., 2025) originally used Llama-3.1-8B-Instruct as the reward-model backbone for **dP**RM; for a fair comparison, we use R1-Distill models for both **dO**RM and **dP**RM.

Table 2: **Summary of hyperparameters**.

| Method | LoRA | | | Training | | | | | | Inference | | |
|---|---|---|---|---|---|---|---|---|---|---|---|---|
| | Rank $r$ | $\alpha$ | Dropout $p$ | Batch | Optim. | Epochs | LR | Decay | Scheduler | Package | Temp. $\tau$ | $M$ |
| **dO**RM & **dP**RM | 16 | 32 | 0.1 | 16 | AdamW | 1 | 1e-4 | 1e-2 | Cosine | - | - | - |
| **gO**RM & **gP**RM | 32 | 16 | 0.1 | 16 | AdamW | 1 | 1e-4 | 1e-2 | Linear | vLLM | 0.6 | 10 or 16 |

**Hyperparameters.** We apply LoRA (Hu et al., 2022) for parameter-efficient fine-tuning, optimize with AdamW (Loshchilov & Hutter, 2019), and use vLLM (Kwon et al., 2023) for fast inference. At inference, we sample $M{=}16$ verification CoTs for the math domain and $M{=}10$ for the multi-domain setting. Hyperparameters are summarized in Tab. 2: for **dO**RM/**dP**RM we adopt those of Zeng et al. (2025), and for **gO**RM/**gP**RM we follow Khalifa et al. (2025). Note that in preliminary experiments we set $r{=}32$ and $\alpha{=}16$ for **dO**RM/**dP**RM to compare fairly with **gO**RM/**gP**RM (also using $r{=}32$ and $\alpha{=}16$). However, we observed an **overall performance degradation** (*e.g.*, $\approx$2%), so we follow the

settings of Zeng et al. (2025). The hyperparameters in Tab. 2 are shared across all experiments and we do not perform exhaustive tuning. We report means over three independent runs for the math domain and a single run for the multi-domain setting due to resource constraints.

**Verification CoTs for `gORM` and `gPRM`.** Following Khalifa et al. (2025), we sample 4 different verification CoTs for each question $q$ and CoT $r_{1:T}$ pair in the training dataset by prompting QwQ-32B (Qwen Team, 2025) with `temperature`=0.6, `top_k`=20, `top_p`=0.95, and `min_p`=0 using the formats in Figs. 20 and 21. Note that Khalifa et al. (2025) originally used QwQ-32B-Preview (Team, 2024b). In preliminary, we found QwQ-32B more likely to follow instructions and produce more parsable verification CoTs (*e.g.*, 1K vs. 7K for **gP**RM in the law domain), so we use it throughout.

For the math domain we set `category` as `math`; for the multi-domain setting we use `category` $\in\{$`law, ..., computer science`$\}$ and leave it blank for `other`. We discard any verification CoT that (i) has **unparsable labels**; (ii) contains **Chinese characters**; (iii) exceeds **the token limit**—4,096 for math (Khalifa et al., 2025) or 8,192 for multi-domain; or (iv) whose parsed labels are **inconsistent with the targets** (*e.g.*, $y$ or $z_{1:T}$), corresponding to the *consensus filtering* in §2.2. We also balance the counts of `Yes`/`No` examples. The resulting training sets contain 34,286 CoTs for **gO**RM and 35,666 CoTs for **gP**RM in the math domain, and 171,780 (**gO**RM) and 94,156 (**gP**RM) in the multi-domain setting.

# E    TRAINING EXAMPLES

In this section, we present training examples including verification CoTs:
- Fig. 23: **Training example of `gORM`** on the law domain of MMLU-pro (Wang et al., 2024c).
- Fig. 24: **Training example of `gPRM`** on the law domain of MMLU-pro (Wang et al., 2024c).

**[user]**
You are a {category} teacher. Grade the solution, verifying correctness step by step.
At the end of Solution verification, when you give your final grade, write it in the form "Verification: Is the answer correct (Yes/No)? X", where X is either Yes or No.

[Law Problem]
Question: A shopper in a store sees that a display case is about to fall on a second shopper's head. The shopper does nothing to warn or assist the second shopper, who is a stranger to her. She could have taken action to protect the second shopper without harming herself. Did the shopper have a duty to take action to try and protect the second shopper from injury?
A. No, the shopper is not required to intervene unless she is a store employee or security personnel.
B. Yes, the shopper must take action under the G̈ood Samaritanl̈aw.
C. Yes, the fact that a person sees that action is needed for another's aid or protection does impose a legal duty to act. [OMITTED...]

[Solution]

The shopper could have taken action to protect the second shopper without harming herself, so the issue is whether there is a legal duty to intervene.

The "Good Samaritan" law protects individuals from liability for failing to provide assistance, so B is incorrect.

There is no special relationship between the shopper and the second shopper, so I is incorrect.

[OMITTED...]

Therefore, the shopper is not required to intervene, and the correct answer is D.

The answer is (D).
**[/user] [assistant] [think]**

Let's verify step by step:
The question is about whether a shopper has a duty to act to prevent harm to another stranger when she can do so without risk. The solution concludes D is correct.

First, the solution correctly dismisses B because Good Samaritan laws typically protect those who do help, not impose a duty to act. That seems right.

For option I, the solution says no special relationship exists. Since the shopper and victim are strangers, that's accurate. So I is wrong.

[OMITTED...]

The solution mentions "duty to rescue" in certain circumstances like special relationships or being first to notice. However, in the problem, there's no indication the shopper was first. So D holds.

Wait, but the problem states "she could have taken action without harming herself," which is a key point. Some jurisdictions might have a duty if it's safe, but traditionally, common law doesn't. The solution's conclusion aligns with that.

Therefore, the answer D is correct. The solution's reasoning is sound, dismissing other options properly. The final answer D matches the analysis. So the solution is correct.

**[/think]**

Verification: Is the answer correct (Yes/No)? Yes **[assistant]**

Figure 23: **Training example of gORM** on the law domain of MMLU-pro (Wang et al., 2024c).

> **[user]** You are given a law problem and a proposed step-by-step solution:
>
> [Law Problem]
> Question: A plaintiff sued a defendant for injuries that the plaintiff suffered when he was struck by the defendant's car. At trial, the plaintiff testified that he was walking across the street inside a crosswalk at the time the defendant's car hit him. `[OMITTED...]` In support of the defendant's testimony, the defendant calls a neighbor to testify that she has observed the plaintiff frequently during the years that she has lived on that street, and that the plaintiff "always crosses the street outside the crosswalk." The neighbor's testimony is likely to be
> A. admitted, because it tends to show that the plaintiff was careless.
> `[OMITTED...]`
> G. admitted, because it is probative of the plaintiff's habit of crossing the street outside the crosswalk.
> H. excluded, because it is irrelevant to the specific incident.
> I. admitted, because it provides context to the plaintiff's behavior.
>
> [Solution]
> Step 1: The neighbor's testimony is about the plaintiff's habit of crossing the street outside the crosswalk, which is relevant to the case.
> Step 2: Habit evidence is admissible to prove that a person regularly acts in conformity with the habit.
> Step 3: The neighbor's testimony is an example of habit evidence, as it describes the plaintiff's regular behavior of crossing the street outside the crosswalk.
> Step 4: The answer is (G).
>
> Review and critique each step in the proposed solution to determine whether each step is correct. If the solution is incomplete, only verify the provided steps. **[/user] [assistant] [think]**
> Let's verify step by step:
>
> First, the problem is about whether the neighbor's testimony about the plaintiff's habit is admissible. The proposed solution says step 1 is correct because the testimony is about habit and relevant. That seems right since the case hinges on where the plaintiff was crossing.
>
> Step 2 states habit evidence is admissible to show conformity. Under Federal Rules of Evidence, habit (as opposed to character) is admissible. So step 2 is correct.
>
> Step 3 says the neighbor's testimony is habit evidence. But wait, habit requires a regular, routine practice. The neighbor says "frequently" over years, which might qualify as a habit. So step 3 is correct.
>
> Step 4 concludes G is correct. So all steps seem okay. Wait, but maybe "frequently" isn't enough? Some jurisdictions require more specific regularity. But generally, habit evidence can be admissible if it's a settled pattern. So steps are correct. Hmm, but let me check again. The options: G says "probative of plaintiff's habit". So yes, the steps are correct. The answer should be G. So all steps are correct.
> **[/think]**
>
> Step 1: The step is \boxed{correct}
> Step 2: The step is \boxed{correct}
> Step 3: The step is \boxed{correct}
> Step 4: The step is \boxed{correct}
> Is the solution correct? Yes
> **[assistant]**

Figure 24: **Training example of `gPRM`** on the law domain of MMLU-pro (Wang et al., 2024c).

# F  ADDITIONAL RESULTS

In this section, we present additional results as follows:

- Fig. 25: **Weighted majority voting results using Llama-3.1-8B-Instruct** on MMLU-Pro with R1-Distill-Llama-8B backbone for reward models.

- Fig. 26: **Best-of-$N$ results using Llama-3.1-8B-Instruct when trained and evaluated on each domain** of MMLU-Pro with R1-Distill-Llama-8B backbone for reward models.

- Fig. 27: **Weighted majority voting results using Llama-3.1-8B-Instruct when trained and evaluated on each domain** of MMLU-Pro with R1-Distill-Llama-8B backbone for reward models.

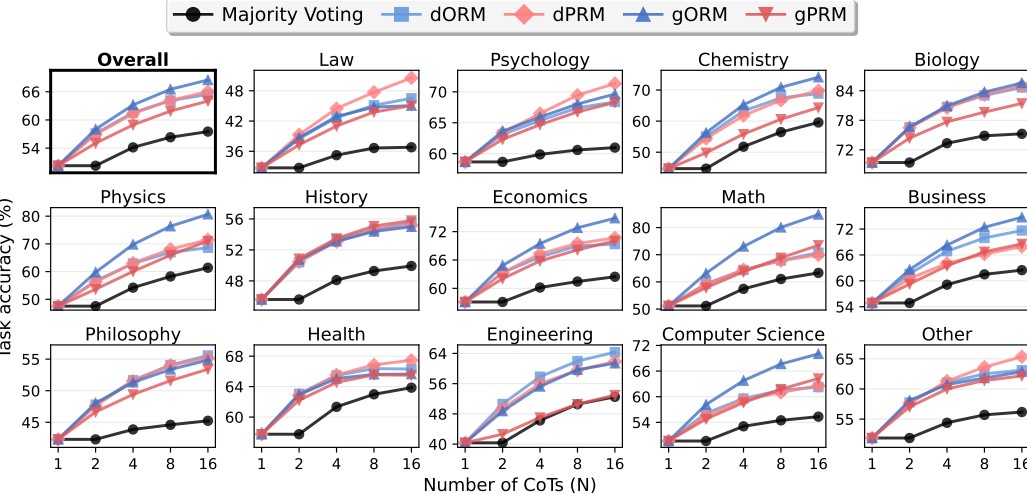

Figure 25: **Weighted majority voting results using Llama-3.1-8B-Instruct** on MMLU-Pro with R1-distill-Qwen-14B backbone for reward models.

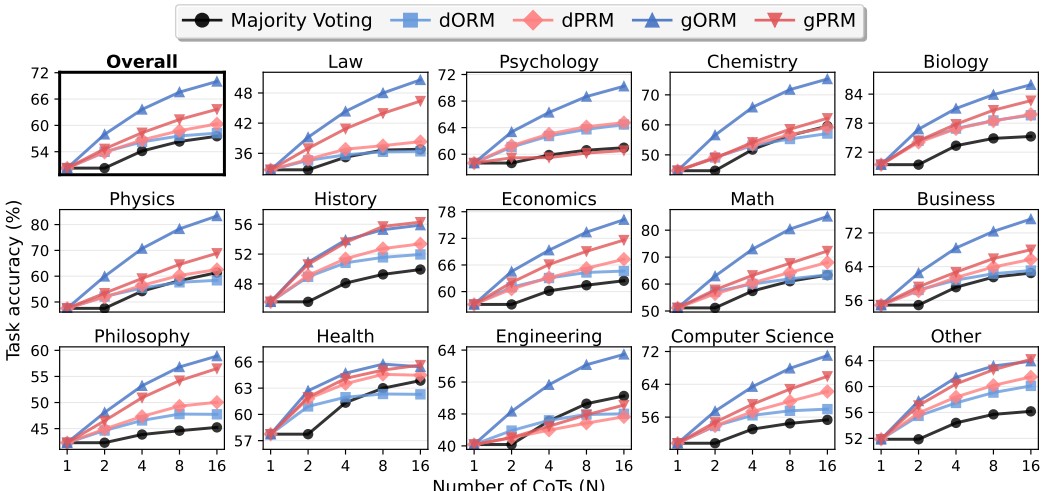

Figure 26: **Best-of-$N$ performance using Llama-3.1-8B-Instruct when trained and evaluated on each domain** of MMLU-Pro with R1-distilled-Llama-8B backbone for reward models.

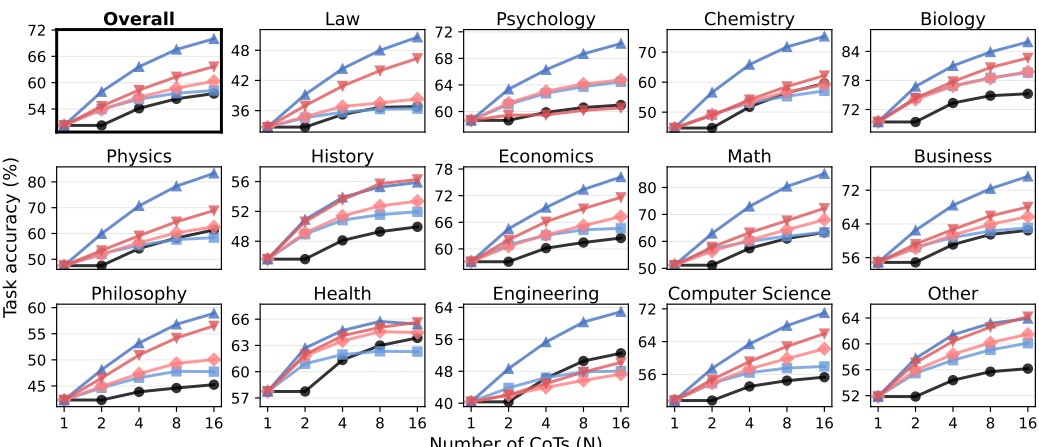

Figure 27: **Weighted majority voting performance using Llama-3.1-8B-Instruct when trained and evaluated on each domain** of MMLU-Pro with R1-distilled-Llama-8B backbone for reward models.

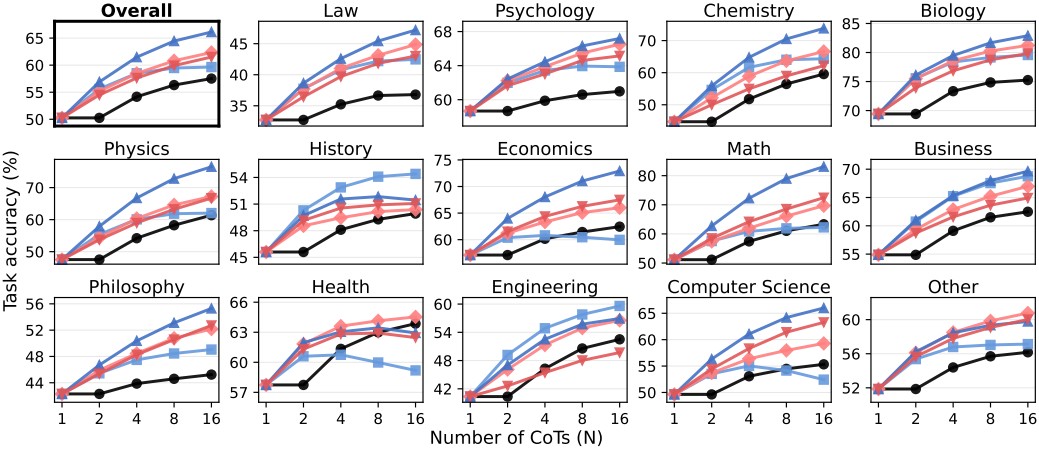

Figure 28: **Best-of-$N$ results using Llama-3.1-8B-Instruct** on MMLU-Pro (Wang et al., 2024c) with R1-distill-Llama-8B backbone for reward models.

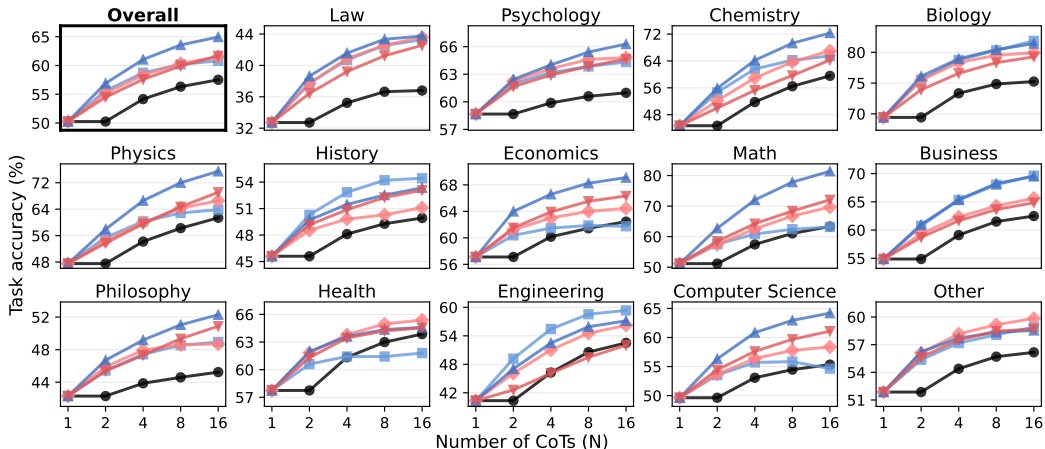

Figure 29: **Weighted majority voting results using Llama-3.1-8B-Instruct** on MMLU-Pro (Wang et al., 2024c) with R1-distill-Llama-8B backbone for reward models.

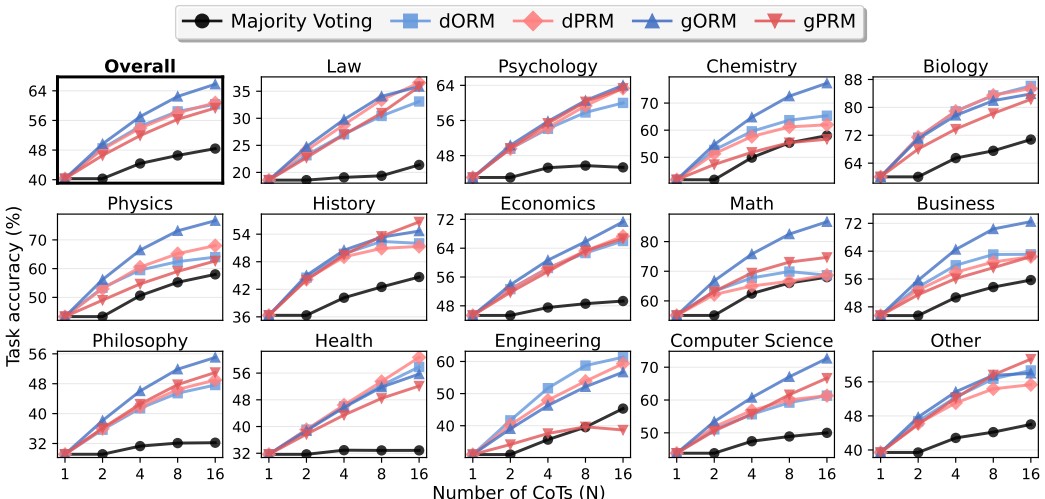

Figure 30: **Best-of-$N$ results using SmolLM3-3B** on MMLU-Pro with R1-distill-Llama-8B backbone for reward models.

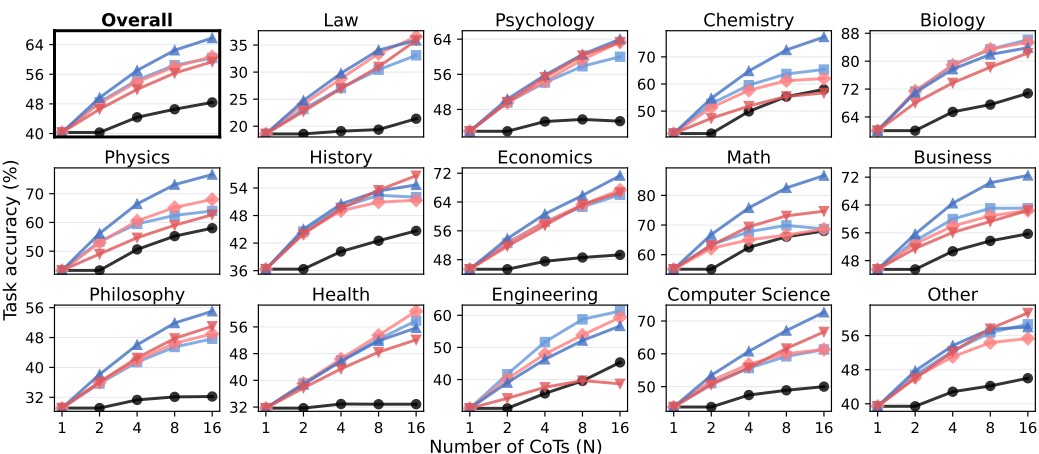

Figure 31: **Weighted majority voting results using SmolLM3-3B** on MMLU-Pro with R1-Distill-Qwen-14B backbone for reward models.

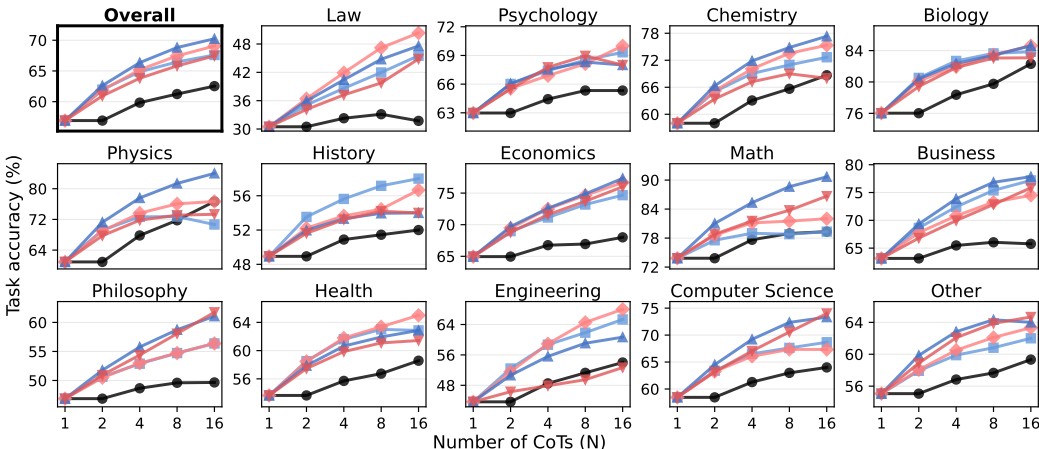

Figure 32: **Best-of-$N$ results using Qwen2.5-7B-Instruct** on MMLU-Pro with R1-distilled-Llama-14B backbone for reward models.

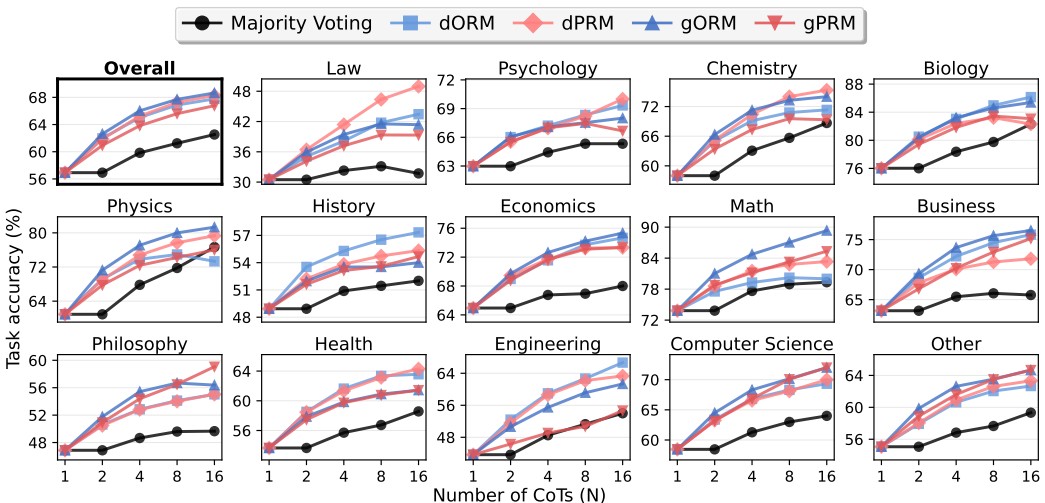

Figure 33: **Weighted majority voting results using Qwen2.5-7B-Instruct** on MMLU-Pro with R1-Distill-Qwen-14B backbone for reward models.

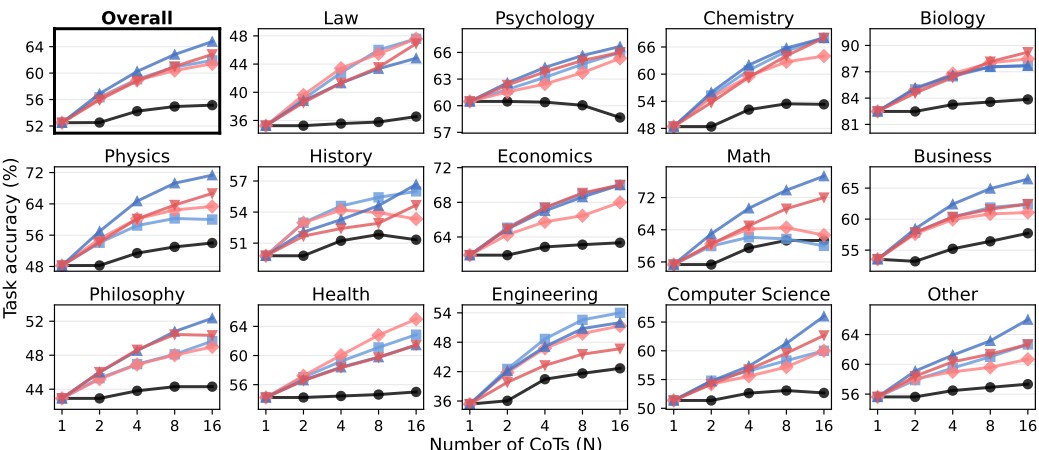

Figure 34: **Best-of-$N$ results using gemma-2-9b-it** on MMLU-Pro with R1-Distill-Qwen-14B backbone for reward models.

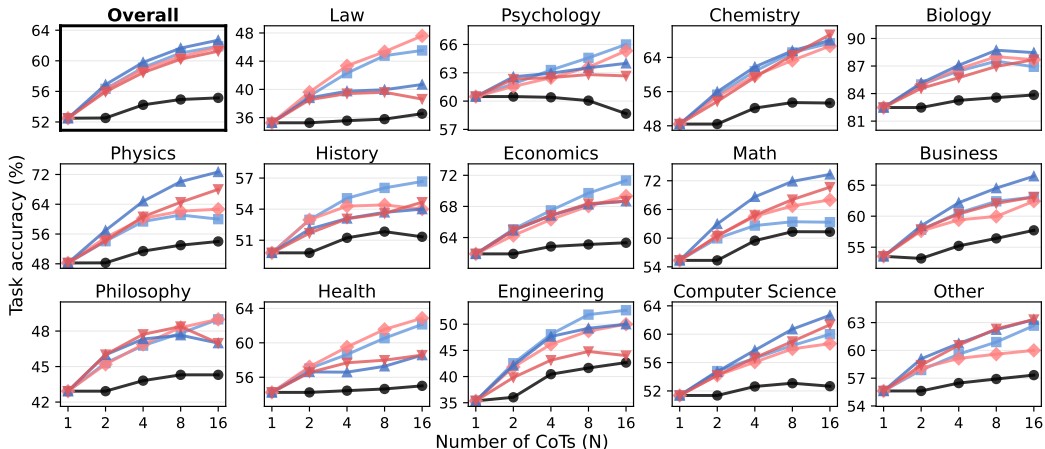

Figure 35: **Weighted majority voting results using gemma-2-9b-it** on MMLU-Pro with R1-Distill-Qwen-14B backbone for reward models.

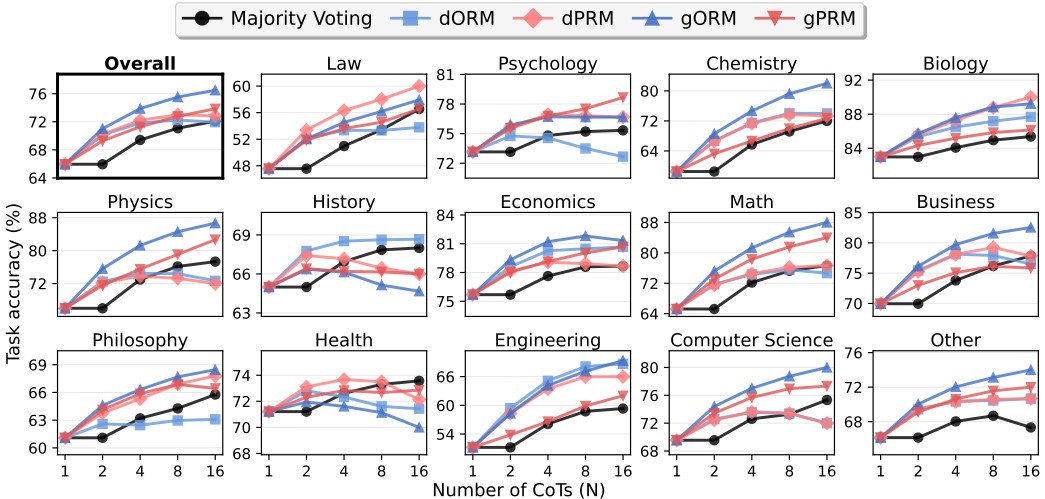

Figure 36: **Best-of-$N$ results using Llama-3.1-70B-Instruct** on MMLU-Pro with R1-Distill-Qwen-14B backbone for reward models.

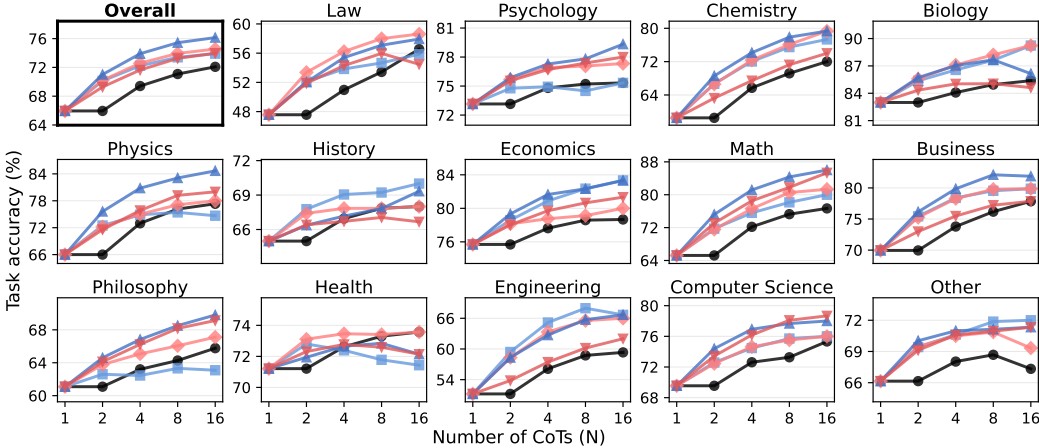

Figure 37: **Weighted majority voting results using Llama-3.1-70B-Instruct** on MMLU-Pro with R1-Distill-Qwen-14B backbone for reward models.

# G ADDITIONAL ANALYSIS

In this section, we present additional analysis on the failure of PRMs.

- Fig. 38: **Majority voting results of Llama-3.1-8B-Instruct vs. CoT length** on MMLU-pro.
- Fig. 39 (the full version of Fig. 9): **results on "aha" CoTs** in ProcessBench using R1-Distill-Qwen-1.5B and -7B.
- Fig. 40 (the full version of Fig. 13): **Outcome-verification results of PRMs vs. label noise** on GSM8K.
- Tab. 3: **Wasserstein distance in the multi-domain setting** before and after filtering for **gO**RM and **gP**RM. To reduce the CoT-length distribution shift (*i.e.*, the Wasserstein distance) for **gP**RM, we apply (i) **label refinement** using Gemini-2.0 Flash (Comanici et al., 2025): due to a parsing issue, **59.96%** of process labels are replaced; and (ii) **relaxation** of the *consensus filtering* rule: when $y = 1$, we keep the verification CoTs $v_{1:L^+}$ with $\hat{z}_{1:T} = 1_T$, and when $y = 0$, we keep $v_{1:L^+}$ if there exists $t \in \{1, \ldots, T\}$ such that $z_t = 0$.
- Tab. 4: **Wasserstein distance in the math domain** before and after filtering for **gO**RM and **gP**RM.
- Tab. 5: **Surviving proportion (%)** under the multi-domain setting. We compare **gO**RM and **gP**RM under (i) **label refinement** using Gemini-2.0 Flash (Comanici et al., 2025), and (ii) **relaxed** *consensus filtering*.
- Tab. 6: Best-of-$N$ results on MMLU-Pro using Llama-3.1-8B-Instruct with **label refinement**. We use R1-distill-Qwen-14B as the backbone for reward models.
- Tab. 7: Best-of-$N$ results on MMLU-Pro using Llama-3.1-8B-Instruct with **filtering relaxation**. We use R1-Distill-Qwen-14B backbone for reward models.
- Tab. 8: Best-of-$N$ results (overall) on MMLU-Pro using Llama-3.1-8B-Instruct by **varying learning rate and LoRA rank $r$ for PRM variants**. We use R1-Distill-Qwen-14B backbone for reward models.
- Fig. 41: Best-of-$N$ results using five different $p_{\text{LLM}}$ on **GPQA-diamond** (Rein et al., 2024).
- Fig. 42: Weighted majority voting results using five different $p_{\text{LLM}}$ on **GPQA-diamond** (Rein et al., 2024).

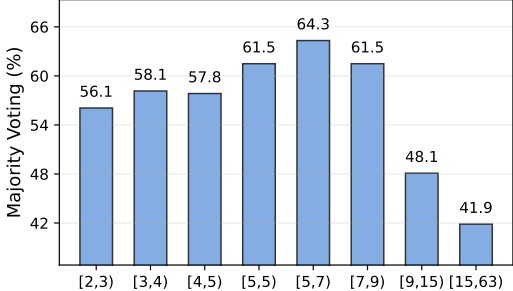

Figure 38: **Majority voting results of Llama-3.1-8B-Instruct vs. CoT length** on MMLU-pro.

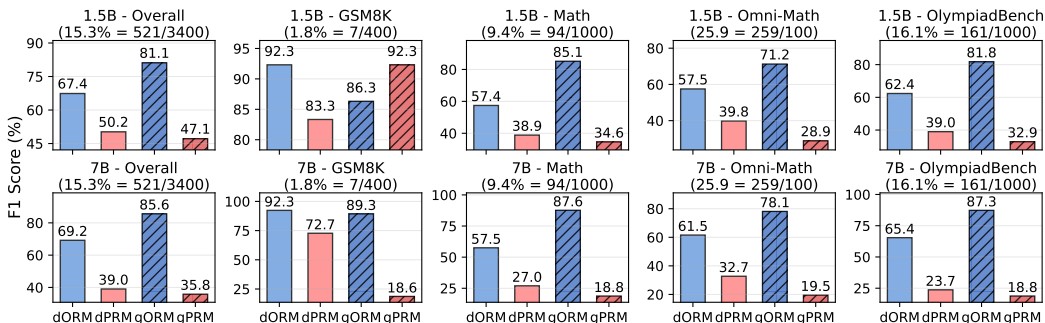

Figure 39: **Performance on "aha" CoTs** in ProcessBench using R1-distill-Qwen-1.5B and -7B.

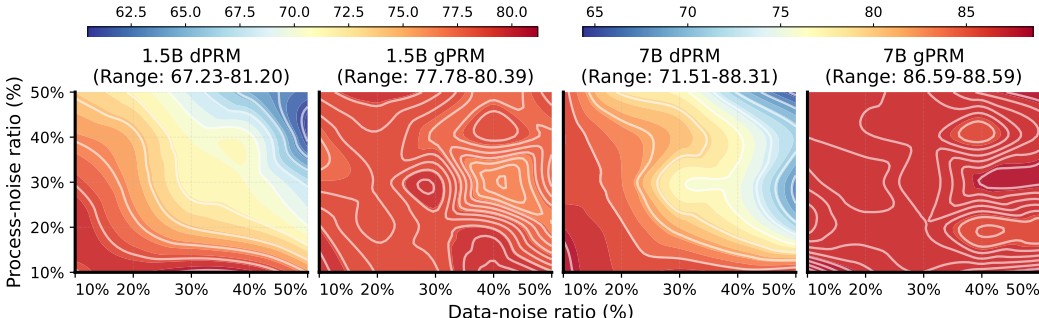

Figure 40: **Outcome-verification results of PRMs vs. label noise** on GSM8K.

Table 3: **Wasserstein distance in the multi-domain setting** before and after filtering for **gO**RM and **gP**RM. To reduce the CoT-length distribution shift (*i.e.*, the Wasserstein distance) for **gP**RM, we apply (i) **label refinement** using Gemini-2.0 Flash (Comanici et al., 2025): due to a parsing issue, **59.96%** of process labels are replaced; and (ii) **relaxation** of the *consensus filtering* rule: when $y = 1$, we keep the verification CoTs $v_{1:L^+}$ with $\hat{z}_{1:T} = 1_T$, and when $y = 0$, we keep $v_{1:L^+}$ if there exists $t \in \{1, \ldots, T\}$ such that $z_t = 0$.

| | Overall | Law | Psychology | Chemistry | Biology |
|---|---|---|---|---|---|
| Train | 0.202 | 0.090 | 0.203 | 0.393 | 0.264 |
| gORM | 0.532 | 0.089 | 0.218 | 1.128 | 0.506 |
| gPRM | 3.083 | 1.284 | 0.742 | 6.922 | 2.039 |
| gPRM (label refinement) | 3.265 (+0.182) | 1.397 (+0.113) | 0.893 (+0.151) | 7.183 (+0.261) | 2.416 (+0.377) |
| gPRM (relaxed filtering) | 2.001 (-1.082) | 0.885 (-0.399) | 0.397 (-0.345) | 4.939 (-1.983) | 1.311 (-0.728) |
| | **Physics** | **History** | **Economics** | **Math** | **Business** |
| Train | 0.628 | 0.069 | 0.311 | 0.167 | 0.322 |
| gORM | 1.201 | 0.154 | 0.564 | 0.282 | 0.491 |
| gPRM | 5.952 | 0.581 | 1.782 | 4.655 | 4.267 |
| gPRM (label refinement) | 6.104 (+0.152) | 0.752 (+0.171) | 2.044 (+0.262) | 4.852 (+0.197) | 4.494 (+0.227) |
| gPRM (relaxed filtering) | 4.371 (-1.581) | 0.203 (-0.378) | 1.094 (-0.688) | 2.571 (-2.084) | 2.777 (-1.490) |
| | **Philosophy** | **Health** | **Engineering** | **Computer science** | **Other** |
| Train | 0.129 | 0.105 | 1.234 | 0.353 | 0.093 |
| gORM | 0.545 | 0.213 | 3.611 | 0.338 | 0.312 |
| gPRM | 1.235 | 0.979 | 12.735 | 3.459 | 0.927 |
| gPRM (label refinement) | 1.299 (+0.064) | 1.157 (+0.178) | 13.058 (+0.323) | 3.742 (+0.283) | 1.030 (+0.103) |
| gPRM (relaxed filtering) | 0.505 (-0.730) | 0.554 (-0.425) | 9.536 (-3.199) | 2.363 (-1.096) | 0.460 (-0.467) |

Table 4: **Wasserstein distance in the math domain** before and after filtering for **gO**RM and **gP**RM.

| | Overall | GSM8K | Math | Omni-Math | OlympiadBench |
|---|---|---|---|---|---|
| Train (PRM800K) | 2.760 | 5.113 | 3.813 | 2.027 | 1.514 |
| gORM | 2.430 | 4.780 | 3.480 | 1.695 | 1.194 |
| gPRM | 1.600 | 3.680 | 2.348 | 1.448 | 1.203 |

Table 5: **Surviving proportion (%)** under the multi-domain setting. We compare **gO**RM and **gP**RM under (i) **label refinement** using Gemini-2.0 Flash (Comanici et al., 2025), and (ii) **relaxed** *consensus filtering*. Please see the caption of Tab. 3 for more details.

| | Overall | Law | Psychology | Chemistry | Biology |
|---|---|---|---|---|---|
| gORM | 51.1 | 51.6 | 28.3 | 71.9 | 42.0 |
| gPRM | 28.0 | 22.7 | 22.8 | 30.1 | 30.4 |
| gPRM (label refinement) | 44.0 (+16.0) | 23.6 (+0.9) | 42.1 (+19.3) | 33.5 (+3.4) | 55.4 (+25.0) |
| gPRM (relaxed filtering) | 45.5 (+17.5) | 24.9 (+2.2) | 43.7 (+20.9) | 35.8 (+5.7) | 54.6 (+24.2) |
| | **Physics** | **History** | **Economics** | **Math** | **Business** |
| gORM | 77.0 | 28.4 | 50.9 | 70.3 | 54.3 |
| gPRM | 34.2 | 26.0 | 37.0 | 30.8 | 24.7 |
| gPRM (label refinement) | 35.8 (+1.6) | 49.3 (+23.3) | 64.9 (+27.9) | 65.5 (+34.7) | 56.0 (+31.3) |
| gPRM (relaxed filtering) | 37.3 (+3.1) | 41.0 (+15.0) | 64.1 (+27.1) | 66.7 (+35.9) | 57.8 (+33.1) |
| | **Philosophy** | **Health** | **Engineering** | **Computer Science** | **Other** |
| gORM | 48.6 | 40.5 | 37.0 | 70.1 | 42.9 |
| gPRM | 26.8 | 27.7 | 13.5 | 38.9 | 31.0 |
| gPRM (label refinement) | 46.6 (+19.8) | 51.3 (+23.6) | 13.9 (+0.4) | 46.9 (+8.0) | 37.5 (+6.5) |
| gPRM (relaxed filtering) | 48.0 (+21.2) | 58.7 (+31.0) | 14.1 (+0.6) | 48.8 (+9.9) | 38.6 (+7.6) |

Table 6: Best-of-$N$ results on MMLU-Pro using Llama-3.1-8B-Instruct with **label refinement**. We use R1-distill-Qwen-14B as the backbone for reward models. The number in parentheses denotes the change after refining the process labels. Please see the caption of Tab. 3 for more details.

| Method | N | | | | |
|---|---|---|---|---|---|
| | 1 | 2 | 4 | 8 | 16 |
| Majority voting | 50.27 | 50.27 | 54.15 | 56.14 | 57.16 |
| dORM | 50.27 | 57.30 | 61.54 | 63.95 | 65.38 |
| dPRM | 50.27 | 57.18 | 61.51 | 64.10 | 65.55 |
| dPRM (label refinement) | 50.27 | 56.99 (-0.19) | 61.41 (-0.10) | 64.38 (+0.28) | 66.57 (+1.02) |
| gORM | 50.27 | 58.24 | 63.88 | 67.82 | 70.02 |
| gPRM | 50.27 | 55.24 | 59.06 | 62.10 | 64.26 |
| gPRM (label refinement) | 50.27 | 54.99 (-0.25) | 58.86 (-0.20) | 62.10 (+0.00) | 64.84 (+0.58) |
| Pass@$N$ | 50.27 | 61.74 | 71.56 | 79.77 | 86.05 |

Table 7: Best-of-$N$ results on MMLU-Pro using Llama-3.1-8B-Instruct with **filtering relaxation**. We use R1-distill-Qwen-14B as the backbone for reward models. The number in paranthesis denotes the change after relaxing the consensus filtering. Please see the caption of Tab. 3 for more details.

| Method | N | | | | |
|---|---|---|---|---|---|
| | 1 | 2 | 4 | 8 | 16 |
| Majority voting | 50.27 | 50.27 | 54.15 | 56.14 | 57.16 |
| dORM | 50.27 | 57.30 | 61.54 | 63.95 | 65.38 |
| dPRM | 50.27 | 57.18 | 61.51 | 64.10 | 65.55 |
| gORM | 50.27 | 58.24 | 63.88 | 67.82 | 70.02 |
| gPRM | 50.27 | 55.24 | 59.06 | 62.10 | 64.26 |
| gPRM (filtering relaxation) | 50.27 | 54.93 (-0.31) | 58.88 (-0.18) | 62.23 (+0.13) | 64.82 (+0.56) |
| Pass@$N$ | 50.27 | 61.74 | 71.56 | 79.77 | 86.05 |

Table 8: Best-of-$N$ results (overall) on MMLU-Pro using Llama-3.1-8B-Instruct by **varying learning rate and LoRA rank $r$ for PRM variants**. We use with R1-distill-Qwen-14B backbone for reward models. The number in parentheses denotes the change.

| Method | Learning rate | $r$ | N | | | | |
|---|---|---|---|---|---|---|---|
| | | | 1 | 2 | 4 | 8 | 16 |
| **Majority voting** | – | – | 50.27 | 50.27 | 54.15 | 56.14 | 57.16 |
| dORM | 1e-4 | 16 | 50.27 | 57.30 | 61.54 | 63.95 | 65.38 |
| dPRM (default) | 1e-4 | 16 | 50.27 | 57.18 | 61.51 | 64.10 | 65.55 |
| dPRM (changed) | 5e-5 | 16 | 50.27 | 56.77 (-0.41) | 60.84 (-0.67) | 62.99 (-1.11) | 64.04 (-1.51) |
| dPRM (changed) | 1e-4 | 32 | 50.27 | 57.01 (-0.17) | 61.31 (-0.20) | 64.11 (+0.01) | 66.02 (+0.47) |
| gORM | 1e-4 | 32 | 50.27 | 58.24 | 63.88 | 67.82 | 70.02 |
| gPRM (default) | 1e-4 | 32 | 50.27 | 55.24 | 59.06 | 62.10 | 64.26 |
| gPRM (changed) | 5e-5 | 32 | 50.27 | 55.09 (-0.15) | 58.94 (-0.12) | 61.96 (-0.14) | 64.26 (+0.00) |
| gPRM (changed) | 1e-4 | 64 | 50.27 | 54.94 (-0.30) | 58.73 (-0.33) | 61.88 (-0.22) | 64.55 (+0.29) |
| **Pass@$N$** | – | – | 50.27 | 61.74 | 71.56 | 79.77 | 86.05 |

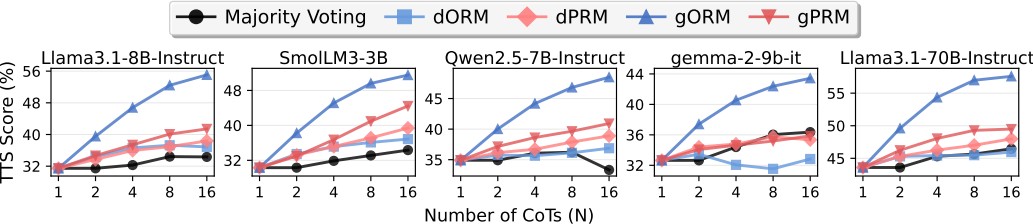

Figure 41: Best-of-$N$ results using five different $p_{\text{LLM}}$ on **GPQA-diamond** (Rein et al., 2024).

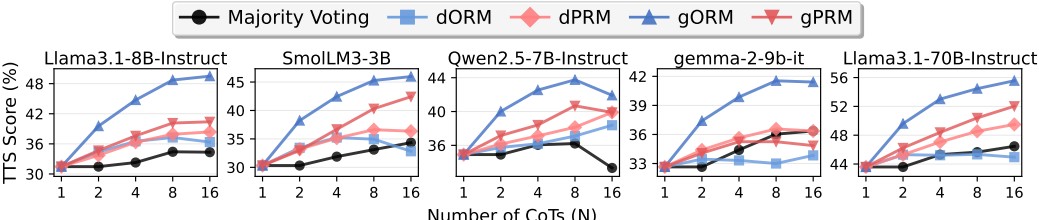

Figure 42: Weighted majority voting results using five different $p_{\text{LLM}}$ on **GPQA-diamond** (Rein et al., 2024).

## H THE USE OF LLMs

We used LLMs solely for light editing such as correcting grammatical errors and polishing some words. They did not contribute to research ideation, experiments, analysis, or substantive writing.

