# OpenReview forum: "Rethinking Reward Models for Multi-Domain Test-Time Scaling"
_ICLR.cc/2026/Conference — Submitted to ICLR 2026_

### Official Review · Reviewer_WbK9 · 2025-10-30

**Soundness:** 3
**Presentation:** 4
**Contribution:** 2
**Rating:** 6
**Confidence:** 3

**Summary:**

This paper compares four types of external verifiers: dPRM, dORM, gPRM, and gORM. (d for discriminative, and g for generative)
Findings are divided by domain and cot-type (short or long) and summarized on the last page of the paper.

In general, the paper suggests ORM-type verifiers to show the best (most robust performance), and to prefer gORM over dORM when compute allows. They hypothesize this is likely because PRMs are likely to accumulate errors or are not a suitable match with the current long-reasoning paradigm, where models can self-recover from errors.

I have mixed feelings about this paper. While, after reading it, I think some of the results are obvious, I understand it's an important part of science for someone to conduct a well-controlled experiment to document and prove such obviousness.

**Strengths:**

While the paper is not of the type that proposes a new method or dataset, it aims to broaden our understanding of external verifiers. And might be a good reference material for those in the industry who have to choose which verifier to use in certain scenarios.

The presentation is nice and I would say fancy.

**Weaknesses:**

(1) I think we can summarize scenarios where we need external verifiers into two cases. (a) at test-time: where we want to boost the performance of a fixed language model. The paper may be helpful to practitioners who need to choose a generator-verifier combination. One thing that worries me is that the paper highly relies on empirical findings, and I'm not sure whether the same rules will apply to new datasets, domains, training datasets, etc. (b) at train-time: where we use external verifiers to guide training. For instance, see Figure 4 of [1], which provides an experiment on whether their analysis on verifiers can be applied to PPO-style training. The paper lacks analysis in this direction. In a nutshell, while this may appear a bit off-topic for the authors, I think the findings are heavily reliant on empirical observations on the test-time side and lack exploration in the train-time side.


(2) The paper emphasizes that the limitations of past works are being narrowed down to math only. However, I think this paper still lacks exploration in non-math domains. The dataset used to train the verifiers for MMLU-Pro is automatically generated via Llama-3.1-8/70B. Evaluation is done only on MMLU-Pro, a very standardized MCQA benchmark. Especially with the diverse evaluation datasets flooding recently, it would have been better for the authors to collect a diverse set of benchmarks (even if they had to sample it).

(3) An increasing usage of external verifiers is in training where we the community now aims for harder and harder datasets. It would have been interesting to include the results in harder datasets.

**Questions:**

See weakness.

---

> ### Author Response · Authors · 2025-11-19
> **Response to Reviewer WbK9 (1/3)**
>
> We sincerely appreciate your time and constructive comments, which improve our paper. We address your concerns as follows.
>
> ---
>
> > **[Q1]** I have mixed feelings about this paper. While, after reading it, I think some of the results are obvious, I understand it's an important part of science for someone to conduct a well-controlled experiment to document and prove such obviousness.
>
> - We sincerely appreciate the reviewer for **highlighting this perspective** and for acknowledging the **importance of conducting well-controlled experiments**, even when some findings may initially appear intuitive or obvious.
> - We agree that certain results (e.g., the failure of PRMs on "aha" CoTs, the increasing risk of PRMs with longer CoT sequences, and the vulnerability arising from noisy process labels) may seem obvious in hindsight.
> - However, we believe that our study provides a new and valuable intuition by (i) identifying the **potential risks of PRMs in multi-domain settings** and (ii) **connecting empirical observations with a concrete analysis framework**, which, to the best of our knowledge, has not been systematically investigated in prior work.
> - Lastly, we assume that the perceived obviousness may stem from the **clarity of our presentation**: our motivation, experimental design, and results may make the conclusions feel natural and expected. We view this as a **strength**, as our goal is to provide a clear, reproducible, and well-grounded understanding of PRM behaviors.
>
>
> ---
>
> >**[Q2]** I think we can summarize scenarios where we need external verifiers into two cases. (a) at test-time: where we want to boost the performance of a fixed language model. The paper may be helpful to practitioners who need to choose a generator-verifier combination. One thing that worries me is that the paper highly relies on empirical findings, and I'm not sure whether the same rules will apply to new datasets, domains, training datasets, etc. (b) at train-time: where we use external verifiers to guide training. For instance, see Figure 4 of [1], which provides an experiment on whether their analysis on verifiers can be applied to PPO-style training. The paper lacks analysis in this direction. In a nutshell, while this may appear a bit off-topic for the authors, I think the findings are heavily reliant on empirical observations on the test-time side and lack exploration in the train-time side.
>
> - Although reference [1] is missing in your review, we believe that we correctly understand your concern: our study **focuses solely on test-time scaling**.
> - We sincerely agree that **(b) training-time usage** of reward models is another important (and potentially even more impactful) research direction. However, we respectfully argue that its evaluation is **beyond the scope** of our study for the following reasons:
> - As indicated in our title, we investigate the effectiveness of four reward model variants for **multi-domain test-time scaling**, following standardized evaluation protocols used in prior work [2, 3, 4].
> - **In line with prior studies**, we focus on verifiable tasks (e.g., math and multiple-choice QA), where reward signals can be defined reliably, clearly, and efficiently.
> - For these tasks, even considering **potential generalization errors** of reward models and their **computational overhead**, we believe there is **no strong motivation** to post-train LLMs with rewards derived from reward models.
> - While PRMs have been shown to be more effective for LLM post-training [5] by offering **denser intermediate supervision** compared to ground-truth outcome-level rewards in verifiable settings (e.g., GRPO [6]), they may also introduce **noisy training signals** due to imperfections in process reward modeling. Similarly, the benefits of using ORMs for post-training in these verifiable-task settings remain unclear.
> - Therefore, the most meaningful setting in which to investigate (b) the training-time usage of reward models would be **open-ended, multi-domain tasks**, where verifiable rewards **cannot be defined reliably, clearly, or efficiently**. We consider such an exploration to be **beyond the scope** of our current study (please see our response to **[Q1]** of reviewer TwoX for more details).
> - We will include this clarification in the final version.
>
> ---

---

> ### Author Response · Authors · 2025-11-19
> **Response to Reviewer WbK9 (2/3)**
>
> ---
>
> >**[Q3]** The paper emphasizes that the limitations of past works are being narrowed down to math only. However, I think this paper still lacks exploration in non-math domains. The dataset used to train the verifiers for MMLU-Pro is automatically generated via Llama-3.1-8/70B. Evaluation is done only on MMLU-Pro, a very standardized MCQA benchmark. Especially with the diverse evaluation datasets flooding recently, it would have been better for the authors to collect a diverse set of benchmarks (even if they had to sample it).
>
> - We sincerely appreciate the reviewer for raising this point, which helps strengthen our study. To address your concern, we conduct additional experiments on a more challenging dataset covering biology, chemistry, and physics: the **GPQA-diamond** dataset [7].
> - For each question, we generate $N = 16$ CoTs using the following problem-solving LLMs, which are identical to the models used in our MMLU-Pro experiments (i.e., $p\_\mathtt{LLM}$ in `L105`): (i) **Llama-3.1-8B-Instruct**, (ii) **SmolLM3-3B**, (iii) **Qwen2.5-7B-Instruct**, (iv) **gemma-2-9B-it**, and (v) **Llama-3.1-70B-Instruct**.
>
> - Note that we **do not train** the reward models on the GPQA-diamond dataset and directly use the reward models to evaluate their effectiveness on test-time scaling.
>
>     **[Table WbK9-1]** Best-of-$N$ results using **Llama-3.1-8B-Instruct** on GPQA-diamond. The best results are **bolded**. The number in parentheses denotes the improvement over the second-best result.
>
>     |Method|$N=1$|$N=2$|$N=4$|$N=8$|$N=16$|
>     |:-|:-:|:-:|:-:|:-:|:-:|
>     |MV|31.46|31.46|32.28|34.42|34.34|
>     |$\texttt{dORM}$|31.46|34.38|36.61|37.29|36.87|
>     |$\texttt{dPRM}$|31.46|33.71|35.96|36.93|38.38|
>     |$\texttt{gORM}$|31.46|**39.53** (**+4.85**)|**46.78** (**+9.38**)|**52.42** (**+12.32**)|**55.05** (**+13.64**)|
>     |$\texttt{gPRM}$|31.46|34.68|37.40|40.10|41.41|
>     |Pass@$N$|31.46|47.66|65.22|80.25|88.38|
>
>
>     **[Table WbK9-2]** Best-of-$N$ results using **SmolLM3-3B** on GPQA-diamond. TThe best results are **bolded**. The number in parentheses denotes the improvement over the second-best result.
>
>     |Method|$N=1$|$N=2$|$N=4$|$N=8$|$N=16$|
>     |:-|:-:|:-:|:-:|:-:|:-:|
>     |MV|30.32|33.44|31.88|33.12|34.34|
>     |$\texttt{dORM}$|30.32|33.44|35.14|36.10|36.87|
>     |$\texttt{dPRM}$|30.32|33.20|35.16|37.14|39.39|
>     |$\texttt{gORM}$|30.32|**38.23** (**+4.79**)|**45.11** (**+8.38**)|**49.61** (**+8.66**)|**51.51** (**+7.07**)|
>     |$\texttt{gPRM}$|30.32|32.97|36.73|40.95|44.44|
>     |Pass@$N$|30.32|46.70|63.67|77.31|84.85|
>
>
>     **[Table WbK9-3]** Best-of-$N$ results using **Qwen2.5-7B-Instruct** on GPQA-diamond. The best results are **bolded**. The number in parentheses denotes the improvement over the second-best result.
>
>     |Method|$N=1$|$N=2$|$N=4$|$N=8$|$N=16$|
>     |:-|:-:|:-:|:-:|:-:|:-:|
>     |MV|34.91|34.91|36.06|36.22|33.33|
>     |$\texttt{dORM}$|34.91|35.75|35.65|36.13|36.87|
>     |$\texttt{dPRM}$|34.91|36.15|36.70|37.89|38.89|
>     |$\texttt{gORM}$|34.91|**40.01** (**+2.85**)|**44.17** (**+5.55**)|**46.81** (**+7.16**)|**48.48** (**+7.57**)|
>     |$\texttt{gPRM}$|34.91|37.16|38.62|39.65|40.91|
>     |Pass@$N$|34.91|47.69|60.48|71.25|78.79|
>
>     **[Table WbK9-4]** Best-of-$N$ results using **gemma-2-9b-it** on GPQA-diamond. The best results are **bolded**. The number in parentheses denotes the improvement over the second-best result.
>
>     |Method|$N=1$|$N=2$|$N=4$|$N=8$|$N=16$|
>     |:-|:-:|:-:|:-:|:-:|:-:|
>     |MV|32.65|32.65|34.40|36.03|36.36|
>     |$\texttt{dORM}$|32.65|33.45|32.06|31.54|32.83|
>     |$\texttt{dPRM}$|32.65|34.40|34.79|35.74|35.35|
>     |$\texttt{gORM}$|32.65|**37.40** (**+3.00**)|**40.55** (**+5.76**)|**42.38** (**+6.64**)|**43.43** (**+7.57**)|
>     |$\texttt{gPRM}$|32.65|34.07|34.64|35.19|35.86|
>     |Pass@$N$|32.65|42.45|51.84|60.63|66.67|
>
>     **[Table WbK9-5]** Best-of-$N$ results using **Llama-3.1-70B-Instruct** on GPQA-diamond. The best results are **bolded**. The number in parentheses denotes the improvement over the second-best result.
>
>     |Method|$N=1$|$N=2$|$N=4$|$N=8$|$N=16$|
>     |:-|:-:|:-:|:-:|:-:|:-:|
>     |MV|43.59|43.59|45.31|45.65|46.46|
>     |$\texttt{dORM}$|43.59|45.30|45.42|45.48|45.96|
>     |$\texttt{dPRM}$|43.59|45.29|46.26|47.01|47.98|
>     |$\texttt{gORM}$|43.59|**49.60** (**+3.37**)|**54.35** (**+6.30**)|**56.98** (**+7.69**)|**57.58** (**+8.09**)|
>     |$\texttt{gPRM}$|43.59|46.23|48.05|49.29|49.49|
>     |Pass@$N$|43.59|58.10|71.88|82.76|89.39|

---

> ### Author Response · Authors · 2025-11-19
> **Response to Reviewer WbK9 (3/3)**
>
> - In `Tables WbK9-1, 2, 3, 4, and 5`, we first observe that $\texttt{dORM}$ performs similarly to $\texttt{dPRM}$, consistent with the results reported in `Figs. 6 and 7` on MMLU-Pro.
> - In `Table WbK9-4`, gemma-2-9b-it often generates **non-alphabetical answers**, for example: “The answer is (**You'll need to determine the correct answer based on the stereochemistry analysis mentioned above**)” or “The answer is (**none of the above**)”. Such outputs make it difficult for reward models to evaluate responses, causing them to **underperform compared to majority voting**. Even under these noisy conditions, $\texttt{gORM}$ still shows **robust improvements** over majority voting.
> - Interestingly, $\texttt{gPRM}$ shows **meaningful gains over the discriminative variants** (i.e., $\texttt{dORM}$ and $\texttt{dPRM}$), except when using gemma-2-9b-it. Consistent with the results in `Fig. 8`, we attribute this improvement to the **greater data efficiency of generative reward models**. Since we do not train reward models on GPQA-diamond, the discriminative variants may struggle to generalize to unseen dataset.
> - As expected, we observe that $\texttt{gORM}$ **delivers consistent and strong gains** over all other reward model variants (e.g., +4.85/+9.38/+12.32/+13.64 over the second-best for $N=2/4/8/16$ using Llama-3.1-8B-Instruct).
> - We believe that these results empirically strengthen our recommendation to use **generative outcome verification** for **multi-domain deployment and test-time scaling**.
> - We incorporated `Tables WbK9-1, 2, 3, 4, and 5` into `Figs. 41 and 42` in `Appendix G`.
>
>
> ---
>
> >**[Q4]** An increasing usage of external verifiers is in training where we the community now aims for harder and harder datasets. It would have been interesting to include the results in harder datasets.
>
> - We fully agree with your point. As discussed in our response to **[Q2]**, we believe that its importance becomes more evident in **open-ended generation tasks**, which correspond to comparatively more challenging tasks and datasets.
> - However, we consider this direction to be **beyond the scope** of the current study, and we will include this clarification in the revised version.
> - Instead, we conduct experiments on the GPQA-diamond dataset in response to **[Q3]**, which we assume to be a more challenging benchmark. For example, the majority voting performance ($N=16$) of Llama-3.1-8B-Instruct is **57.16** on MMLU-Pro, whereas it is **34.34** on GPQA-diamond.
> - We believe that the results in `Tables WbK9-1, 2, 3, 4, and 5` clearly address the concern regarding whether our conclusion remains consistent on a more challenging dataset.
>
>
> ---
>
> ### Reference
>
> [1] Missing in the review.
>
> [2] Zeng, Thomas, et al. "VersaPRM: Multi-domain process reward model via synthetic reasoning data." ICML 2025.
>
> [3] Zhang, Lunjun, et al. "Generative verifiers: Reward modeling as next-token prediction." ICLR 2025.
>
> [4] Khalifa, Muhammad, et al. "Process reward models that think." arXiv 2025.
>
> [5] Yin, Zhangyue, et al. "Dynamic and generalizable process reward modeling." ACL. 2025.
>
> [6] Shao, Zhihong, et al. "DeepSeekMath: Pushing the limits of mathematical reasoning in open language models." arXiv 2024.
>
> [7] Rein, David, et al. "GPQA: A graduate-level google-proof q&a benchmark." First Conference on Language Modeling. 2024
>
> ---

---

> > ### Comment · Reviewer_WbK9 · 2025-11-24
> >
> > Thanks to the authors for the comprehensive additional experiments. I would like to give an updated rating of 7, but since that option isn't available in the system, I will instead raise the contribution score from 2 to 4.

---

> > > ### Author Response · Authors · 2025-11-25
> > >
> > > Thank you for your thoughtful consideration and positive assessment of our study.
> > > Please feel free to contact us if you have any remaining concerns or questions.

---

### Official Review · Reviewer_YvuS · 2025-10-31

**Soundness:** 3
**Presentation:** 3
**Contribution:** 3
**Rating:** 6
**Confidence:** 3

**Summary:**

This paper challenges the belief that process reward models (PRMs) are superior to outcome reward models (ORMs). While PRMs excel in narrow math domains, this study conducts the first unified evaluation of four reward model variants (dORM, dPRM, gORM, gPRM) across diverse domains. The key finding is that Generative ORM (gORM) is the most robust and delivers significant and consistent gains across every tested domain. In this multi-domain setting, dORM performs on par with dPRM, and gPRM is not competitive. The authors demonstrate that PRMs fail because their stepwise error aggregation compounds errors as reasoning length grows. PRMs also struggle to evaluate long reasoning chains that involve self-correction ("aha" moments), a common feature in complex, non-math domains.

**Strengths:**

1. This paper directly challenges the widely held belief that Process Reward Models (PRMs) are superior to Outcome Reward Models (ORMs). It correctly identifies that this consensus was formed from studies on narrow, math-adjacent domains. By conducting the first unified evaluation of four model variants (dORM, dPRM, gORM, gPRM) across diverse domains, it provides a more generalizable finding: gORM (Generative ORM) is the most robust and yields significant and consistent gains across every tested domain.

1. The authors don't just present what happens, but why. It offers a theoretical analysis showing that the log-error lower bound for PRMs (dPRM and gPRM) grows linearly with the length of the reasoning chain ($T$). In contrast, the error bound for ORMs is independent of $T$. This provides a formal explanation for why PRMs fail on the longer, more complex reasoning chains found in multi-domain settings.

1. This paper diagnoses specific, practical failure modes of PRMs that dPRM is highly sensitive to label noise, which is prevalent in multi-domain datasets auto-labeled by LLMs. It also finds that gPRM suffers from a severe shift in its CoT-length distribution because its strict consensus filtering removes too many long CoTs, fatally mismatching its training data to the test set.

**Weaknesses:**

1. The study's evaluations are confined to tasks with verifiable outcomes (i.e., math and multiple-choice questions). The authors also explicitly state that these findings may not generalize to open-ended generation tasks like dialogue or summarization.

1. The authors do not perform exhaustive tuning, as mentioned in the appendix. This may have led to an unfair comparison. The poor performance of PRM-style models (dPRM, gPRM) on this new, noisy, multi-domain task might be because their inherited hyperparameters were not optimal, while the hyperparameters inherited by the ORM-style models just happened to be more robust.

1. The paper strongly demonstrates that gPRM fails because its strict consensus filtering mechanism leads to a severe lack of long CoT samples in its training data, thereby creating a length distribution shift. This weakness might be caused by gPRM's data collection pipeline, not by the concept of gPRM itself. A gPRM that adopted a different data collection strategy (e.g., less strict filtering or sampling specifically for long CoTs) might still be very competitive.

**Questions:**

1. The dPRM's failure is attributed to its sensitivity to label noise from the single Llama-3.1-70B annotator. Is this a fundamental flaw of dPRM, or an artifact of this specific annotator's noise profile? If regenerating the process labels with a more advanced model, can the dPRM's performance be recovered with higher-quality process labels?

---

> ### Author Response · Authors · 2025-11-19
> **Response to Reviewer YvuS (1/6)**
>
> We sincerely appreciate your time and constructive comments, which improve our paper. We address your concerns as follows.
>
> ---
>
> >**[Q1]** The study's evaluations are confined to tasks with verifiable outcomes (i.e., math and multiple-choice questions). The authors also explicitly state that these findings may not generalize to open-ended generation tasks like dialogue or summarization.
>
> - We sincerely appreciate the reviewer for raising this constructive comment. We repectfully agree that extending our analysis to open-ended tasks is an important future direction. However, we kindly note that our focus on verifiable tasks (e.g., math and multiple-choice QA) is a **deliberate methodological choice** for the following reasons.
>
> - Our primary goal is to analyze the key properties of different reward models (process vs. outcome, generative vs. discriminative). To do this rigorously, we require an **objective ground truth**. Following prior work on reasoning and test-time scaling [1, 2, 3], we therefore focus on verifiable task settings, which allow us to isolate architectural differences among the four models and evaluate their core capabilities without the confounding factors inherent in subjective evaluation.
>
> - By contrast, the evaluation of open-ended tasks lacks objective ground truth. A verification score would need to be compared against **human preference judgments** or **another LLM-as-a-judge**. This would shift the focus of our study from understanding reward model behavior to benchmarking alignment with a **potentially biased judge**, undermining our objective and reproducible evaluation framework. As the community has increasingly shifted toward objective evaluation on verifiable tasks (e.g., GRPO [4]), we believe our choice aligns with this trend.
>
> - Therefore, we believe that our **current setup provides the most reliable framework** for analyzing reward model designs under a standardized, objective evaluation protocol.
>
> - Instead, during this rebuttal, we conduct additional experiments on **a more challenging multiple-choice QA dataset**, GPQA-diamond [5], which spans biology, chemistry, and physics. Its difficulty is empirically supported by the majority-voting performance of Llama-3.1-8B-Instruct with $N=16$: **57.16 on MMLU-Pro** versus **34.34 on GPQA-diamond** in our experiments.
>
> - As expected, we observe that $\texttt{gORM}$ **delivers consistent and strong gains** over all other rewardmodel variants (e.g., +4.85/+9.38/+12.32/+13.64 over the second-best for $N=2/4/8/16$ using Llama-3.1-8B-Instruct). Please see our response to **[Q4]** of reviewer WbK9 for more details.
>
> - Despite the scope of our analysis, we would like to discuss our **hypothesis** regarding open-ended tasks: our central conclusion that the generative outcome reward model ($\texttt{gORM}$) performs best for long and complex reasoning chains is likely to **generalize to open-ended tasks**.
>
> - A major limitation we identify in PRMs is that they cannot adequately reward "aha moments" or self-correction. We believe that this issue is **not unique to verifiable tasks**. For instance, a long narrative may begin with weaker opening paragraphs but develop into a compelling conclusion. A PRM-based preference model, trained to myopically penalize each locally suboptimal segment, would likely fail to capture the emergent quality of the story, whereas a $\texttt{gORM}$-based holistic model could better evaluate the piece as a whole.
>
> - We will add a detailed discussion of this limitation and our hypothesis regarding the generalization of our conclusion to open-ended tasks in the final version of the paper. We also incorporated the results on GPQA-diamond into `Figs. 41 and 42` in `Appendix G`.
>
> ---

---

> ### Author Response · Authors · 2025-11-19
> **Response to Reviewer YvuS (2/6)**
>
> ---
>
> > **[Q2]** The authors do not perform exhaustive tuning, as mentioned in the appendix. This may have led to an unfair comparison. The poor performance of PRM-style models (dPRM, gPRM) on this new, noisy, multi-domain task might be because their inherited hyperparameters were not optimal, while the hyperparameters inherited by the ORM-style models just happened to be more robust.
>
> - We sincerely appreciate the reviewer for raising this concern. First, we kindly note that we **rigorously follow prior work** for all hyperparameters (e.g., learning rate, scheduler, LoRA rank $r$ and scaling $\alpha$, weight decay, etc.). Specifically, we follow VersaPRM [1] for $\texttt{dPRM}$ and ThinkPRM [3] for $\texttt{gPRM}$, respectively.
>
> - However, we agree that the chosen hyperparameters might favor ORM variants to some extent. To address this concern, we conduct additional experiments by varying two key hyperparameters: (i) **learning rate** and (ii) **LoRA rank $r$**, for both $\texttt{dPRM}$ and $\texttt{gPRM}$. The results are summarized below:
>
>     **[Table YvuS-1]** Best-of-$N$ results (overall) on MMLU-Pro using Llama-3.1-8B-Instruct by **varying hyperparameters of PRM variants**. The number in paranthesis denotes the change over default.
>
>     |Method|Learning rate|LoRA rank ($r$)|$N=1$|$N=2$|$N=4$|$N=8$|$N=16$|
>     |:-|:-:|:-:|:-:|:-:|:-:|:-:|:-:|
>     |MV|-|-|50.27|50.27|54.15|56.14|57.16|
>     |$\texttt{dORM}$|1e-4|16|50.27|57.30|61.54|63.95|65.38|
>     |$\texttt{dPRM}$ (default)|1e-4|16|50.27|57.18|61.51|64.10|65.55|
>     |$\texttt{dPRM}$ (changed)|5e-5|16|50.27|56.77 (**-0.41**)|60.84 (**-0.67**)|62.99 (**-1.11**)|64.04 (**-1.51**)|
>     |$\texttt{dPRM}$ (changed)|1e-4|32|50.27|57.01 (**-0.17**)|61.31 (**-0.20**)|64.11 (**+0.01**)|66.02 (**+0.47**)|
>     |$\texttt{gORM}$|1e-4|32|50.27|58.24|63.88|67.82|70.02|
>     |$\texttt{gPRM}$ (default)|1e-4|32|50.27|55.24|59.06|62.10|64.26|
>     |$\texttt{gPRM}$ (changed)|5e-5|32|50.27|55.09 (**-0.15**)|58.94 (**-0.12**)|61.96 (**-0.14**)|64.26 (**+0.00**)|
>     |$\texttt{gPRM}$ (changed)|1e-4|64|50.27|54.94 (**-0.30**)|58.73 (**-0.33**)|61.88 (**-0.22**)|64.55 (**+0.29**)|
>     |Pass@$N$|-|-|50.27|61.74|71.56|79.77|86.05|
>
> - As shown in `Table YvuS-1`, these hyperparameter variations **do not** change the performance trend significantly.
> - We believe this further supports the validity of our analysis in `§4`, including the discussion on (i) “aha” CoTs, (ii) PRM risks with increasing CoT length, (iii) PRM label noise, and (iv) the consensus-induced length shift.
> - We incorporated the results of `Table YvuS-1` into `Table 8` in `Appendix G`.
>
> ---

---

> ### Author Response · Authors · 2025-11-19
> **Response to Reviewer YvuS (3/6)**
>
> ---
> > **[Q3]** The paper strongly demonstrates that gPRM fails because its strict consensus filtering mechanism leads to a severe lack of long CoT samples in its training data, thereby creating a length distribution shift. This weakness might be caused by gPRM's data collection pipeline, not by the concept of gPRM itself. A gPRM that adopted a different data collection strategy (e.g., less strict filtering or sampling specifically for long CoTs) might still be very competitive.
> - We sincerely appreciate the reviewer’s thoughtful understanding. We kindly note that the data-processing pipeline is also a **defining property of $\texttt{gPRM}$ in prior work** [3, 6].
> - Nevertheless, this is an **excellent observation**, and we had already explored it during the preliminary phase of our study. Specifically, we attempted to relax the *consensus filtering* of $\texttt{gPRM}$ by relying only on the ground-truth (GT) outcome label $y$ together with the parsed process labels $\hat{z}_{1:\tilde{T}}$.
> - In this relaxed version, we retain a verification CoT $v_{1:L^+}$ if:
>
>   (i) for $y=1$: all parsed labels are one (i.e., $\hat{z}_{1:T} = 1_T$);
>   (ii) for $y=0$: at least one parsed label is zero (i.e., $\hat{z}_t = 0$ for some $t \in \\{1,\ldots,T\\}$).
>
>     **[Table YvuS-2]** The surviving proportion (%) before and after **filtering relaxation**. The number in parentheses denotes the change after relaxing the consensus filtering.
>     |Method|Overall|Law|Psychology|Chemistry|Biology|Physics|History|Economics|Math|Business|Philosophy|Health|Engineering|CS|Other|
>     |:-|:-:|:-:|:-:|:-:|:-:|:-:|:-:|:-:|:-:|:-:|:-:|:-:|:-:|:-:|:-:|
>     |$\texttt{gORM}$|51.1|51.6|28.3|71.9|42.0|77.0|28.4|50.9|70.3|54.3|48.6|40.5|37.0|70.1|42.9|
>     |$\texttt{gPRM}$|28.0|22.7|22.8|30.1|30.4|34.2|26.0|37.0|30.8|24.7|26.8|27.7|13.5|38.9|31.0|
>     |$\texttt{gPRM}$ (**filtering relaxation**)|45.5 (**+17.5**)|24.9 (**+2.2**)|43.7 (**+20.9**)|35.8 (**+5.7**)|54.6 (**+24.2**)|37.3 (**+3.1**)|41.0 (**+15.0**)|64.1 (**+27.1**)|66.7 (**+35.9**)|57.8 (**+33.1**)|48.0 (**+21.2**)|58.7 (**+31.0**)|14.1 (**+0.6**)|48.8 (**+9.9**)|38.6 (**+7.6**)|
>
> - In `Table YvuS-2`, we report the proportion of verification CoTs that survived consensus filtering before and after applying the relaxation. We observe that relaxation **dramatically increases (+17.5% overall)** the surviving proportion of verification CoTs, i.e., the training data for $\texttt{gPRM}$.
>
>     **[Table YvuS-3]** Wasserstein distance before and after **filtering relaxation**. The number in paranthesis denotes the change after relaxing the consensus filtering.
>
>     |Method|Overall|Law|Psychology|Chemistry|Biology|Physics|History|Economics|Math|Business|Philosophy|Health|Engineering|CS|Other|
>     |:-|:-:|:-:|:-:|:-:|:-:|:-:|:-:|:-:|:-:|:-:|:-:|:-:|:-:|:-:|:-:|
>     |Train|0.202|0.090|0.203|0.393|0.264|0.628|0.069|0.311|0.167|0.322|0.129|0.105|1.234|0.353|0.093|
>     |$\texttt{gORM}$|0.532|0.089|0.218|1.128|0.506|1.201|0.154|0.564|0.282|0.491|0.545|0.213|3.611|0.338|0.312|
>     |$\texttt{gPRM}$|3.083|1.284|0.742|6.922|2.039|5.952|0.581|1.782|4.655|4.267|1.235|0.979|12.735|3.459|0.927|
>     |$\texttt{gPRM}$ (**filtering relaxation**)|2.001 (**−1.082**)|0.885 (**−0.399**)|0.397 (**−0.345**)|4.939 (**−1.983**)|1.311 (**−0.728**)|4.371 (**−1.581**)|0.203 (**−0.378**)|1.094 (**−0.688**)|2.571 (**−2.084**)|2.777 (**−1.490**)|0.505 (**−0.730**)|0.554 (**−0.425**)|9.536 (**−3.199**)|2.363 (**−1.096**)|0.460 (**−0.467**)|
>
> - As shown in `Table YvuS-3`, the Wasserstein distance of $\texttt{gPRM}$ is also slightly reduced after applying filtering relaxation (**-1.082 overall**). However, the **overall gap (2.001) remains much larger** than that of $\texttt{gORM}$ (0.532), particularly in domains such as engineering (9.536 vs. 3.611).
>
>     **[Table YvuS-4]** Best-of-$N$ results (overall) on MMLU-Pro using Llama-3.1-8B-Instruct **filtering relaxation**. The number in paranthesis denotes the change after relaxing the consensus filtering.
>
>     |Method|$N=1$|$N=2$|$N=4$|$N=8$|$N=16$|
>     |:-|:-:|:-:|:-:|:-:|:-:|
>     |MV|50.27|50.27|54.15|56.14|57.16|
>     |$\texttt{dORM}$|50.27|57.30|61.54|63.95|65.38|
>     |$\texttt{dPRM}$|50.27|57.18|61.51|64.10|65.55|
>     |$\texttt{gORM}$|50.27|58.24|63.88|67.82|70.02|
>     |$\texttt{gPRM}$|50.27|55.24|59.06|62.10|64.26|
>     |$\texttt{gPRM}$ (**filtering relaxation**)|50.27|54.93 (**-0.31**)|58.88 (**-0.18**)|62.23 (**+0.13**)|64.82 (**+0.56**)|
>     |Pass@$N$|50.27|61.74|71.56|79.77|86.05|
>
> - As shown in `Table YvuS-4`, the overall performance **does not increase** significantly. This indicates that while overly relaxing *consensus filtering* (i.e., using ORM-style filtering) can reduce the CoT-length distribution shift, it may **fail to remove low-quality data** for $\texttt{gPRM}$ and therefore does not lead to meaningful performance gains.
>
> - We incorporated the results of `Table YvuS-2, 3, and 4` into `Table 3, 5 and 7` in `Appendix G`, respectively.
>
> ---

---

> ### Author Response · Authors · 2025-11-19
> **Response to Reviewer YvuS (4/6)**
>
> ---
>
> > **[Q4]** The dPRM's failure is attributed to its sensitivity to label noise from the single Llama-3.1-70B annotator. Is this a fundamental flaw of dPRM, or an artifact of this specific annotator's noise profile? If regenerating the process labels with a more advanced model, can the dPRM's performance be recovered with higher-quality process labels?
>
> - We again sincerely appreciate your in-depth understanding of our study. We strongly believe that this is **not** a fundamental flaw of $\texttt{dPRM}$, but rather a consequence of **noisy annotation**.
> - Furthermore, wefully agree that higher-quality process labels could **recover** the performance of $\texttt{dPRM}$, especially in multi-domain scenarios.
> - To investigate this, we refine the process labels of MMLU-Pro provided by VersaPRM [1] using one of the state-of-the-art closed models, **Gemini-2.0 Flash** [8], and retrain $\texttt{dPRM}$ and $\texttt{gPRM}$ with the refined labels. Due to a parsing issue, the process labels for **59.96%** of the training data are replaced with labels extracted by Gemini-2.0 Flash.
>
>     **[Table YvuS-5]** Best-of-$N$ results (overall) on MMLU-Pro using Llama-3.1-8B-Instruct with **label refinement**. The number in parentheses denotes the change after label refinement.
>
>     |Method|$N=1$|$N=2$|$N=4$|$N=8$|$N=16$|
>     |:-|:-:|:-:|:-:|:-:|:-:|
>     |MV|50.27|50.27|54.15|56.14|57.16|
>     |$\texttt{dORM}$|50.27|57.30|61.54|63.95|65.38|
>     |$\texttt{dPRM}$|50.27|57.18|61.51|64.10|65.55|
>     |$\texttt{dPRM}$ (**label refinement**)|50.27|56.99 (**–0.19**)|61.41 (**–0.10**)|64.38 (**+0.28**)|66.57 (**+1.02**)|
>     |$\texttt{gORM}$|50.27|58.24|63.88|67.82|70.02|
>     |$\texttt{gPRM}$|50.27|55.24|59.06|62.10|64.26|
>     |$\texttt{gPRM}$ (**label refinement**)|50.27|54.99 (**–0.25**)|58.86 (**–0.20**)|62.10 (**+0.00**)|64.84 (**+0.58**)|
>     |Pass@$N$|50.27|61.74|71.56|79.77|86.05|
>
>
> - `Table YvuS-5` presents the results on MMLU-Pro using the refined process labels. We observe a **meaningful improvement** in certain settings for $\texttt{dPRM}$ (e.g., **+1.02** at $N=16$), whereas **no notable gain** is observed for $\texttt{gPRM}$.

---

> ### Author Response · Authors · 2025-11-19
> **Response to Reviewer YvuS (5/6)**
>
> - We attribute the marginal gains of $\texttt{gPRM}$ to the **imperfections of the training data generator LLM** (i.e., $p\_\mathtt{LLM-j}$, LLM-as-a-judge, or QwQ-32B [7] in our case). Specifically, we argue that such distribution shift can occur **even with cleaner process labels** (e.g., obtained by Gemini-2.0 Flash) due to the inherent **CoT-length distribution** of the multi-domain setting.
>
>     **[Table YvuS-6]** The surviving proportion (%) before and after **label refinement**. The number in parentheses denotes the change after the label refinement.
>     |Method|Overall|Law|Psychology|Chemistry|Biology|Physics|History|Economics|Math|Business|Philosophy|Health|Engineering|CS|Other|
>     |:-|:-:|:-:|:-:|:-:|:-:|:-:|:-:|:-:|:-:|:-:|:-:|:-:|:-:|:-:|:-:|
>     |$\texttt{gORM}$|51.1|51.6|28.3|71.9|42.0|77.0|28.4|50.9|70.3|54.3|48.6|40.5|37.0|70.1|42.9|
>     |$\texttt{gPRM}$|28.0|22.7|22.8|30.1|30.4|34.2|26.0|37.0|30.8|24.7|26.8|27.7|13.5|38.9|31.0|
>     |$\texttt{gPRM}$ (**label refinement**)|44.0 (**+16.0**)|23.6 (**+0.9**)|42.1 (**+19.3**)|33.5 (**+3.4**)|55.4 (**+25.0**)|35.8 (**+1.6**)|49.3 (**+23.3**)|64.9 (**+27.9**)|65.5 (**+34.7**)|56.0 (**+31.3**)|46.6 (**+19.8**)|51.3 (**+23.6**)|13.9 (**+0.4**)|46.9 (**+8.0**)|37.5 (**+6.5**)|
>
>
>     **[Table YvuS-7]** Wasserstein distance before and after **label refinement**. The number in parentheses denotes the change after the label refinement.
>
>     |Method|Overall|Law|Psychology|Chemistry|Biology|Physics|History|Economics|Math|Business|Philosophy|Health|Engineering|CS|Other|
>     |:-|:-:|:-:|:-:|:-:|:-:|:-:|:-:|:-:|:-:|:-:|:-:|:-:|:-:|:-:|:-:|
>     |Train|0.202|0.090|0.203|0.393|0.264|0.628|0.069|0.311|0.167|0.322|0.129|0.105|1.234|0.353|0.093|
>     |$\texttt{gORM}$|0.532|0.089|0.218|1.128|0.506|1.201|0.154|0.564|0.282|0.491|0.545|0.213|3.611|0.338|0.312|
>     |$\texttt{gPRM}$|3.083|1.284|0.742|6.922|2.039|5.952|0.581|1.782|4.655|4.267|1.235|0.979|12.735|3.459|0.927|
>     |$\texttt{gPRM}$ (**label refinement**)|3.265 (**+0.182**)|1.397 (**+0.113**)|0.893 (**+0.151**)|7.183 (**+0.261**)|2.416 (**+0.377**)|6.104 (**+0.152**)|0.752 (**+0.171**)|2.044 (**+0.262**)|4.852 (**+0.197**)|4.494 (**+0.227**)|1.299 (**+0.064**)|1.157 (**+0.178**)|13.058 (**+0.323**)|3.742 (**+0.283**)|1.030 (**+0.103**)|
>
> - `Table YvuS-6` reports the surviving proportion (%) before and after applying process-label refinement, where the proportion **increases significantly (+16.0% overall)**. However, `Table YvuS-7` shows that the Wasserstein distance between the $\texttt{gPRM}$ training dataset and the test split (based on the CoT-length distribution) actually increases slightly (**+0.182 overall**).
>
> - Considering that Gemini-2.0 Flash is stronger than Llama-3.1-70B-Instruct, this result suggests that the **CoT-length distribution shift still persists**, even when using comparatively cleaner labels.
> - We hypothesize that this is **attributed to the imperfection of the training data generator** (i.e., $p_\mathtt{LLM-j}$, LLM-as-a-judge, or QwQ-32B [7] in our case), whose error rate also tends to increase with CoT length.
> - Specifically, even with ground-truth process labels, consensus must be reached between the training data generator and the GT labels. Due to the imperfections of generator, many CoTs are **naturally filtered out**.
> - Intuitively, as the CoT length increases, the likelihood of errors from the data generator also increases, which in turn makes longer CoTs **more likely to be filtered**.
>
> - In summary, we conclude and hypothesize that:
>   (i) there is evidence that the performance of $\texttt{dPRM}$ can be **recovered with high-quality process labels** (or at least with stronger LLM annotators); and
>   (ii) the CoT-length distribution shift of $\texttt{gPRM}$ persists in multi-domain settings, even with improved process labels, which we attribute to the **imperfection of the training data generator** (i.e., $p_\mathtt{LLM-j}$, LLM-as-a-judge, or QwQ-32B [7] in our case).
>
> - We incorprated the results of `Table YvuS-5, 6, and 7` into `Table 6, 3 and 4` in `Appendix G`, respectively.
>
> ---

---

> ### Author Response · Authors · 2025-11-19
> **Response to Reviewer YvuS (6/6)**
>
> ---
>
> ### Reference
>
> [1] Zeng, Thomas, et al. "Versaprm: Multi-domain process reward model via synthetic reasoning data." ICML 2025.
>
> [2] Zhang, Lunjun, et al. "Generative verifiers: Reward modeling as next-token prediction." ICLR 2025.
>
> [3] Khalifa, Muhammad, et al. "Process reward models that think." arXiv 2025.
>
> [4] Shao, Zhihong, et al. "DeepSeekMath: Pushing the limits of mathematical reasoning in open language models." arXiv 2024.
>
> [5] Rein, David, et al. "GPQA: A graduate-level google-proof q&a benchmark." First Conference on Language Modeling. 2024
>
> [6] Zhao, Jian, et al. "GenPRM: Scaling test-time compute of process reward models via generative reasoning." arXiv 2025.
>
> [7] Team, Qwen. "QwQ-32b: Embracing the power of reinforcement learning." Mar. 2025,
>
> [8] Comanici, Gheorghe, et al. "Gemini 2.5: Pushing the frontier with advanced reasoning, multimodality, long context, and next generation agentic capabilities." arXiv 2025.
>
> ---

---

> > ### Author Response · Authors · 2025-11-25
> > **Gentle Reminder**
> >
> > Dear Reviewer YvuS,
> >
> > We sincerely appreciate your time and consideration.
> > We respectfully believe that our response has thoroughly addressed the concerns raised.
> > If you have any remaining concerns or questions, please feel free to contact us and we would be happy to discuss and clarify them.
> >
> > Best,
> >
> > The Authors

---

### Official Review · Reviewer_ueA4 · 2025-11-01

**Soundness:** 3
**Presentation:** 3
**Contribution:** 2
**Rating:** 4
**Confidence:** 4

**Summary:**

This paper evaluates several variants of RMs on math and general domains with controlled experiments. By unifying the setups of several previous works, the authors found out that math and multi domain settings might lead to different trends among those RM variants. Then intuitive understandings of these differences were proposed, namely error accumulation along CoT trajectories, label noise, and length distribution shift. Results suggest that PRM might not be always better than ORMs especially in multidomain setup, and provide practical guidelines.

**Strengths:**

* The paper presents controlled experiments on RM variants to address the debate around them on math and multi domain setup.
* The paper is well written, with clear narratives and good figures. The main message is clearly conveyed with experiments, followed by in-depth analysis.

**Weaknesses:**

* For math, process reward labels come from PRM 800k, which is typically believed to have good quality. For MMLU-pro, they are fully synthetic, from Llama 3 70b. Potentially the degradation of label quality on MMLU-pro can lead to the result that PRM underperforms ORM. In contrast, the ORM labels of both PRM 800k and MMLU pro should be good. The difference in label quality itself might undermine the validity of the conclusions
* Although MMLU pro is different from math, they are still both tasks with verifiable rewards. It is still unclear how ORM and PRM compare in a more open ended setup.
* Weak theoretical results. It is well known that PRM error accumulates as the CoT length grows. In this sense, the results are of almost no significance. Besides, this point alone does not establish the advantage of ORM over PRM.

**Questions:**

* The paper shows gPRM is robust to label noise in the math domain but fails in the multi-domain setting due to a distribution shift. Is this shift purely an artifact of consensus filtering on noisy labels? In other words, would gPRM still fail in a multi-domain setting even if it were trained on perfectly clean, human-annotated process labels (like PRM 800k)?
* The paper states CoTs account for 15.3% of ProcessBench. What is the prevalence of "aha" CoTs in MMLU-Pro? Is it significantly higher, and could this (in addition to CoT length) be a primary factor of the PRM performance collapse?

---

> ### Author Response · Authors · 2025-11-19
> **Response to Reviewer ueA4 (1/7)**
>
> We sincerely appreciate your time and constructive comments, which improve our paper. We address your concerns as follows.
>
> ---
>
> >**[Q1]** For math, process reward labels come from PRM 800k, which is typically believed to have good quality. For MMLU-Pro, they are fully synthetic, from Llama-3-70B. Potentially, the degradation of label quality on MMLU-Pro can lead to the result that PRM underperforms ORM. In contrast, the ORM labels of both PRM 800k and MMLU-Pro should be good. The difference in label quality itself might undermine the validity of the conclusions.
>
> - We sincerely appreciate the reviewer for this observation and acknowledge that our presentation of the contribution of this study may lack clarity. However, we respectfully believe that this concern stems from a misunderstanding of our **key message** and the **practical guidelines** in `§5`.
> - The issue raised is actually the **central message** of our study. We aim to explain the failure of PRMs from multiple perspectives, among which **label noise** is one contributing factor.
> - Prior work [1, 2, 3] has shown that $\texttt{dPRM}$ outperforms $\texttt{dORM}$, and that generative variants perform better than discriminative variants in the **math domain**, which our experiments also confirm (Figs. 3 and 4 in `§3.2`).
> - However, we observe that this trend does **not hold** in **multi-domain settings** (`Figs. 5, 6 and 7` in `§3.2`).
> - In `§4`, we attribute the degradation of PRMs to:
>   (i) the increasing **risk of PRMs with longer CoT lengths** (`§4.1`),
>   (ii) **label noise** in PRM training data (`§4.2`).
> - Specifically, we assume that the label noise may be introduced by the **automatic annotation of large language models (LLMs)**. As discussed in `L420-425`, the deployment of LLMs for annotation is due to that it is more hard to obtain human-annotated process labels in the multi-domaing settings (e.g., law and medicine) in compared to the math domain.
> - Under these potential risks and based on our analysis in `§4`, we provide **practical guidelines** in `§5`, where we explicitly recommend using $\texttt{dPRM}$ when the labels are assumed to be clean (`L465`).
> - Therefore, rather than undermining our conclusions, this observation is itself our **main contribution**, as it identifies the risk of PRMs in the multi-domain settings and provides practical recommendations for their reliable use.
> - We will clarify this contribution and the accompanying guidelines in the final version of the paper.
>
> ---

---

> ### Author Response · Authors · 2025-11-19
> **Response to Reviewer ueA4 (2/7)**
>
> ---
>
> >**[Q2]** Although MMLU pro is different from math, they are still both tasks with verifiable rewards. It is still unclear how ORM and PRM compare in a more open ended setup.
>
> - We sincerely appreciate the reviewer for raising this constructive comment. We repectfully agree that extending our analysis to open-ended tasks is an important future direction. However, we kindly note that our focus on verifiable tasks (e.g., math and multiple-choice QA) is a **deliberate methodological choice** for the following reasons.
>
> - Our primary goal is to analyze the key properties of different reward models (process vs. outcome, generative vs. discriminative). To do this rigorously, we require an **objective ground truth**. Following prior work on reasoning and test-time scaling [2, 3, 4], we therefore focus on verifiable task settings, which allow us to isolate architectural differences among the four models and evaluate their core capabilities without the confounding factors inherent in subjective evaluation.
>
> - By contrast, the evaluation of open-ended tasks lacks objective ground truth. A verification score would need to be compared against **human preference judgments** or **another LLM-as-a-judge**. This would shift the focus of our study from understanding reward model behavior to benchmarking alignment with a **potentially biased judge**, undermining our objective and reproducible evaluation framework. As the community has increasingly shifted toward objective evaluation on verifiable tasks (e.g., GRPO [5]), we believe our choice aligns with this trend.
>
> - Therefore, we believe that our **current setup provides the most reliable framework** for analyzing reward model designs under a standardized, objective evaluation protocol.
>
> - Instead, during this rebuttal, we conduct additional experiments on **a more challenging multiple-choice QA dataset**, GPQA-diamond [6], which spans biology, chemistry, and physics. Its difficulty is empirically supported by the majority-voting performance of Llama-3.1-8B-Instruct with $N=16$: **57.16 on MMLU-Pro** versus **34.34 on GPQA-diamond** in our experiments.
>
> - As expected, we observe that $\texttt{gORM}$ **delivers consistent and strong gains** over all other rewardmodel variants (e.g., +4.85/+9.38/+12.32/+13.64 over the second-best for $N=2/4/8/16$ using Llama-3.1-8B-Instruct). Please see our response to **[Q4]** of reviewer WbK9 for more details.
>
> - Despite the scope of our analysis, we would like to discuss our **hypothesis** regarding open-ended tasks: our central conclusion that the generative outcome reward model ($\texttt{gORM}$) performs best for long and complex reasoning chains is likely to **generalize to open-ended tasks**.
>
> - A major limitation we identify in PRMs is that they cannot adequately reward "aha moments" or self-correction. We believe that this issue is **not unique to verifiable tasks**. For instance, a long narrative may begin with weaker opening paragraphs but develop into a compelling conclusion. A PRM-based preference model, trained to myopically penalize each locally suboptimal segment, would likely fail to capture the emergent quality of the story, whereas a $\texttt{gORM}$-based holistic model could better evaluate the piece as a whole.
>
> - We will add a detailed discussion of this limitation and our hypothesis regarding the generalization of our conclusion to open-ended tasks in the final version of the paper. We also incorporated the results on GPQA-diamond into `Figs. 41 and 42` in `Appendix G`.
>
>
> ---

---

> ### Author Response · Authors · 2025-11-19
> **Response to Reviewer ueA4 (3/7)**
>
> ---
>
> > **[Q3]** Weak theoretical results. It is well known that PRM error accumulates as the CoT length grows. In this sense, the results are of almost no significance. Besides, this point alone does not establish the advantage of ORM over PRM.
>
> - We repectfully agree that the theoretical results may appear intuitive and well-known. However, we would like to emphasize that our theoretical analysis is **not intended as a standalone contribution** of this study.
> - The central message of our work is to explain the **failure of PRMs in multi-domain settings** from multiple perspectives, including: (i) the increasing **risk of PRMs with longer CoT lengths** (`§4.1`), and (ii) **label noise** in PRM training data (`§4.2`).
> - More importantly, we connect the phenomenon of increasing PRM risk with CoT length to the emergence of **"aha" moments**, which are widely observed in strong reasoning models [7].
> - We believe this interpretation has **not been discussed in prior work**, yet it is both intuitive and well-supported by our empirical observations in `Figs. 9 and 10` of `§4.1`, where we find that PRMs particularly struggle with "aha" CoTs compared to ORMs.
> - Based on the observations, we **formalize the qualitative insight** by deriving **error bounds for PRMs**, showing that these bounds **increase with CoT length** (`Theorems 4.1, 4.2, 4.3, and A.4`), which is also empirically validated in `Fig. 12`.
> - In summary, while we acknowledge that the theoretical results themselves may not be fundamentally surprising, the **integrated analysis**, e.g., combining theoretical, empirical, and interpretative perspectives to explain **why PRMs fail in multi-domain settings**, is, to our knowledge, **not presented before** and constitutes a **key contribution** of our study.
>
> ---

---

> ### Author Response · Authors · 2025-11-19
> **Response to Reviewer ueA4 (4/7)**
>
> ---
>
> > **[Q4-1]** The paper shows gPRM is robust to label noise in the math domain but fails in the multi-domain setting due to a distribution shift. Is this shift purely an artifact of consensus filtering on noisy labels?
>
> - We sincerely appreciate your in-depth understanding of our work. We believe that the observed distribution shift arises not only from consensus filtering on **noisy labels**, but also from the **imperfections of the training data generator LLM** (i.e., $p_\mathtt{LLM-j}$, LLM-as-a-judge, or QwQ-32B [8] in our case).
> - Specifically, we respectfully argue that such distribution shift can occur **even with clean (ground-truth) process labels** due to the inherent **CoT-length distribution** of the multi-domain setting.
> - As shown in `Fig. 13`, the outcome-verification F1 score of $\texttt{dPRM}$ is **highly sensitive** to label noise. However, the performance gap between $\texttt{dORM}$ and $\texttt{dPRM}$ remains relatively small in multi-domain settings (e.g., `Figs. 5–7`), suggesting that while label noise exists, its **severity may not be dominant**.
> - Nevertheless, the CoT-length distribution in multi-domain tasks is much more **skewed** than in math, particularly for **“aha” CoTs** (`Fig. 11`).
> - As noted in `L447–449`, such skewness makes achieving consensus more difficult. We believe this difficulty would persist **even with ground-truth labels**.
> - Specifically, even with GT process labels, consensus must be reached between the **training data generator LLM** (i.e., $p_\mathtt{LLM-j}$, LLM-as-a-judge, or QwQ-32B [8] in our case) and the GT labels, as demonstrated in `L189-192`. Due to imperfections of the data generator, many CoTs are **naturally filtered out**.
> - Intuitively, as CoT length increases, the likelihood of error from the data generator also increases, which in turn makes longer CoTs **more likely to be filtered**.
> - To illustrate this, we report the CoT-length distribution of the surviving samples:
>
>   **[Table ueA4-1] CoT-length distribution** for $\texttt{gORM}$ and $\texttt{gPRM}$.
>
>   |Dataset|Method|Min|Max|Mean|Std|Median|0–25% mean|25–50% mean|50–75% mean|75–100% mean|
>   |:-:|:-:|:-:|:-:|:-:|:-:|:-:|:-:|:-:|:-:|:-:|
>   |PRM800K|$\texttt{gORM}$|1|297|10.40|5.56|9.0|5.54|7.94|10.66|17.48|
>   |PRM800K|$\texttt{gPRM}$|1|52|8.51|5.34|8.0|2.74|5.94|9.56|15.80|
>   |MMLU-Pro|$\texttt{gORM}$|2|152|8.22|7.74|6.0|3.41|5.04|7.22|17.22|
>   |MMLU-Pro|$\texttt{gPRM}$|1|59|4.68|2.77|4.0|1.91|3.61|4.98|8.21|
>
> - As shown in `Table ueA4-1`, the mean and median CoT lengths for gORM remain **consistently higher** than those of gPRM, even in PRM800K, which is assumed to contain ground-truth process labels. By dividing the length distribution into 25% quantiles, we observe a more distinct difference in CoT lengths between $\texttt{gORM}$ and $\texttt{gPRM}$.

---

> ### Author Response · Authors · 2025-11-19
> **Response to Reviewer ueA4 (5/7)**
>
> - To empirically support our hypothesis further, we refine the process labels using a stronger model, **Gemini-2.0 Flash** [9]. Due to a parsing issue, the process labels for **59.96%** of the training data were replaced with labels extracted by Gemini-2.0 Flash.
>
>
>     **[Table ueA4-2]** The surviving proportion (%) before and after **label refinement**. The number in parentheses denotes the change after the label refinement.
>
>     |Method|Overall|Law|Psychology|Chemistry|Biology|Physics|History|Economics|Math|Business|Philosophy|Health|Engineering|CS|Other|
>     |:-|:-:|:-:|:-:|:-:|:-:|:-:|:-:|:-:|:-:|:-:|:-:|:-:|:-:|:-:|:-:|
>     |$\texttt{gORM}$|51.1|51.6|28.3|71.9|42.0|77.0|28.4|50.9|70.3|54.3|48.6|40.5|37.0|70.1|42.9|
>     |$\texttt{gPRM}$|28.0|22.7|22.8|30.1|30.4|34.2|26.0|37.0|30.8|24.7|26.8|27.7|13.5|38.9|31.0|
>     |$\texttt{gPRM}$ (**label refinement**)|44.0 (**+16.0**)|23.6 (**+0.9**)|42.1 (**+19.3**)|33.5 (**+3.4**)|55.4 (**+25.0**)|35.8 (**+1.6**)|49.3 (**+23.3**)|64.9 (**+27.9**)|65.5 (**+34.7**)|56.0 (**+31.3**)|46.6 (**+19.8**)|51.3 (**+23.6**)|13.9 (**+0.4**)|46.9 (**+8.0**)|37.5 (**+6.5**)|
>
>
>     **[Table ueA4-3]** Wasserstein distance before and after **label refinement**. The number in parentheses denotes the change after the label refinement.
>
>     |Method|Overall|Law|Psychology|Chemistry|Biology|Physics|History|Economics|Math|Business|Philosophy|Health|Engineering|CS|Other|
>     |:-|:-:|:-:|:-:|:-:|:-:|:-:|:-:|:-:|:-:|:-:|:-:|:-:|:-:|:-:|:-:|
>     |Train|0.202|0.090|0.203|0.393|0.264|0.628|0.069|0.311|0.167|0.322|0.129|0.105|1.234|0.353|0.093|
>     |$\texttt{gORM}$|0.532|0.089|0.218|1.128|0.506|1.201|0.154|0.564|0.282|0.491|0.545|0.213|3.611|0.338|0.312|
>     |$\texttt{gPRM}$|3.083|1.284|0.742|6.922|2.039|5.952|0.581|1.782|4.655|4.267|1.235|0.979|12.735|3.459|0.927|
>     |$\texttt{gPRM}$ (**label refinement**)|3.265 (**+0.182**)|1.397 (**+0.113**)|0.893 (**+0.151**)|7.183 (**+0.261**)|2.416 (**+0.377**)|6.104 (**+0.152**)|0.752 (**+0.171**)|2.044 (**+0.262**)|4.852 (**+0.197**)|4.494 (**+0.227**)|1.299 (**+0.064**)|1.157 (**+0.178**)|13.058 (**+0.323**)|3.742 (**+0.283**)|1.030 (**+0.103**)|
>
> - `Table ueA4-2` reports the surviving proportion (%) before and after applying process-label refinement, where the proportion **increases significantly (+16.0% overall)**. However, `Table ueA4-3` shows that the Wasserstein distance between the $\texttt{gPRM}$ training dataset and the test split (based on the CoT-length distribution) actually increases slightly **(+0.182 overall)**.
>
> - Considering that Gemini-2.0 Flash is stronger than Llama-3.1-70B-Instruct, this suggests that the **CoT-length distribution shift still persists**, even when using comparatively clean labels.
>
> - In summary, we conclude that the CoT-length distribution shift observed in $\texttt{gPRM}$ under multi-domain settings is **not purely due to label noise**, but is also influenced by the **imperfection of the training data generator** (i.e., $p_\mathtt{LLM-j}$, LLM-as-a-judge, or QwQ-32B [8] in our case), whose error tends to increase with CoT length.
> - We included the above discussion on the imperfection of the training data generator in `L446–449`, and incorporated the results of `Table ueA4-2` and `Table ueA4-3` into `Table 3` and `Table 5` in `Appendix G`.
>
> ---

---

> ### Author Response · Authors · 2025-11-19
> **Response to Reviewer ueA4 (6/7)**
>
> ---
>
> > **[Q4-2]** In other words, would gPRM still fail in a multi-domain setting even if it were trained on perfectly clean, human-annotated process labels (like PRM 800K)?
>
> - First, we cannot conclude this with certainty because we do not have access to truly clean, human-annotated process labels. Our study evaluates the problem under the **current realistic condition** in which process labels are noisy.
> - Nevertheless, we **tentatively** suggest that even with perfectly clean process labels, $\texttt{gORM}$ would likely remain a **stronger alternative** to $\texttt{gPRM}$, although this does not necessarily imply that $\texttt{gPRM}$ would fail.
> - This hypothesis is motivated by two factors:
>
>   (i) the **imperfection of the training data generator** (i.e., $p\_\mathtt{LLM-j}$, LLM-as-a-judge, or QwQ-32B [8] in our case), which can still induce CoT-length distribution shift (as discussed in our response to **[Q4-1]**); and
>
>   (ii) the increasing risk associated with $\texttt{gPRM}$ for longer CoT lengths (`§4.1`).
>
> - Our suggestion is supported empirically by `Figs. 3 and 4`, where the performance of $\texttt{gORM}$ is comparable to that of $\texttt{gPRM}$ in the math domain, which we assume to have relatively clean process supervision.
> - To further examine the potential effect of cleaner process labels in multi-domain settings, we additionally train $\texttt{dPRM}$ and $\texttt{gPRM}$ with 14B backbones using process labels refined by **Gemini-2.0 Flash** [9], as used in `Table ueA4-2` and `Table ueA4-3` in our response to **[Q4-1]**.
>
>     **[Table ueA4-4]** Best-of-$N$ results (overall) on MMLU-Pro using Llama-3.1-8B-Instruct with **label refinement**. The number in parentheses denotes the change after label refinement.
>
>     |Method|$N=1$|$N=2$|$N=4$|$N=8$|$N=16$|
>     |:-|:-:|:-:|:-:|:-:|:-:|
>     |MV|50.27|50.27|54.15|56.14|57.16|
>     |$\texttt{dORM}$|50.27|57.30|61.54|63.95|65.38|
>     |$\texttt{dPRM}$|50.27|57.18|61.51|64.10|65.55|
>     |$\texttt{dPRM}$ (**label refinement**)|50.27|56.99 (**–0.19**)|61.41 (**–0.10**)|64.38 (**+0.28**)|66.57 (**+1.02**)|
>     |$\texttt{gORM}$|50.27|58.24|63.88|67.82|70.02|
>     |$\texttt{gPRM}$|50.27|55.24|59.06|62.10|64.26|
>     |$\texttt{gPRM}$ (**label refinement**)|50.27|54.99 (**–0.25**)|58.86 (**–0.20**)|62.10 (**+0.00**)|64.84 (**+0.58**)|
>     |Pass@$N$|50.27|61.74|71.56|79.77|86.05|
>
> - `Table ueA4-4` presents the results on MMLU-Pro using the refined process labels. We observe a meaningful improvement in certain settings for $\texttt{dPRM}$ (e.g., **+1.02** at $N=16$), whereas **no notable gain** is observed for $\texttt{gPRM}$.
> - However, we believe that PRM variants still offer several advantages over ORM variants, for instance, in leveraging **process rewards for LLM post-training** [10] (which remains out of scope for this study and is therefore not discussed in our paper).
> - We incorporated the results of `Table ueA4-4` into `Table 6` in `Appendix G`.
>
> ---

---

> ### Author Response · Authors · 2025-11-19
> **Response to Reviewer ueA4 (7/7)**
>
> ---
>
> > **[Q5]** The paper states CoTs account for 15.3% of ProcessBench. What is the prevalence of "aha" CoTs in MMLU-Pro? Is it significantly higher, and could this (in addition to CoT length) be a primary factor of the PRM performance collapse?
>
> - We thank the reviewer for raising this insightful point. Before addressing it directly, we would like to briefly recap the notion of "aha" CoTs.
> - To the best of our knowledge, there is **no widely accepted rigorous definition** of an "aha moment." It is usually described intuitively as a case where LLMs **recover from its own reasoning flaw** during the course of solving a problem [7].
> - In this study, we formalize "aha" moments (or CoTs) using **process labels** $z_{1:T}$: an "aha" CoT consists of reasoning steps $r_{1:T}$ where at least one intermediate step is incorrect ($\exists t \in \\{1, \ldots, T-1\\}: z_t = 0$), but the final answer is correct ($y = 1$), as elaborated in `L359–361`.
> - Therefore, identifying "aha" CoTs under our formulation requires access to **ground-truth (GT) process labels** $z_{1:T}$. In the ProcessBench dataset, such process labels are carefully curated [11] and assumed to be GT. In `§4.1`, we select the subset of CoTs that satisfy the above condition and found that approximately 15.3% of the CoTs in ProcessBench are "aha" CoTs.
> - For the MMLU-Pro dataset, VersaPRM [4] provides process labels **only for the training split** (used for the visualization in `Fig. 11`), as these labels are required for training PRMs. The proportion of "aha" CoTs in this training split is approximately **10.6%**.
> - However, we note that these labels **cannot be assumed to be GT**, since they are annotated by Llama-3.1-70B-Instruct, which is inherently imperfect.
> - In the **test split** of MMLU-Pro provided by VersaPRM [4], i.e., CoTs generated by Llama-3.1-8B-Instruct to solve test questions, **no process labels are available**, not even the potentially noisy ones.
> - Therefore, due to (i) the **potential label noise** and (ii) the **absence of process labels in the test split**, we only conducted our "aha" CoT analysis on ProcessBench.
> - Under these circumstances, we believe it is **not possible to conclude** that "aha" CoTs are the *primary factor* behind the degradation of PRM performance, since we cannot determine the **exact prevalence of "aha" CoTs** in multi-domain settings.
> - Rather, we regard them as **one of several potential risk factors** for PRMs in multi-domain settings, as discussed in `§4.1`.
> - Accordingly, we provide a practical guideline recommending the use of **ORM variants** when frequent error recoveries are expected (`L466`).
> - We will further elaborate and clarify the above connection between our observations in `§4.1` and the practical guidelines provided in `§5` in the final version.
>
> ---
>
> ### Reference
>
> [1] Lightman, Hunter, et al. "Let's verify step by step." The Twelfth International Conference on Learning Representations. 2023.
>
> [2] Zhang, Lunjun, et al. "Generative verifiers: Reward modeling as next-token prediction." ICLR 2025.
>
> [3] Khalifa, Muhammad, et al. "Process reward models that think." arXiv 2025.
>
> [4] Zeng, Thomas, et al. "Versaprm: Multi-domain process reward model via synthetic reasoning data." ICML 2025.
>
> [5] Shao, Zhihong, et al. "Deepseekmath: Pushing the limits of mathematical reasoning in open language models." arXiv 2024.
>
> [6] Rein, David, et al. "Gpqa: A graduate-level google-proof q&a benchmark." First Conference on Language Modeling. 2024
>
> [7] Guo, Daya, et al. "Deepseek-r1: Incentivizing reasoning capability in llms via reinforcement learning." arXiv 2025.
>
> [8] Team, Qwen. "Qwq-32b: Embracing the power of reinforcement learning." Mar. 2025,
>
> [9] Comanici, Gheorghe, et al. "Gemini 2.5: Pushing the frontier with advanced reasoning, multimodality, long context, and next generation agentic capabilities." arXiv 2025.
>
> [10] Yin, Zhangyue, et al. "Dynamic and generalizable process reward modeling." ACL. 2025.
>
> [11] Zheng, Chujie, et al. "Processbench: Identifying process errors in mathematical reasoning." ACL 2025.
>
> ---

---

> ### Author Response · Authors · 2025-11-25
> **Gentle Reminder**
>
> Dear Reviewer ueA4,
>
> We sincerely appreciate your time and consideration.
> We respectfully believe that our response has thoroughly addressed the concerns raised.
> If you have any remaining concerns or questions, please feel free to contact us and we would be happy to discuss and clarify them.
>
> Best,
>
> The Authors

---

### Official Review · Reviewer_TwoX · 2025-11-03

**Soundness:** 3
**Presentation:** 4
**Contribution:** 3
**Rating:** 8
**Confidence:** 2

**Summary:**

This paper provides a unified evaluation of four reward model variants for test-time scaling—discriminative and generative outcome reward models (dORM, gORM) and process reward models (dPRM, gPRM)—across 14 domains. While prior work favored PRMs (step-level supervision) over ORMs (final-answer supervision), the authors find that this assumption breaks down in multi-domain reasoning. Specifically, gORM consistently outperforms all others, dORM ≈ dPRM, and gPRM underperforms despite prior success in math. The paper attributes these results to compounded stepwise errors and label noise in long or noisy CoTs. Theoretical analysis establishes that PRM log-error grows linearly with reasoning length, while ORM error remains bounded. Empirical results (e.g., Figs. 5–8, 12, 15) confirm this, and the authors provide practical guidelines for model selection.

**Strengths:**

- This paper is very well written with lots of careful analysis. I appreciate all the figures and they clearly conveys the results.
- The paper compares four reward model variants across 14 diverse domains (law, biology, philosophy, etc.), using multiple backbones and datasets, and the study the quite diverse and comprehensive.

**Weaknesses:**

- Limited task diversity in form: All benchmarks are multiple-choice or verifiable; generalization to open-ended reasoning or generation remains untested.

**Questions:**

n/a

---

> ### Author Response · Authors · 2025-11-19
> **Response to Reviewer TwoX**
>
> We sincerely appreciate your time and constructive comments, which improve our paper. We address your concerns as follows.
>
> ---
>
> > **[Q1]** Limited task diversity in form: All benchmarks are multiple-choice or verifiable; generalization to open-ended reasoning or generation remains untested.
>
> - We sincerely appreciate the reviewer for raising this constructive comment. We repectfully agree that extending our analysis to open-ended tasks is an important future direction. However, we kindly note that our focus on verifiable tasks (e.g., math and multiple-choice QA) is a **deliberate methodological choice** for the following reasons.
>
> - Our primary goal is to analyze the key properties of different reward models (process vs. outcome, generative vs. discriminative). To do this rigorously, we require an **objective ground truth**. Following prior work on reasoning and test-time scaling [1, 2, 3], we therefore focus on verifiable task settings, which allow us to isolate architectural differences among the four models and evaluate their core capabilities without the confounding factors inherent in subjective evaluation.
>
> - By contrast, the evaluation of open-ended tasks lacks objective ground truth. A verification score would need to be compared against **human preference judgments** or **another LLM-as-a-judge**. This would shift the focus of our study from understanding reward model behavior to benchmarking alignment with a **potentially biased judge**, undermining our objective and reproducible evaluation framework. As the community has increasingly shifted toward objective evaluation on verifiable tasks (e.g., GRPO [4]), we believe our choice aligns with this trend.
>
> - Therefore, we believe that our **current setup provides the most reliable framework** for analyzing reward model designs under a standardized, objective evaluation protocol.
>
> - Instead, during this rebuttal, we conduct additional experiments on **a more challenging multiple-choice QA dataset**, GPQA-diamond [5], which spans biology, chemistry, and physics. Its difficulty is empirically supported by the majority-voting performance of Llama-3.1-8B-Instruct with $N=16$: **57.16 on MMLU-Pro** versus **34.34 on GPQA-diamond** in our experiments.
>
> - As expected, we observe that $\texttt{gORM}$ **delivers consistent and strong gains** over all other rewardmodel variants (e.g., +4.85/+9.38/+12.32/+13.64 over the second-best for $N=2/4/8/16$ using Llama-3.1-8B-Instruct). Please see our response to **[Q4]** of reviewer WbK9 for more details.
>
> - Despite the scope of our analysis, we would like to discuss our **hypothesis** regarding open-ended tasks: our central conclusion that the generative outcome reward model ($\texttt{gORM}$) performs best for long and complex reasoning chains is likely to **generalize to open-ended tasks**.
>
> - A major limitation we identify in PRMs is that they cannot adequately reward "aha moments" or self-correction. We believe that this issue is **not unique to verifiable tasks**. For instance, a long narrative may begin with weaker opening paragraphs but develop into a compelling conclusion. A PRM-based preference model, trained to myopically penalize each locally suboptimal segment, would likely fail to capture the emergent quality of the story, whereas a $\texttt{gORM}$-based holistic model could better evaluate the piece as a whole.
>
> - We will add a detailed discussion of this limitation and our hypothesis regarding the generalization of our conclusion to open-ended tasks in the final version of the paper. We also incorporated the results on GPQA-diamond into `Figs. 41 and 42` in `Appendix G`.
>
> ---
>
> ### Reference
>
> [1] Zeng, Thomas, et al. "VersaPRM: Multi-domain process reward model via synthetic reasoning data." ICML 2025.
>
> [2] Zhang, Lunjun, et al. "Generative verifiers: Reward modeling as next-token prediction." ICLR 2025.
>
> [3] Khalifa, Muhammad, et al. "Process reward models that think." arXiv 2025.
>
> [4] Shao, Zhihong, et al. "DeepSeekMath: Pushing the limits of mathematical reasoning in open language models." arXiv 2024.
>
> [5] Rein, David, et al. "GPQA: A graduate-level google-proof q&a benchmark." First Conference on Language Modeling. 2024
>
>
> ---

---

> ### Author Response · Authors · 2025-11-25
> **Gentle Reminder**
>
> Dear Reviewer TwoX,
>
> We sincerely appreciate your time and consideration.
> We respectfully believe that our response has thoroughly addressed the concerns raised.
> If you have any remaining concerns or questions, please feel free to contact us and we would be happy to discuss and clarify them.
>
> Best,
>
> The Authors

---

### Author Response · Authors · 2025-11-19
**General Response**

We sincerely thank all reviewers for their time and constructive feedback, which has significantly improved our paper.

We are especially grateful that the reviewers recognize several strengths of our work, including the **first unified evaluation of four RM variants across diverse domains** (TwoX, ueA4, YvuS), our **clear presentation** (TwoX, ueA4, YvuS, WbK9), the **practical value for selecting external verifiers** (TwoX, WbK9), and our **theoretical and empirical analysis of PRM failure modes** (TwoX, ueA4, YvuS).

We believe that we have successfully addressed all major concerns in this rebuttal. Below, we summarize **the changes** that have been included in the revised version (shown in blue):

---

- **`L446–449`** in **`§4.2`**: in response to reviewer **ueA4**, we expanded our discussion on the **limitations of the training data generator** (i.e., $p\_\mathtt{LLM-j}$, LLM-as-a-judge, or QwQ-32B [1] in our case) and how such **imperfections contribute to distribution shift**.

- **`Table 3`** in **`Appendix G`**: in response to reviewers **ueA4** and **YvuS**, we added the Wasserstein distance of $\texttt{gPRM}$ in the multi-domain setting with (i) **label refinement** using Gemini-2.0 Flash [2] and (ii) **relaxation** of the *consensus filtering*.

- **`Table 5`** in **`Appendix G`**: in response to reviewers **ueA4** and **YvuS**, we included the **surviving proportion** (%) after *consensus filtering* of verification CoTs for $\texttt{gORM}$, $\texttt{gPRM}$, $\texttt{gPRM}$ with label refinement, and $\texttt{gPRM}$ with relaxed filtering.

- **`Table 6`** in **`Appendix G`**: in response to reviewers **ueA4** and **YvuS**, we added best-of-$N$ results on MMLU-Pro using Llama-3.1-8B-Instruct with **label refinement**.

- **`Table 7`** in **`Appendix G`**: in response to reviewer **YvuS**, we added best-of-$N$ results on MMLU-Pro using Llama-3.1-8B-Instruct with **relaxed filtering**.

- **`Table 8`** in **`Appendix G`**: in response to reviewer **YvuS**, we added best-of-$N$ results on MMLU-Pro using Llama-3.1-8B-Instruct while varying the **learning rate** and **LoRA rank $r$** for $\texttt{dPRM}$ and $\texttt{gPRM}$.

- **`Figs. 41 and 42`** in **`Appendix G`**: In response to reviewer **WbK9**, we added results on a **more challenging multiple-choice QA dataset**, GPQA-diamond [3] (biology, chemistry, and physics), using five different LLMs with $N=16$.

- **`Final version`**: in response to **all reviewers**, we will include a **detailed discussion of the limitations on open-ended generation** and **our hypothesis regarding generalization** of our study to such settings.

- **`Final version`**: in response to reviewer **ueA4**, we will further clarify the connection between the observations in `§4.1` and the practical guidelines in `§5`.


---

### Reference

[1] Team, Qwen. "Qwq-32b: Embracing the power of reinforcement learning." Mar. 2025,

[2] Comanici, Gheorghe, et al. "Gemini 2.5: Pushing the frontier with advanced reasoning, multimodality, long context, and next generation agentic capabilities." arXiv 2025.

[3] Rein, David, et al. "GPQA: A graduate-level google-proof q&a benchmark." First Conference on Language Modeling. 2024

---

---

### Comment · Area_Chair_R6B8 · 2025-11-26

Dear Reviewers,

Thank you once again for your service to ICLR 2026. Now that the authors have submitted their rebuttal, I kindly ask you to take the following steps (if you have not done so already):

Read the authors’ response and other reviews.
Consider whether the rebuttal and additional comments affect your assessment of the paper.
Engage in interactive discussion with the authors -- Note the Author-Reviewer-AC discussion period ends on 12/3 9 PM UTC. You are recommended to keep active before that deadline. If you have more concerns/questions (e.g., requesting clarifications, new results), it is recommended to post your request asap, so that the authors have enough time to address them.
The current reviews for this paper are mixed (scores: 6/6/8/4). Your further contributions are essential for forming a well-informed final decision.

I am happy to join and support the discussions between you and the authors. Please feel free to share your thoughts and participate actively in the discussion. Thanks!

Best regards,

AC

---

> ### Author Response · Authors · 2025-11-28
> **Gentle Reminder**
>
> Dear Reviewers,
>
> Thank you for your thoughtful evaluations and the constructive discussion so far. We also appreciate the AC’s guidance and the reminder regarding the upcoming discussion deadline.
> As we approach the end of the Author-Reviewer-AC discussion period, we kindly welcome any further questions, clarifications, or concerns you may have.
>
> Thank you again for your consideration and for participating actively in this process.
>
> Best regards,
>
> The Authors

---

### Author Response · Authors · 2025-11-29
**Letter to AC and Reviewers (2/2)**

---

### Reviewer YvuS

> **[Q1]** The reviewer raised a concern about **limited task diversity**.

- We clarify the **scope of this study**, which follows prior work on reasoning and test-time scaling [1, 2, 3], and we provide our hypothesis that our conclusions **can generalize to open-ended generation tasks**.
- We also conduct additional experiments on **GPQA-diamond** [4] and observe a consistent conclusion that **generative outcome verification delivers robust gains**.

> **[Q2]** The reviewer raised a concern about the **limited hyperparameter search for PRMs**.

- We conduct additional experiments by **varying the learning rate and LoRA rank $r$ of PRMs**, and observe **no improvement**, which is consistent with our current conclusions.


> **[Q3]** The reviewer raised a concern about **the data collection pipeline of $\texttt{gPRM}$**.

- We conduct additional experiments by **relaxing the consensus filtering** of $\texttt{gPRM}$ and observe no improvement, which is consistent with our current conclusions.

> **[Q4]** The reviewer raised a concern about the **performance of PRM variants with improved label quality**.

- We conduct additional experiments by **refining labels with Gemini-2.0 Flash** [5], and observe that the performance of $\texttt{dPRM}$ improves slightly while that of $\texttt{gPRM}$ does not. This is consistent with our current conclusions.
- We attribute this to **imperfections in the LLM used to generate the training data**.

---

### Reviewer WbK9

> **[Rating]** The reviewer raised the **contribution score from 2 to 4**.

- The reviewer appreciated our comprehensive additional experiments on GPQA-diamond [4] and intended to update the overall rating from 6 to 7. However, since the system only allows ratings of 6 or 8, the reviewer instead raised the contribution score from 2 to 4.

> **[Q1]** The reviewer raised a concern about **obviousness of our study**.

- We clarify that our study (i) identifies the **potential risks of PRMs in multi-domain settings** and (ii) connects **empirical observations with a concrete analysis framework**, which have not been systematically investigated in prior work.

> **[Q2]** The reviewer raised a concern about the **training-time usage** of reward model variants.

- We clarify that the **scope of this study** follows the verifiable-reward setting [1, 2, 3], which does not motivate **training-time usage**.


> **[Q3]** The reviewer raised a conern about **limited evaluation on MMLU-Pro**.

- We conduct additional experiments on **GPQA-diamond** [4] and observe a consistent conclusion that **generative outcome verification delivers robust gains**.


> **[Q4]** The reviewer raised a concern about **harder tasks than MMLU-Pro**.

- We clarify that GPQA-diamond is indeed a **harder task than MMLU-Pro**, which is empirically supported by the **performance of majority voting**.

---

### Reference

[1] Zeng, Thomas, et al. "VersaPRM: Multi-domain process reward model via synthetic reasoning data." ICML 2025.

[2] Zhang, Lunjun, et al. "Generative verifiers: Reward modeling as next-token prediction." ICLR 2025.

[3] Khalifa, Muhammad, et al. "Process reward models that think." arXiv 2025.

[4] Rein, David, et al. "GPQA: A graduate-level google-proof q&a benchmark." First Conference on Language Modeling. 2024

[5] Comanici, Gheorghe, et al. "Gemini 2.5: Pushing the frontier with advanced reasoning, multimodality, long context, and next generation agentic capabilities." arXiv 2025.


---

---

### Author Response · Authors · 2025-11-29
**Letter to AC and Reviewers (1/2)**

Dear AC and Reviewers,

We sincerely appreciate your time and efforts in reviewing our paper. Due to the unfortunate incident during the rebuttal period, ICLR decided to halt the discussion between reviewers and authors and to reassign the corresponding AC. Accordingly, we would like to provide a summary of our rebuttal for the new AC.

We first summarize **several strengths** recognized by reviewers:
- the **first unified evaluation of four reward model variants across diverse domains** (TwoX, ueA4, YvuS),
- our **clear presentation** (TwoX, ueA4, YvuS, WbK9),
- the **practical value for selecting external verifiers** (TwoX, WbK9), and
- our **theoretical and empirical analysis of PRM failure modes** (TwoX, ueA4, YvuS).

We next summarize **how we addressed each reviewer’s concerns**, which we believe have been successfully resolved. Furthermore, if the new AC has any remaining concerns or questions raised by the reviewers, we would be happy to address them.

---

### Reviewer TwoX

> **[Q1]** The reviewer raised a concern about **limited task diversity**.

- We clarify the **scope of this study**, which follows prior work on reasoning and test-time scaling [1, 2, 3], and we provide our hypothesis that our conclusions **can generalize to open-ended generation tasks**.
- We also conduct additional experiments on **GPQA-diamond** [4] and observe a consistent conclusion that **generative outcome verification delivers robust gains**.

---

### Reviewer ueA4

> **[Q1]** The reviewer claimed that the difference in **label quality might undermine the validity of our conclusions**.

- We clarify that rather than undermining our conclusions, this observation is itself our **main contribution** along with our analysis of PRM degradation in `§4`:
  (i) the increasing **risk of PRMs with longer CoT lengths** (`§4.1`),
  (ii) **label noise** in PRM training data (`§4.2`).

> **[Q2]** The reviewer raised a concern about **limited task diversity**.

- We clarify the **scope of this study**, which follows prior work on reasoning and test-time scaling [1, 2, 3], and we provide our hypothesis that our conclusions **can generalize to open-ended generation tasks**.
- We also conduct additional experiments on **GPQA-diamond** [4] and observe a consistent conclusion that **generative outcome verification delivers robust gains**.

> **[Q3]** The reviewer raised a concern about **weak theoretical results**.

- We agree that the theoretical results may not be ground-breaking.
- However, we emphasize that the **integrated analysis**, combining theoretical, empirical, and interpretative perspectives to explain **why PRMs fail in multi-domain settings**, has not been presented before.



> **[Q4]** The reviewer raised a concern about the **performance of PRM variants with improved label quality**.

- We conduct additional experiments by **refining labels with Gemini-2.0 Flash** [5], and observe that the performance of $\texttt{dPRM}$ improves slightly while that of $\texttt{gPRM}$ does not. This is consistent with our current conclusions.
- We attribute this to **imperfections in the LLM used to generate the training data**.


> **[Q5]** The reviewer raised a concern about the **impact of "aha" CoTs** in multi-domain settings.

- We clarify **the definition of "aha" CoTs** in our study and explain why we **cannot strictly confirm that they are the primary factor**.
- Instead, we view them as **one of several potential risks**, thereby recommending **generative outcome verification over PRMs**.


---

---

### Meta-Review · Area_Chair_yeMd · 2026-01-09

**Summary:**

This paper compares the use of process reward models vs outcome reward models for scoring language model outputs.  The paper challenges the commonly held belief that process reward models outperform outcome reward models (because PRMs aggregate rewards stepwise through the whole sequence instead of just at the end they are commonly seen as better).  The authors argue that although PRMs provide intermediate (step-wise) rewards this also causes them to accumulate error, which compounds over the length of the sequence. The reviews were rather mixed with mostly borderline scores of 8, 6, 6, 4.  The reviewers uniformly praised the clarity and presentation of the paper ("fancy").  They found the topic interesting and worthy of study and the experiments "diverse" and "comprehensive".  However, the reviewers seemed concerned with the scope of the experiments, i.e. confined to verifiable rewards in limited domains.  Also, the they found the paper scientifically valuable, but multiple reviewers seemed to find the results somewhat unsurprising.  Reviewer wbk9 gives a good summary of what seems like a common sentiment among reviewers ("I have mixed feelings about this paper. While, after reading it, I think some of the results are obvious, I understand it's an important part of science for someone to conduct a well-controlled experiment to document and prove such obviousness.").

Unfortunately a confidence weighted average would drive the average score down, since the most confident reviewer gave a 4 and the least confident gave an 8.  Also, while one of the reviewers was willing to change their score, they indicated that they were willing to give a 7 but didn't seem to be comfortable giving an accept (8).  This suggests the reviewer still found the paper more borderline.

**Reviewer Concerns:**

Reviewer wbk9 seemed concerned that the experiments weren't comprehensive enough to justify the claims of the paper.  However, they indicated that they found the author response convincing ("Thanks to the authors for the comprehensive additional experiments. I would like to give an updated rating of 7, but since that option isn't available in the system, I will instead raise the contribution score from 2 to 4.").  It is perhaps worth noting that the reviewer was not willing to raise their score to 8.

The authors provided additional experiments (GPQA-diamond), which provided some more empirical strength behind their conclusions.  They also put in considerable effort to address reviewer comments point-by-point.  However, they didn't really address what I believe are the core concerns of the reviewers, which were the too narrow scope and what the reviewers refer to as significance or "obviousness" of the theory / conclusions.

**Reviewer Scores:**

wbk9 seemed willing to raise their score to 7 but not 8.

For the other reviewers, it doesn't seem like the core concerns were substantively addressed.  Of course, it seems like changing the conclusions or significantly increasing the scope of the paper isn't really feasible.

---

### Decision · Program_Chairs · 2026-01-26

Reject